**Improving hydrological projection performance under contrasting**
**climatic conditions using spatial coherence through a hierarchical**
**Bayesian regression framework**
**Zhengke Pan[a,b], Pan Liu[a,b,*], Shida Gao[a,b], Jun Xia [a,b,c], Jie Chen [a,b], Lei Cheng [a,b]**
[a]State Key Laboratory of Water Resources and Hydropower Engineering Science, Wuhan
University, Wuhan 430072, China
[b]Hubei Provincial Key Lab of Water System Science for Sponge City Construction, Wuhan
University, Wuhan, Hubei, China
[c]Chinese Academy of Sciences, Beijing 100864, China
*Corresponding author. Email: liupan@whu.edu.cn; Tel: +86-27-68775788; Fax:

16 +86-27-68773568

## ABSTRACT

Understanding the projection performance of hydrological models under contrasting climatic conditions supports robust decision making, which highlights the need to adopt time-varying parameters in hydrological modeling to reduce performance degradation. Many existing literatures model the time-varying parameters as functions of physically-based covariates; however, a major challenge remains in finding effective information to control the large uncertainties that are linked to the additional parameters within the functions. This paper formulated the time-varying parameters for a lumped hydrological model as explicit functions of temporal covariates and used a hierarchical Bayesian (HB) framework to incorporate the spatial coherence of adjacent catchments to improve the robustness of the projection performance. Four modeling scenarios with different spatial coherence schemes, and one scenario with a stationary scheme for model parameters, were used to explore the transferability of hydrological models under contrasting climatic conditions. Three spatially adjacent catchments in southeast Australia were selected as case studies to examine the validity of the proposed method. Results showed that (1) the time-varying function improved the model performance but also amplified the projection uncertainty compared with stationary setting of model parameters; (2) the proposed HB method successfully reduced the projection uncertainty and improved the robustness of model performance; and (3) model parameters calibrated over dry years were not suitable for predicting runoff over wet years because of a large degradation in projection performance. This study improves our understanding of the spatial coherence of time-varying parameters, which will help improve the projection performance under differing climatic conditions.

**Keywords:** Climate change; Hierarchical Bayesian; Hydrological model parameters; Spatial coherence; Streamflow projection; Contrasting climatic conditions

## 1. INTRODUCTION

Long-term streamflow projection is an important part of effective water resources planning because it can predict future scarcity in water supply and help prevent floods. Streamflow projections typically involve the following: (i) calibrating hydrological model parameters with partial historical observations (e.g., precipitation, evaporation, and streamflow); (ii) projecting streamflow under periods that are outside of those for model calibration; and (iii) evaluating the model projection performance with certain criteria. One of the most basic assumptions of this process—that the calibrated model parameters are stationary and can be applied to predict catchment behaviors in the near future, has been widely questioned (Brigode et al., 2013; Broderick et al., 2016; Chiew et al., 2014; Chiew et al., 2009; Ciais et al., 2005; Clarke, 2007; Cook et al., 2004; Coron et al., 2012; Deng et al., 2016; Merz et al., 2011; Moore and Wondzell, 2005; Moradkhani et al., 2012; Moradkhani et al., 2005; Pathiraja et al., 2016; Pathiraja et al., 2018; Patil and Stieglitz, 2015; Westra et al., 2014; Xiong et al., 2019; Zhang et al., 2018).

Many previous studies have explored the transferability of stationary parameters to periods with different climatic conditions. They have concluded that hydrological model parameters are sensitive to the climatic conditions of the calibration period (Chiew et al., 2014; Chiew et al., 2009; Coron et al., 2012; Merz et al., 2011; Renard et al., 2011; Seiller et al., 2012; Vaze et al., 2010). For instance, Merz et al. (2011) calibrated model parameters using six consecutive 5-year periods between 1976 and 2006 for 273 catchments in Austria and found that the calibrated parameters

representing snow and soil moisture processes showed a significant trend in the study
area. Other studies have found that degradation in model performance was directly
related to the difference in precipitation between the calibration and verification
periods (Coron et al., 2012; Vaze et al., 2010). One proposal for managing this
problem is to calibrate model parameters in periods with similar climatic conditions to
the near future, but future streamflow observations are unavailable. Thus, it is still
necessary to reduce the magnitude of performance loss and improve the robustness of
the projection performance using calibrated parameters based on the historical records,
even though the climatic conditions in the future may be dissimilar to those used for
model calibration.
Several recent studies have found that hydrological models with time-varying
parameters exhibited a significant improvement in its projection performance
compared with the stationary parameters (Deng et al., 2016; Deng et al., 2018; Westra
et al., 2014). The functional method is one of the most promising ways to model
time-varying parameters and shows its excellence in improving the model projection
performance (Guo et al., 2017; Westra et al., 2014; Wright et al., 2015). This method
models the time-varying parameter(s) as the function(s) of physically-based
covariates (e.g., temporal covariate and Normalized Difference Vegetation Index).
Generally, the hydrological model is run with various assumed functions, the best
functional forms of time-varying parameters can be obtained by comparing the
evaluation criteria. However, a major challenge for the application of the functional
method remains in finding effective information to control the large uncertainties that
are linked to the additional parameters describing these regression functions.

95        Similarity of adjacent catchments has been verified its validity in controlling the

estimation uncertainty of model parameters (Bracken et al., 2018; Cha et al., 2016;
Cooley et al., 2007; Lima and Lall, 2009; Najafi and Moradkhani, 2014; Sun and Lall,
2015; Sun et al., 2015; Yan and Moradkhani, 2015). The level of similarity of
different catchments is known as spatial coherence. For instance, Sun and Lall (2015)
used the spatial coherence of trends in annual maximum precipitation in the United
States, and successfully reduced the parameter estimation uncertainty in their at-site
frequency analysis. In general, there are three methods to consider the spatial
coherence between different catchments in parameter estimation. The first one is no
pooling, which means every catchment is modeled independently, and all parameters
are catchment-specific. The second one is complete pooling, which means all
parameters are considered to be common across all catchments. The third/last one is
hierarchical Bayesian (HB) framework, also known as partial pooling, which means
some parameters are allowed to vary by catchments and some parameters are assumed
to drown from a common hyper-distribution across the region that consists of
different catchments. In these three approaches, the HB framework has been proved
as the most efficient method to incorporate the spatial coherence to reduce the
estimation uncertainty because it has the advantage of shrinking the local parameter
toward the common regional mean and including an estimation of its variance or
covariance across the catchments (Bracken et al., 2018; Sun and Lall, 2015; Sun et al.,
2015). In the field of hydrological modeling, most proceeding literatures were focused

on no pooling models that neglect the spatial coherence between catchments (Heuvelmans et al., 2006; Lebecherel et al., 2016; Merz and Bloschl, 2004; Oudin et al., 2008; Singh et al., 2012; Tegegne and Kim, 2018; Xu et al., 2018); little attention has been paid to the HB framework. Thus, we want to fill this gap and explore the applicability of the spatial coherence through the HB framework in hydrological modeling with the time-varying parameters.

The objectives of this paper were to: (1) verify the effect of the time-varying model parameter scheme on model projection performance and uncertainty analysis compared with stationary model parameters; (2) verify the projection performance of considering spatial coherence of adjacent catchments through the HB framework compared with spatial incoherence; and (3) compare the model projection performance for different climatic transfer schemes.

The rest of the paper is organized as follows. Section 2 outlines the methodology employed in this study including differential split-sample test (DSST) for segmenting the historical series, the hydrological model, and the two-level HB framework for incorporating spatial coherence from adjacent catchments. Section 3 presents the information on the study area and data. The results and discussion are described in section 4. Section 5 summarizes the main conclusions of the study.

## 2. METHODOLOGY

The methodology is outlined by a flowchart in Figure 1, and is summarized as follows:

(1) A temporal parameter transfer scheme is implemented (described in section

2.1) using a classic DSST procedure in which the available data are divided into wet

and dry years;

(2) A daily conceptual rainfall-runoff model is used (outlined in section 2.2);

(3) A two-level HB framework is used to incorporate spatial coherence in

hydrological modeling (described in section 2.3). The process layer (first level) of the

framework models the temporal variation in the model parameters using a

time-varying function, while the prior layer (second level) models the spatial

coherence of the regression parameters in the time-varying function. Four modeling

scenarios with different spatial coherence schemes, and one scenario with a stationary

scheme for the model parameters, are used to evaluate the transferability of

hydrological models under contrasting climatic conditions;

(4) Likelihood function and parameter estimation methods are applied (outlined

in section 2.4); and

(5) The criteria are used to evaluate the model performance for various model

scenarios (described in section 2.5).

## 2.1 Differential split sampling test

To verify the projection performance of the rainfall-runoff model under

contrasting climatic conditions (wet and dry years), a classic DSST using annual

rainfall records was adopted.

Two separate tasks were needed to develop the DSST method into a working

system. The first step was to define "dry years". The method to define the dry years is

adopted from Saft et al. (2015), which is a rigorous identification method that treats

autocorrelation in the regression residuals, undertakes global significance testing, and defines the start and end of the droughts individually for each catchment. Saft et al. (2015) tested several algorithms for dry years delineation, which considered different combinations of dry run length, dry run anomaly and various boundary criteria, and found that the identification results of dry years by one of the algorithms showed marginal dependence on the algorithm and the main results were robust to different algorithms. The detailed processes could be found on Saft et al. (2015) and also are generalized as follows.

Firstly, the annual rainfall data were calculated relative to the annual mean, and the anomaly series was divided by the mean annual rainfall and smoothed with a 3-year moving window. Secondly, the first year of the drought remained the start of the first 3 years negative anomaly period. Thirdly, the exact end date of the dry years was determined through analysis of the unsmoothed anomaly data from the last negative 3-year anomaly. The end year was identified as the last year of this 3 year period unless: (i) there was a year with a positive anomaly >15% of the mean, in which case the end year is set to the year prior to that year; or (ii) if the last two years have slightly positive anomalies (but each <15% of the mean), in which case the end year is set to the first year of positive anomaly; (iii) to ensure that the dry years are sufficiently long and severe, in the subsequent analysis, the authors use dry years with the following characteristics: length $\geq$ 7 years; mean dry years anomaly<-5%.

In the second step, the wet years were defined as the complement of the dry years in the historical records. A similar approach to define the dry and wet years was

used by Fowler et al. (2016).
In the DSST method, the model parameters calibrated in the wet years were
evaluated in the dry years, and vice versa. In addition, criteria, i.e, NSE$_{sqrt}$, BIAS, DIC,
MaxF, and MinF illustrated in section 2.5, were used to evaluate the performance of
the calibrated parameters for different transfer schemes.

## 2.2 The rainfall-runoff model

The hydrological model used in this study is the GR4J (modèle du Génie Rural à
4 paramètres Journalier), which is a lumped conceptual rainfall-runoff model (Perrin
et al., 2003). The original version of the GR4J model (Figure 2) comprised four
parameters (Perrin et al., 2003): production store capacity ($\theta_1$ mm), groundwater
exchange coefficient ($\theta_2$ mm), 1-day-ahead maximum capacity of the routing store
($\theta_3$ mm), and the time base of the unit hydrograph ($\theta_4$ days). More details on the
GR4J model can be found in Perrin et al. (2003).
The GR4J model is a parsimonious, but efficient model. The model has been
used successfully across a wide range of hydro-climatic conditions across the world,
including the crash testing of model performance under contrasting climatic
conditions (Coron et al., 2012), and the simulation of runoff for revisiting the
deficiency in insufficient model calibration (Fowler et al., 2016). For example, Fowler
et al. (2016) verified that conceptual rainfall-runoff models were more capable under
changing climatic conditions than previously thought. These characteristics make the
GR4J particularly suitable as a starting point for implementing modifications and/or
improving predictive ability under changing climatic conditions.

## 2.3 The HB framework for the time-varying model parameter

In this study, various versions were constructed for evaluating the projection capabilities of models for contrasting climatic conditions (wet and dry years), and for considering the temporal variation and spatial coherence of parameter $\theta_1$ .

2.3.1 Process layer: temporal variation of the model parameter

As described in the literature (Pan et al., 2019; Perrin et al., 2003; Renard et al., 2011; Westra et al., 2014), parameter $\theta_1$ , which represents the primary storage of water in the catchment, is the most sensitive parameter in the GR4J model structure, and the stochastic variations of this parameter have the largest impact on model projection performance (Renard et al., 2011; Westra et al., 2014). In addition, the temporal variation in the catchment storage capacity was physically interpretable. Periodic variations in the production store capacity $\theta_1$ can be induced by the periodicity in precipitation (Pan et al., 2018) and in seasonal vegetation growth and senescence. In the present study, $\theta_1$ was constructed to account for the periodical variation that had a significant impact on the extensionality of the model. The periodical variation in catchment storage capacity $\theta_1$ is described by a sine function, using amplitude and frequency.

Thus, for any catchment $c$, the full temporal regression function for $\theta_1$ at the process layer is:

$$\text{Process layer:} \qquad \theta_1(c,t) = \alpha(c) + \beta(c)\sin\left[\omega(c)t\right] \qquad (1)$$

where $\alpha$, $\beta$, $\omega$ are regression parameters for the specific DSST method, and $\alpha$ signifies the intercept, and $\{\beta,\omega\}$ represents the amplitude and frequency of the

sine function, respectively. *t* is the time step. According to the definition of the GR4J
model (Perrin et al., 2003), the value of $\theta_1$ must be a positive value. If model
parameter $\theta_1$ is constant then $\beta=0$, $\alpha>0$ suffice in Eq.1. Meanwhile, the value
of $\omega$ becomes irrelevant. Thus, the resulting model simplifies to a stationary
hydrological model.
2.3.2 Prior layer: spatial coherence of regression parameters
For a heterogeneous region that is distinctly non-uniform in climatic and
geologic conditions, different catchments within the region typically have different
catchment storage capacities and different values of production store capacity $\theta_1$.
For a homogeneous region prescribed by similar climatic and geologic conditions in
each part, the production store capacity (in Eq. 1) is expected to be the same among
different catchments of the region. The model could be improved by considering
spatial input, i.e., the spatial coherence of parameters across adjacent catchments
(Chen et al., 2014; Lima et al., 2016; Merz and Bloschl, 2004; Oudin et al., 2008;
Patil and Stieglitz, 2015; Renard et al., 2011; Sun et al., 2014).
In this study, independent Gaussian prior distributions were used for the
amplitude $\beta$ and frequency $\omega$ at the prior layer to include the potential spatial
coherence. Their equations are as follows:
Prior layer:
$$\begin{aligned}\beta(\text{c}) &= N\left(\mu_2,\sigma_2^{\,2}\right)\\\omega(\text{c}) &= N\left(\mu_3,\sigma_3^{\,2}\right)\end{aligned} \qquad (2)$$

where $\mu_2$, $\mu_3$, $\sigma_2$ and $\sigma_3$ are hyper-parameters, and $N(.)$ represents the
hyper-distribution, i.e., a Gaussian distribution. Independent Gaussian distributions
were assumed for the amplitude $\beta$ and frequency $\omega$ that were used to model
spatial coherence based on practical considerations. The prior layer of the HB
framework aims to describe the variation of $\{\beta, \omega\}$ in space by means of a Gaussian
spatial process in which the mean value depends on covariates describing regional
characteristics. Amplitude $\beta$ and frequency $\omega$ are the most important parameters
in the regression function and can reflect the spatial connection of variation and
cyclicity of catchment production storage capacity among catchments. The Gaussian
distribution is one of the widely used distributions for describing the prior layer
within the HB framework and has been applied in many previous studies, such as Sun
et al (2015, 2016) and Chen et al (2014). In addition, the introduction of the Gaussian
distributions to describe the spatial coherence of $\beta$ and $\omega$ also because that there
are still uncountable factors that may have impacts on the spatial coherence between
adjacent catchments, which might make the coherence tend to converge a central
value but with finite variance, and obey the Central limit theorem.
2.3.3 Modeling scenarios
Five modeling scenarios (Table 1) were carried out to assess the effect of spatial
coherence on the time-varying function. Different levels of spatial coherence of
$\{\beta, \omega\}$ were assumed in scenarios 1 to 4, while in scenario 5 parameter $\theta_1$ was set
to be constant to provide a comparison. It should be noted that the estimates for
spatially coherent regression parameters would be shared by different catchments
while other quantities would be regarded as catchment-specific variables. For
example, amplitude $\beta$ is spatially linked in scenario 1, i.e., $\beta(c) = N(\mu_2, \sigma_2^2)$, which
means that the estimates of $\beta$ are shared by all catchments. Meanwhile, regression
parameters $\omega_{1\text{-}1}$, $\omega_{1\text{-}2}$, and $\omega_{1\text{-}3}$ are used as independent variables to represent the
frequency of model parameter $\theta_1$ in different catchments. The number of unknown
quantities in different scenarios are as follows: fifteen in scenarios 1 and 2, thirteen in
scenario 3 and eighteen in scenario 4. The prior ranges of all unknown quantities
(including model parameters ($\theta_2, \theta_3$, and $\theta_4$), regression parameters $\alpha$, $\beta$ and
$\omega$, and hyper-parameters $\mu_2$, $\sigma_2$, $\mu_3$ and $\sigma_3$) in different scenarios and both
DSST schemes could be found in Table S1 in Supplement material. It should be noted
that in a specific scenario, some unknown quantities might not exist. For example, $\mu_3$
and $\sigma_3$ did not exist in scenario 1 while $\mu_2$ and $\sigma_2$ did not exist in scenario

279      2.

## 280    2.4 Estimation and projection

The objective function and parameter inference methods were used to derive the
posterior distribution of all unknown quantities, as illustrated below.
2.4.1 Objective function
For a specific catchment, the model parameters were calibrated to minimize the
following objective function, which was adopted from Coron et al. (2012).
$$\varepsilon_c\left[\theta_1,\theta_2,\theta_3,\theta_4\right] = -RMSE\left[\sqrt{Q}\right]\left(1+\left|1+BIAS\right|\right) \tag{3}$$
where
$$RMSE\left[\sqrt{Q}\right] = \sqrt{\frac{1}{T}\sum_{t=1}^{T}\left[Q_{sim}(t)-Q_{obs}(t)\right]^2} \tag{4}$$
and $RMSE\left[\sqrt{Q}\right]$ refers to the root-mean-square error, in which $Q_{sim}$ is derived by
the adopted hydrological model. $T$ represents the number of the time series while $t$ is
the time step.
Coron et al. (2012) showed that this objective function performed well. In this
function, the combination of $RMSE\left[\sqrt{Q}\right]$ and *BIAS* (Eq.7) gives weight to dynamic
representation as well as the water balance. Using square-root-transformed flows to
compute the RMSE reduces the influence of high flows during the calibration period
and provides a good compromise between alternative criteria.
In the case of multiple catchments, the objective function of the HB framework
was the product of Eq.3 and the conditional probability of spatial coherence of
regression parameters $f_N$. It was written as follows:

$$Scenario\ 1:\ \Lambda=\prod_{c=1}^{C}\varepsilon_c\left[\theta_1(t,c),\theta_2(c),\theta_3(c),\theta_4(c)\middle|\alpha(c),\beta,\omega(c)\right]\bullet f_N\left(\beta\middle|\mu_2,\sigma_2\right)$$

$$Scenario\ 2:\ \Lambda=\prod_{c=1}^{C}\varepsilon_c\left[\theta_1(t,c),\theta_2(c),\theta_3(c),\theta_4(c)\middle|\alpha(c),\beta(c),\omega\right]\bullet f_N\left(\omega\middle|\mu_3,\sigma_3\right)$$

$$Scenario\ 3:\ \Lambda=\prod_{c=1}^{C}\varepsilon_c\left[\theta_1(t,c),\theta_2(c),\theta_3(c),\theta_4(c)\middle|\alpha(c),\beta,\omega\right]\bullet\prod_{n=1}^{2}f_N\left(\beta,\omega\middle|\mu_2,\sigma_2,\mu_3,\sigma_3\right)\quad(5)$$

$$Scenario\ 4:\ \Lambda=\prod_{c=1}^{C}\varepsilon_c\left[\theta_1(t,c),\theta_2(c),\theta_3(c),\theta_4(c)\right]$$

$$Scenario\ 5:\ \Lambda=\prod_{c=1}^{C}\varepsilon_c\left[\theta_1(c),\theta_2(c),\theta_3(c),\theta_4(c)\right]$$

where the number of catchments in the region is represented by C, and the Gaussian
spatial function between regression parameters $\beta,\omega$ and hyper-parameters $\mu_2$,
$\mu_3$, $\sigma_2$ and $\sigma_3$ are denoted by $f_N()$. N refers to the Gaussian distribution and
*n* represents the number of regression parameters that are spatially coherent.
2.4.2 Inference
The uniform distribution is used as the prior distribution for hyper-parameters
and spatially irrelevant parameters. Meanwhile, spatially relevant parameters are
sampled from the Gaussian distributions. Because the prior distribution has no impact
on the final evaluation of different scenarios, the prior distributions are not presented
in Eq.5. The likelihood functions defined in Eqs. 3 and 5 pose a computational
challenge because their dimensionality grows (primarily related to the number of
catchment-specific parameters) with the number of catchments considered. The
unknown quantities, including model parameters ($\theta_2$, $\theta_3$, and $\theta_4$), regression
parameters $\alpha$, $\beta$ and $\omega$, and hyper-parameters $\mu_2$, $\sigma_2$, $\mu_3$ and $\sigma_3$ (if
presents), are sampled and estimated simultaneously using the Shuffled Complex
Evolution Metropolis (SCEM-UA) sampling method (Ajami et al., 2007; Vrugt et al.,
2003; Vrugt et al., 2009). The SCEM-UA sampling method is a widely used Markov
Chain Monte Carlo algorithm for simulating the posterior probability distribution of
parameters that are conditional on the current choice of parameters and data. When
compared with traditional Metropolis-Hasting samplers, the SCEM-UA algorithm
more efficiently reduces the number of model simulations needed to infer the
posterior distribution of parameters, (Ajami et al., 2007; Duan et al., 2007; Liu et al.,
2014; Liu and Gupta, 2007; Vrugt et al., 2003). Convergence is assessed by evolving
three parallel chains with 30000 random samples, the posterior distributions of
parameters are evaluated by the Gelman-Rubin convergence value and are confirmed
that the convergence value is smaller than the threshold 1.2 (Gelman et al., 2013).

## 327    2.5 Model performance criteria

Five criteria were used to assess the projection performance during the
verification periods.
(1) The first criterion was NSE$_{sqrt}$, known as the arithmetic square root of
Nash-Sutcliffe Efficiency (Coron et al., 2012; Moriasi et al., 2007; Nash and Sutcliffe,
1970). When compared with the classic NSE, NSE$_{sqrt}$ gives an intermediate, more
balanced picture of the overall hydrograph fit because it can reduce the influence of
high flow. It is expressed as:
$$NSE_{sqrt} = 1 - \frac{\sum_{t=1}^{T}\left[\sqrt{Q_{obs}(t)} - \sqrt{Q_{sim}(t)}\right]^2}{\sum_{t=1}^{T}\left[\sqrt{Q_{obs}(t)} - \sqrt{\overline{Q}_{obs}}\right]^2} \tag{6}$$

where $Q_{sim}(t)$ and $Q_{obs}(t)$ represent the simulated and observed daily streamflow
values for the $t^{th}$ day, respectively; $\overline{Q}_{obs}$ is the mean of the observed daily streamflow
for the calculation interval, and $T$ refers to the length of the calculation period.
(2) The second criterion is the BIAS, one of the most popular indexes to reflect
the deviation degree between the modeled runoff and observations, also is a part of
the objective function Eq.3.
$$BIAS = \frac{\sum_{t=1}^{T}\left[Q_{sim}(t) - Q_{obs}(t)\right]}{\sum_{t=1}^{T}\left[Q_{obs}(t)\right]} \tag{7}$$

(3) The third criterion is the Deviance information criterion (DIC), which was
defined by Spiegelhalter et al. (2002). It is a widely used and popular measure
designed for Bayesian model comparison and is a Bayesian alternative to the standard
Akaike Information Criterion. The DIC value for a Bayesian scenario is obtained as:
$$DIC = -2\log\left(p\left(q|\theta_{Bayes},\xi\right)\right) + 2p_{DIC} \tag{8}$$

where $p_{DIC}$ is the effective number of parameters, defined as
$$p_{DIC} = 2\left(\log\left(p\left(q|\theta_{Bayes},\xi\right)\right) - \frac{1}{S}\sum_{s=1}^{S}\log\left(p\left(q|\theta^s,\xi\right)\right)\right) \tag{9}$$

where *p* refers to probability, q represents the observations of streamflow and $\xi$
denotes the time series of model input, e.g., rainfall and potential evapotranspiration.
Posterior mean $\theta_{Bayes}=\text{Expect}(\theta|q,\xi)$ and s=1,…, S, means the sequence number of
the simulated parameter set $\theta^s$ by the adopted SCEM-UA algorithm. According to
Spiegelhalter et al. (2002), scenarios with smaller DIC would be preferred to
scenarios with larger DIC.
(4) The fourth and fifth criteria are the Mean annual maximum flow (MaxF,
mm/d) and Mean annual minimum flow (MinF, mm/d), which are used to qualify the
performance of the high flows and low flows. These criteria are self-explanatory and
have been used in many studies to assess the magnitude of maximum and minimum
levels of flows (Ekstrom et al., 2018). The scenarios with the least absolute variation
between the modeled values and the observed values are recognized as the best
scenarios.

## 3. Study area and data

To evaluate the model performance, we used daily precipitation (mm/day),
potential evapotranspiration (mm/day), and streamflow (mm/day) time series records
for three unregulated and unimpaired catchments in south-eastern Australia, taken
from the national dataset of Australia (Zhang et al., 2013), covering 1976–2011. The
streams were unregulated: they were not subject to dam or reservoir regulations,
which can reduce the impact of human activity. The observed streamflow record
contained at least 11835 daily observations (equivalent to record integrity of greater
than 90%) for 1976–2011, with acceptable data quality. The first complete year of
data was used for model warm-up to reduce the impact of the initial soil moisture
conditions during the calibration period.

374       The attributes of the south-eastern Australian catchments are shown in Table 2

and Figure 3. The IDs of these catchments are 225219 (Glencairn station on the
Macalister River: mean annual rainfall, potential evapotranspiration, and runoff are
1106 mm, 1184 mm, and 368 mm, respectively), 405219 (Dohertys station on the
Goulburn River: mean annual rainfall, potential evapotranspiration, and runoff are
1171 mm, 1196 mm, and 420 mm, respectively), and 405264 (D/S of Frenchman Ck
Jun station on the Big River: mean annual rainfall, potential evapotranspiration, and
runoff are 1408 mm, 1160 mm, and 465 mm, respectively). As shown in Figure 3,
these catchments are adjacent to each other. All catchments experienced a severe
multiyear drought around the end of the millennium. Saft et al. (2015) identified that
the rainfall-runoff relationship in these catchments was altered during the long-term
drought.

## 4. Results and discussion

387       Results from the DSST were used to assess the model projection performance for

five scenarios under contrasting climatic conditions. First, a DSST was conducted in
each catchment to divide original records into wet and dry years. Then, the projection
performance for the five scenarios and associated parameter uncertainties were
evaluated using the criteria described above.

## 4.1 Dry years identification

As illustrated in Table 3 and Figure 4, the drought definition method identified that the three catchments had similar dry years characteristics, with the same drought start (1997) and end (2009) points. The length of dry years for the studied catchments is same, 13 years. The mean dry years' anomaly was more severe in the Macalister catchment (225219), with an 11.70% reduction in the mean dry years' anomaly while the other two catchments experienced reductions of 11.16% (405219) and 11.14% (405264).

In terms of changes in rainfall, on average catchments had an 11% reduction from the wet years to the dry years (Table 3). Meanwhile, these catchments experienced a 26.3% decrease in runoff during the dry years, which is much more severe than the reduction in rainfall. The similar findings can be derived out from the comparison of runoff coefficients of different periods, that is, all catchments experienced a decrease in its runoff coefficients during the dry years.

## 4.2 Model performance in five scenarios

As shown in Figures 5(a), 6(a) and 7, the calibrated model parameters yielded good simulation performance over the calibrated periods for all criteria. For example, the mean $NSE_{sqrt}$ score during the calibration period across these catchments remained close to about 0.7 or slightly higher, regardless of which scenario was chosen. However, when the same parameter sets were verified by simulating streamflow over drier or wetter years, the model performance was degraded, including both the robustness and accuracy of projection performance. Furthermore, the magnitude of

performance loss increases along with the variation in rainfall between the calibration
and verification periods.

416       Figure 5 shows the $NSE_{sqrt}$ performance for calibration in wet years and

verification in the dry years for each scenario in all catchments. All scenarios
performed well in all catchments with the mean $NSE_{sqrt}$ reaching 0.81 during the wet
calibration period, and then all scenarios experienced a slight decrease in performance
($NSE_{sqrt}$ = 0.75) during the dry verification period. Scenario 4 (time-varying
parameters without spatial inputs) or scenario 5 (temporally stable parameters)
generally performed better during the calibration period than the scenarios that
considered different levels of spatial coherence for the regression parameters. During
the verification period, the $NSE_{sqrt}$ rank order changed (Figure 5b). Scenario 4 had a
higher median $NSE_{sqrt}$ performance than scenario 5 in catchments 225219 and 405264.
Although the median estimate in scenario 4 was slightly inferior to the latter in
catchment 405219, its distribution of the $NSE_{sqrt}$ performance was much more
positively biased from the median estimates than scenario 5. Furthermore, the former
reaches higher $NSE_{sqrt}$ performance than the latter when comparing the top $NSE_{sqrt}$
performance of these two scenarios. Thus, it indicates the validity of the time-varying
scheme for improving model performance. However, the introduction of additional
regression parameters ($\alpha, \beta$ and $\omega$) at the same time amplified the model projection
uncertainty in two of three catchments (405219 and 405264) when comparing results
from scenarios 4 and 5. Fortunately, the appropriate adoption of spatial coherence
alleviates this problem. In the DSST scheme of calibrating in the wet years and

verifying in the dry years, scenario 2 exhibited the smallest fluctuation range of

NSE$_{sqrt}$ estimate in catchments 405219 and 405264 and was the second-best scenario

in catchment 225219. Conversely, scenario 3 exhibited the smallest fluctuation range

of NSE$_{sqrt}$ estimate in catchment 225219, and was the second-best scenario in

catchments 405219 and 405264. As for the median NSE$_{sqrt}$ estimate, scenario 2 is the

best scenario (which showed the best performance in catchment 225219 and 405219,

but it was the fourth in catchment 405264), followed by scenario 3 (which is the

second-best scenario in catchments 405219 and 405264 and is the third in catchment

225219). In addition, the highest median NSE$_{sqrt}$ performance in scenarios 4 and 5

during the calibration period did not guarantee the same superior performance during

the verification period. This illustrates the deficiency of time-varying and stationary

schemes of model parameters when spatial inputs from adjacent catchments are not

considered.

Similarly, Figure 6 illustrates the NSE$_{sqrt}$ performance for each scenario in all

catchments for calibration in the dry years and verification in the wet years. All

scenarios performed well for all catchments with the mean NSE$_{sqrt}$ reaching 0.75 in

the dry calibration period and 0.79 in the wet verification period. As shown in Figure

6, models experienced a slight improvement in NSE$_{sqrt}$ performance when transferred

from the dry years to the wet years. However, the projection performance calibrated

using a contrasting climatic condition was inferior to the simulation performance that

was directly calibrated from the climatic condition, compared with Figures 5(a) and

6(b), or Figure 6(a) and 5(b). For example, the NSE$_{sqrt}$ performance in Figure 6(b) is

inferior to that in Figure 5(a). By comparing scenarios in the calibration period, it was

found that scenarios 4 and 5 exhibited the highest performance in two of three

catchments (405219 and 405264), followed successively by scenario 3, scenario 2,

and scenario 1. During the verification period, the median $NSE_{sqrt}$ performance in

scenario 4 was 0.80% higher than scenario 5, however, the variation range in scenario

4 was 53% wider than the latter. These results demonstrate that the time-varying

scheme (scenario 4) for model parameters improved the median $NSE_{sqrt}$ performance

but also amplified the projection uncertainty compared with the results from the

stationary scheme (scenario 5) for model parameters. In the DSST scheme of

calibrating in the dry years and verifying in the wet years, scenario 3, which

considered both spatial coherence of $\beta$ and $\omega$ between different catchments,

exhibited the highest median $NSE_{sqrt}$ for all catchments, had the smallest fluctuation

range in two catchments (225219 and 405264) and is the second smallest scenario in

variation in catchment 40519 during the verification period. Conversely, scenario 2,

the scenario with the best median estimate performance during the verification period

in Figure 5, is just the fourth in all five scenarios in this DSST scheme. Compared

with other model scenarios, the incorporation of spatial coherence of both regression

parameters in scenario 3 reduced the projection uncertainty and improved the

robustness of the model performance, with the smallest fluctuation ranges in most

options under the contrasting climatic conditions. It indicates that the spatial setting of

model parameters between different catchments provided a clear input for reducing

the uncertainty of the model projection performance during the verification period. In

addition, it also should be noted that model parameters calibrated over dry years,
contrastively, were not suitable for predicting runoff over wet years because of a
larger degradation in projection performance than the scheme with the adverse
calibration-verification direction.
Comparing the DIC results for both DSST schemes in Table 4 and Table 5, the
best DIC value is achieved by scenario 3, which incorporates the spatial coherence of
both regression parameters and is the most complex scenario in the comparison. This
finding is consistent with the results by the $NSE_{sqrt}$ criterion and showed the validity
of the spatial coherence of both regression parameters in ensuring the robustness of
the hydrological projection performance. In addition, when comparing DIC results of
scenarios 4 and 5, the setting of time-varying functions improved the DIC
performance in both DSST schemes. This finding also agreed with the results by the
$NSE_{sqrt}$ criterion and indicated the positive implications of the time-varying model
parameters on the projection performance.
Tables 6 and 7 illustrate the performance of high and low flows during the
verification period in terms of MaxF and MinF estimates for the median projected
streamflows in both DSST schemes. As shown in table 7, for the projection of high
flow part, scenario 3 exhibits the best performance in all catchments among five
scenarios under the scheme of calibrating in the dry years and verifying in the wet
years. For the projection performance in the other DSST scheme (Table 6), scenario 3
has the best projection performance in high flow part in catchment 225219 and is the
second-best scenario in the other two catchments. It indicates that the incorporation of
spatial coherence of both amplitude $\beta$ and frequency $\omega$ successfully improves
the projection performance in the high flow part. As for the projection of the low flow
part, the discrepancy between the results of different scenarios and the observed low
flows is not obvious (The absolute differences between the observed values and
modeled values are very small). Furthermore, scenario 3 shows the best-projected
performance in two catchments (405219 and 405264) in the scheme of calibrating in
dry years and verifying in wet years, and is the best scenario in catchment 405264 in
the scheme of calibrating in wet years and verifying in dry years. In addition, scenario
3 is the second-best option in catchments 225219 and 405219 under the scheme of
calibrating in wet years and verifying in dry years. Combined with the projection
performance of both high and low flows, scenario 3 achieves its superior projection
performance mainly by the improvement in the prediction of high flow parts.

514        Figure 7 shows the BIAS estimates for the median of the posterior distribution of

model parameters for all modeling scenarios across all catchments when
transferability between the wet and dry years was examined. Although the BIAS was
a component of the objective function (Eq. 3), the 10-year rolling average BIAS still
deviated considerably from a value of 1 for all the scenarios in the two DSST schemes.
The median estimates of the posterior distribution in both scenarios performed well in
the $NSE_{sqrt}$ criterion for both periods. However, the median estimates did not ensure
unbiased simulations over the modeling period; one scenario with a higher $NSE_{sqrt}$
criterion may have an altered BIAS during the modeling period. The BIAS results in
catchments 225219 and 405219 showed some similarity: all scenarios tended to
underestimate streamflow along the time sequence in both DSST schemes. Conversely,
all scenarios tended to overestimate the streamflow in catchment 405264 in both
schemes. By comparing the BIAS performance for the five scenarios, it was observed
that the spatial setting of modeling scenarios generally tended to enlarge the BIAS in
all catchments, while the difference between scenarios 4 and 5 was very small.

## 4.3 Parameter uncertainty analysis

The uncertainty of the parameters was characterized by the posterior distribution
of the regression parameters and was derived by the MCMC iteration. As mentioned
in section 2.3.2, amplitude $\beta$ and frequency $\omega$ were assumed to have different
levels of spatial coherence in each modeling scenario (Table 1); these scenarios in
each DSST regime are compared in Figs. 8 and 9. It should be mentioned that there
was no regression parameter in scenario 5. Solid lines in the violin plots represent the
25th and 75th percentiles of the posterior distribution. The white dots in the violin plot
denote the median estimate of the posterior distribution. In the upper plots in Figures
8 and 9, it can be clearly seen that the first three scenarios had a much smaller
variation interval than scenario 4 in terms of amplitude $\beta$, which denotes the
amplitude of the sine function. The catchment averages of both schemes of the
median estimates of $\beta$ in the first three scenarios are 2.78, -4.91, and 9.26
respectively, while that in the fourth scenario is much larger, reached at -39.20.
Scenario 3, which considered both spatial coherence of amplitude $\beta$ and frequency
$\omega$, has the narrowest interval of $\beta$ for all catchments, followed successively by
scenario 1 (only considered the spatial coherence of the amplitude $\beta$), scenario 2
(only frequency $\omega$ was spatially coherent), and scenario 4 (no regression parameter
was spatially coherent). With regards to the regression parameter $\omega$, which denotes
the frequency of the sine function (in the lower figures of Figures 8 and 9), its median
estimates in both four scenarios differ slightly. As shown in Figure 8, the catchment
averages of frequency $\omega$ for different scenarios are 0.24, 0.14, 0.15, and 0.18, while
those in Figure 9 are 0.15, 0.26, 0.23, and 0.17 respectively. The period $T$ of the sine
term could be derived based on the estimates of $\omega$ by equation $T = 2\pi/\omega$. Thus,
the mean periods $T$ of model parameter $\theta_1$ for different scenarios are 26.2, 46.3,
41.9 and 35.2 in Figure 8, respectively. Similarly, the mean periods $T$ are 42.9, 24.1,
27.4 and 38.0 in Figure 9, respectively. In addition, we used the Hilbert-Huang
Transform method (Huang et al., 1998) to identify the potential periods of the series
of several climate variables (including the daily rainfall, daily potential
evapotranspiration, daily maximum temperature and daily minimum temperature in
the studied catchments). It was found that these daily series have periods of 22.2~49.1
days. Thus, we guess that the potential periods of these climate variables may be the
possible reasons for the periods of time-varying parameters. It also should be
mentioned that the adopted Hilbert spectrum method is one of the most popular
methods for analyzing nonlinear and non-stationary data. Huang et al. (1999)
indicated that this method is better than the Fourier transform method and Wavelet
Transform method in processing nonlinear and non-stationary data.
In summary, by combining the results of parameter uncertainty estimation and
model projection performance evaluation, the incorporation of spatial coherence
successfully improved the robustness of the projection performance in both DSST
schemes by controlling the estimation uncertainty of amplitude $\beta$.
**5. CONCLUSIONS**
In this study, a two-level HB framework was used to incorporate the spatial
coherence of adjacent catchments to improve the hydrological projection performance
of sensitive time-varying parameters for a lumped conceptual rainfall-runoff model
(GR4J) under contrasting climatic conditions. Firstly, a temporal parameter transfer
scheme was implemented, using a DSST procedure in which the available data were
divided into wet and dry years. Then, the model was calibrated in the wet years and
evaluated in the dry years, and vice versa. In the first level of the proposed HB
framework, the most sensitive parameter in the GR4J model, i.e., the production
storage capacity ($\theta_1$), was allowed to vary with time to account for the periodic
variation that had significant impacts on the extensionality of the model. The periodic
variation in catchment storage capacity was represented by a sine function for $\theta_1$
(parameterized by amplitude and frequency). In the second level, four modeling
scenarios with different spatial coherence schemes, and one scenario with a stationary
scheme of catchment storage capacity, were used to evaluate the transferability of
hydrological models under contrasting climatic conditions. Finally, the proposed
method was applied to three spatially adjacent, unregulated, and unimpaired
catchments in southeast Australia. The study concludes that: (1) the time-varying
setting was valid in improving the model performance but also extended the
projection uncertainty in contrast to the stationary setting; (2) the inclusion of spatial

coherence successfully reduced the projection uncertainty and improved the robustness of model performance; and (3) a large performance degradation has been found in the DSST scheme with its model parameters calibrated over dry years and verified in the wet years. This study improves our understanding of the spatial coherence of time-varying parameters, which will help improve the projection performance under differing climatic conditions. However, there are several unsolved problems that need to be addressed. First, the spatial setting of regression parameters may expand the BIAS between the simulation and streamflow observation with a single objective function; the potential physical mechanism behind this result should be explored further. Secondly, this study was confined to spatially coherent catchments that are similar in climatic and hydrogeological conditions; further research is needed to determine which factors have the most significant impacts on model projection performance when considering obvious inputs from other catchments.

## ACKNOWLEDGMENTS

This study was supported by the National Key Research and Development Program (2018YFC0407202), the National Natural Science Foundation of China (51861125102; 51879193), the Research Council of Norway (FRINATEK Project 274310), and Innovation Team in Key Field of the Ministry of Science and Technology (2018RA4014). The numerical calculations were done on the supercomputing system in the Supercomputing Center of Wuhan University. The

authors would like to thank the editor and anonymous reviewers for their comments, and Professor Chong-Yu Xu in the University of Oslo for proofreading the final version, that helped improve the quality of the paper.

## AUTHOR CONTRIBUTIONS

All of the authors helped to conceive and design the analysis. Zhengke Pan and Pan Liu performed the analysis and wrote the paper. Shida Gao, Jun Xia, Jie Chen, and Lei Cheng contributed to the writing of the paper and made comments.

## COMPLIANCE WITH ETHICAL STANDARDS

**Conflict of interest:** The authors declare that they have no conflict of interest.

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

**TABLES**

Table 1. Different spatial coherence scenarios for amplitude β and frequency ω in the time-varying functional form of model parameter $\theta_1$. To explore the performance of spatial coherence within the time-varying function, different levels of spatial coherence for amplitude β and frequency ω were assumed for the first three scenarios; in contrast, no spatial coherence is assumed in scenario 4, and a temporally stable $\theta_1$ is assumed in scenario 5.

| Category | | Scenario | β | ω | Constraints |
|---|---|---|---|---|---|
| Time-varying | Spatial coherence | 1 | Parameter β is region-related | Parameter ω is catchment-specific | $\theta_1=\alpha(c)+\beta(c)\sin[\omega(c)t]$, while $\beta(c)=N(\mu_2, \sigma_2^2)$ |
| | | 2 | Parameter β is catchment-specific | Parameter ω is region-related | $\theta_1=\alpha(c)+\beta(c)\sin[\omega(c)t]$, while $\omega(c)=N(\mu_3, \sigma_3^2)$ |
| | | 3 | Parameter β is region-related | Parameter ω is region-related | $\theta_1=\alpha(c)+\beta(c)\sin[\omega(c)t]$, while $\beta(c)=N(\mu_2, \sigma_2^2)$ and $\omega(c)=N(\mu_3, \sigma_3^2)$ |
| | No spatial coherence | 4 | Parameter β is catchment-specific | Parameter ω is catchment-specific | $\theta_1=\alpha(c)+\beta(c)\sin[\omega(c)t]$ |
| Time invariant | | 5 | No parameters β or ω | | $\theta_1$ is stationary |

NB: $\theta_1$ represents the production storage capacity of the catchment; β is the slope describing long-term change during the modeling period, and ω is the amplitude of the sine function describing its seasonal variation during the modeling period; $\mu_2$, $\sigma_2$, $\mu_3$, $\sigma_3$ are hyper-parameters.

**Table 2. Comparison of catchments attributes in terms of mean annual rainfall (mm), mean annual evaporation (mm), and mean annual**
**runoff (mm) for 1976–2011.**

| Catchments ID | River Name | Observations start | Observations end | Mean annual rainfall | Mean annual potential evapotranspiration | Mean annual runoff |
|---|---|---|---|---|---|---|
| 225219 | Macalister | 1/1/1976 | 30/12/2011 | 1106 | 1184 | 368 |
| 405219 | Goulburn | 1/1/1976 | 30/12/2011 | 1171 | 1196 | 420 |
| 405264 | Big | 1/1/1976 | 30/12/2011 | 1408 | 1160 | 465 |

**Table 3. Drought identification results for the catchments.**

| Catchments ID | Drought start | Drought end | Length | Mean dry years anomaly | % Complete | $R_1$ | $R_2$ | Change in runoff (%) | Change in rainfall (%) |
|---|---|---|---|---|---|---|---|---|---|
| 225219 | 1997 | 2009 | 13 | -11.70% | 91.5% | 0.34 | 0.28 | -27.21 | -11.27 |
| 405219 | 1997 | 2009 | 13 | -11.16% | 99.9% | 0.38 | 0.31 | -26.04 | -10.97 |
| 405264 | 1997 | 2009 | 13 | -11.14% | 98.5% | 0.35 | 0.29 | -25.63 | -10.51 |

NB: $R_1$ and $R_2$ refer to the runoff coefficient during the wet and dry years, respectively.

**Table 4. Comparison of five scenarios in terms of the deviance information criterion (DIC) when model parameters were calibrated in the wet years and verified in the dry years.**

| Category | | Scenario | DIC |
|---|---|---|---|
| Time-varying | Spatial coherence | 1 | 4961.7 |
| | | 2 | 1202.3 |
| | | 3 | -1254.4 |
| Time-invariant | No spatial coherence | 4 | 5052.8 |
| | | 5 | 5827.3 |

**Table 5. Comparison of five scenarios in terms of the deviance information criterion (DIC) when model parameters were calibrated in the dry years and verified in the wet years.**

| Category | | Scenario | DIC |
|---|---|---|---|
| Time-varying | Spatial coherence | 1 | -6167.0 |
| | | 2 | -5743.6 |
| | | 3 | -10574.0 |
| Time-invariant | No spatial coherence | 4 | -8710.0 |
| | | 5 | -7460.8 |

**Table 6. Comparison of the projection performance of median flows during the verification period associated with the Mean annual maximum flow (MaxF, mm/d) and Mean annual minimum flow (MinF, mm/d) when model parameters were calibrated in the wet years and verified in the dry years. The percentage represents the % variation between the modeled value and the observed value.**

| | Mean annual maximum flow | | | Mean annual minimum flow | | |
|---|---|---|---|---|---|---|
| | 225219 | 405219 | 405264 | 225219 | 405219 | 405264 |
| Observed | 10.58 | 11.98 | 9.23 | 0.050 | 0.093 | 0.17 |
| Scenario 1 | +25.7% | -52.9% | -27.7% | **+0.6%** | -51.3% | -25.6% |
| Scenario 2 | -14.6% | **-14.6%** | -20.9% | +7.1% | -35.0% | -18.3% |
| Scenario 3 | **+3.1%** | -36.1% | +5.6% | -17.9% | **-1.1%** | **-6.4%** |
| Scenario 4 | -44.2% | -54.7% | **+3.3%** | +76.6% | -4.4% | -14.4% |
| Scenario 5 | -52.1% | -49.7% | -13.6% | +72.0% | -6.9% | -29.1% |

Note:

1. The data in 1976 has been used for model warm-up to reduce the impact of the initial soil moisture conditions during the calibration period, and is not counted in the table;

2. The scenarios with bold values are labeled as the best scenario for projecting the streamflow during the verification periods, and the values from these scenarios have the least absolute percentage difference with the observed values.

847

**Table 7. Comparison of the projection performance of median flows during the verification period associated with the Mean annual maximum flow (MaxF, mm/d) and Mean annual maximum flow (MinF, mm/d) when model parameters were calibrated in the dry years and verified in the wet years. The percentage represents the % variation between the modeled value and the observed value.**

853

| | Mean annual maximum flow | | | Mean annual minimum flow | | |
|---|---|---|---|---|---|---|
| | 225219 | 405219 | 405264 | 225219 | 405219 | 405264 |
| Observed | 10.73 | 12.06 | 8.94 | 0.03 | 0.09 | 0.19 |
| Scenario 1 | +15.5% | -43.1% | +44.3% | -26.5% | -51.1% | -52.4% |
| Scenario 2 | +15.7% | -54.2% | +15.3% | -35.7% | **-29.8%** | -55.0% |
| Scenario 3 | **+2.0%** | **-11.5%** | **-6.4%** | **-20.7%** | -41.4% | -50.0% |
| Scenario 4 | +11.7% | -18.3% | +38.1% | -26.3% | -43.7% | **-49.5%** |
| Scenario 5 | +32.2% | -21.6% | +34.0% | -42.8% | -45.1% | -50.0% |

Note:

1. The data in 1997 has been used for model warm-up to reduce the impact of the initial soil moisture conditions during the calibration period, and is not counted in the table;

2. The scenarios with bold values are labeled as the best scenario for projecting the streamflow during the verification periods, and the values from these scenarios have the least absolute percentage difference with the observed values.


**FIGURES**

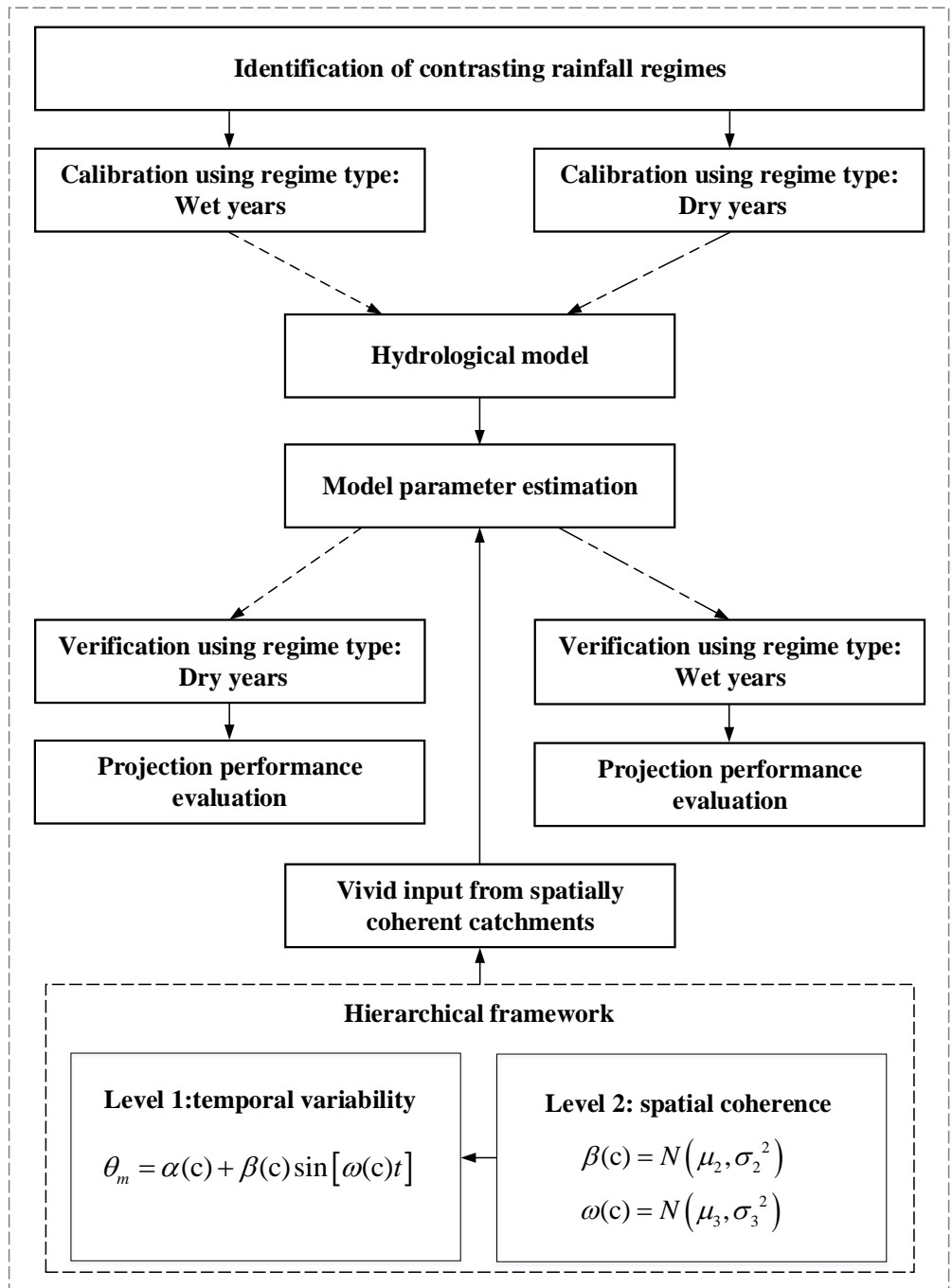


**Figure 1. Flow chart of the methodology for integrating inputs from spatially**
**coherent catchments and temporal variation of model parameters into a**
**hydrological model under contrasting climatic conditions (wet and dry years).**

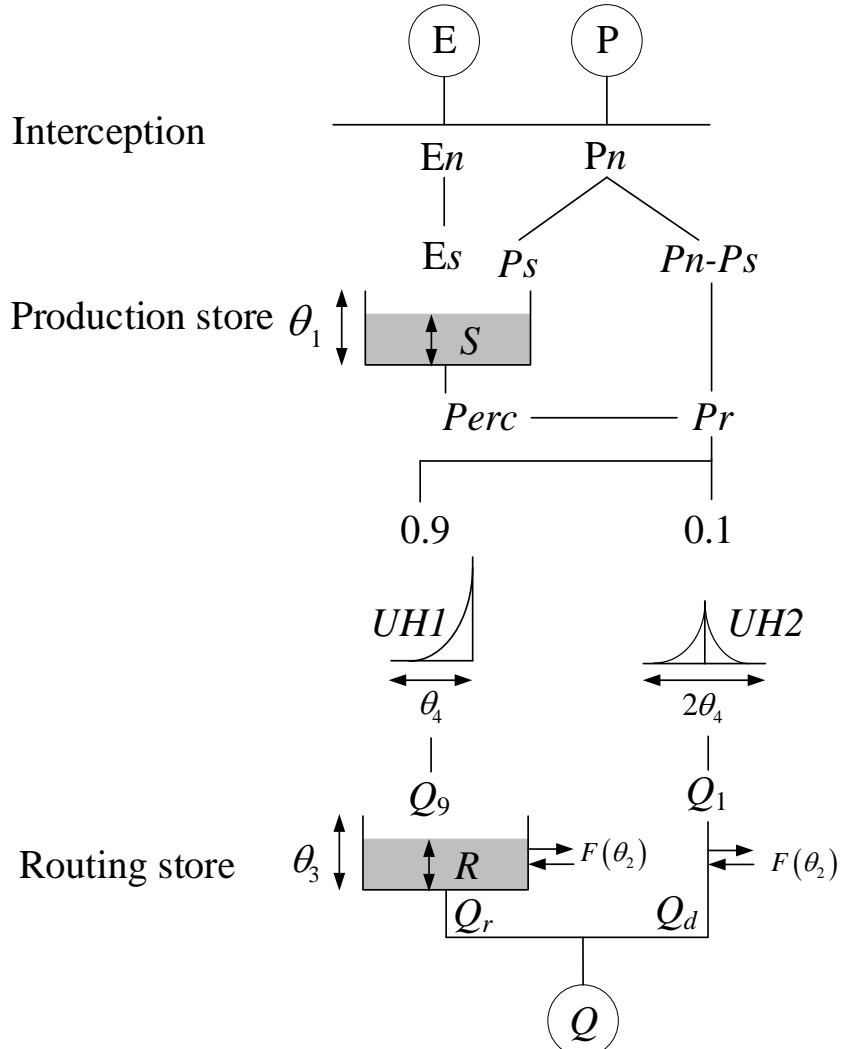


**Figure 2. Schematic diagram of the GR4J rainfall-runoff model adopted by Perrin et al. (2003). In the figure, P and E refer to precipitation and evapotranspiration, respectively; $E_n$ and $P_n$ denote net precipitation and net evapotranspiration, respectively; Ps refers to the part of precipitation that fills the production store (i.e. S). The production store is determined as a function of the water level S in the production store. The $\theta_1, \theta_2, \theta_3$, and $\theta_4$ denote model parameters. The *Perc* refers to the percolation leakage that is a function of production store *S* and parameter $\theta_1$. The Pr refers to the total quantity of water that reaches the routing functions. The UH1 and UH2 denote two-unit hydrographs. The $Q_1$ and $Q_9$ refer the corresponding output of the unit hydrographs, respectively; F indicates the groundwater exchange term; R is the level in the routing store. The $Q_r$ refers to the outflow of the routing store, $Q_d$ is a function of water exchange, and Q refers to the total streamflow.**



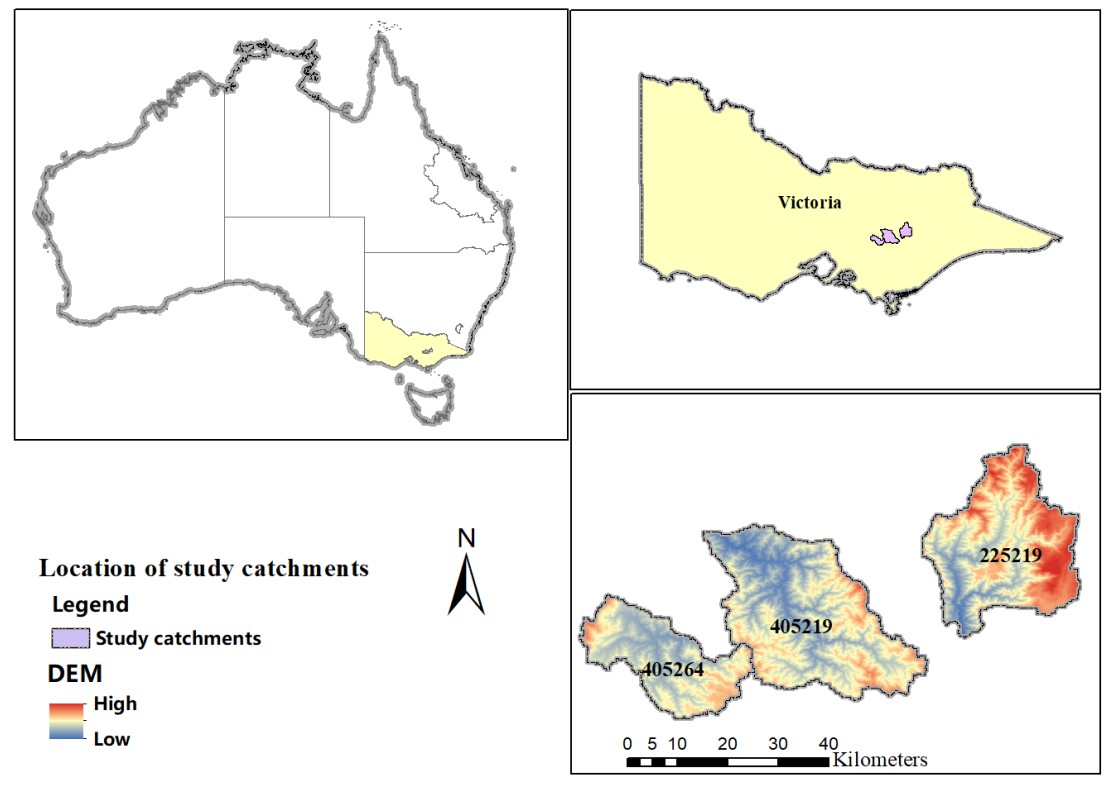


Figure 3. Locations of study catchments in Victoria, Australia. The catchment
IDs are 225219 (Macalister River catchment), 405219 (Goulburn River
catchment), and 405264 (Big River catchment).


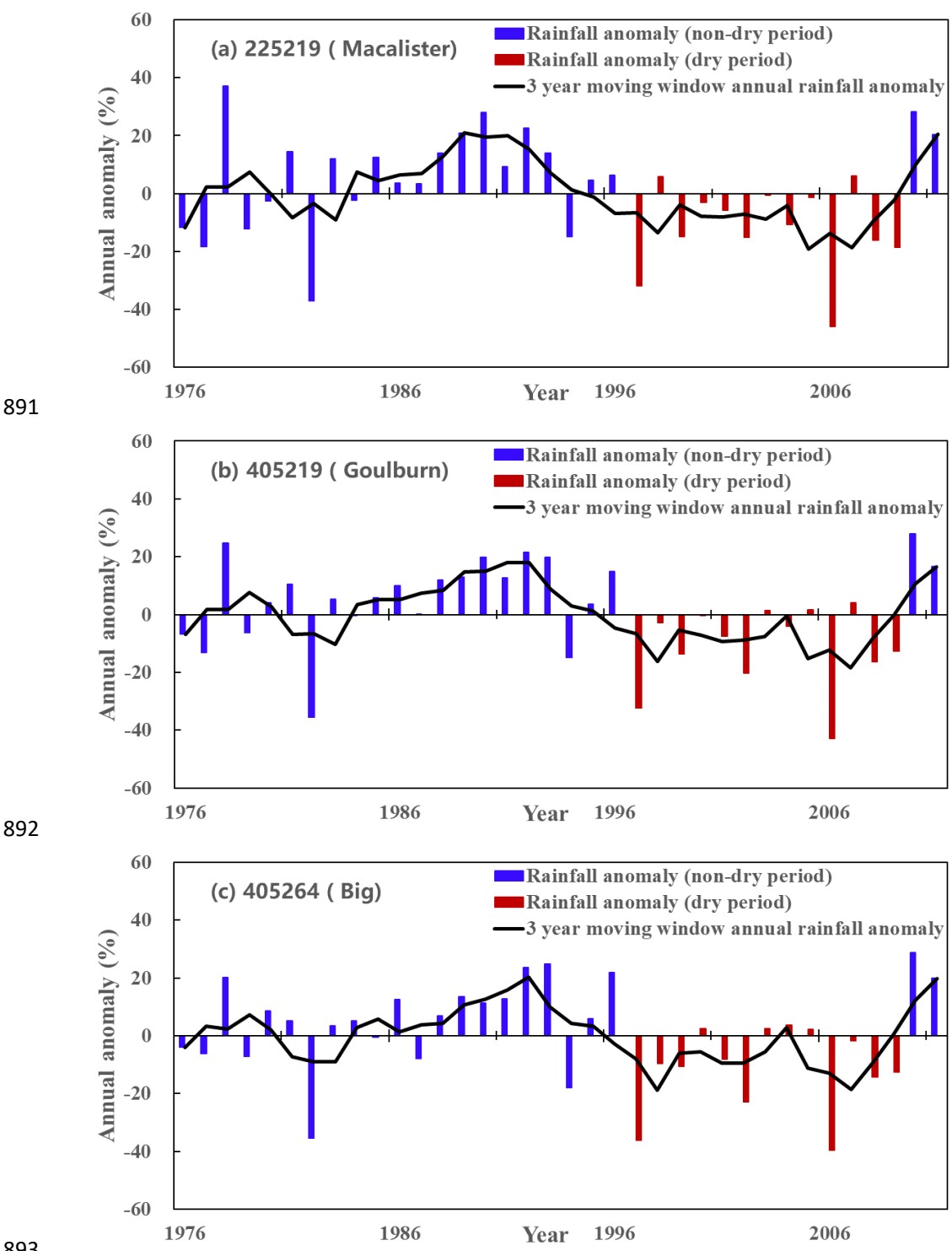




**Figure 4. The identified dry years in all catchments. The annual anomaly is defined as a percentage of the mean annual rainfall**


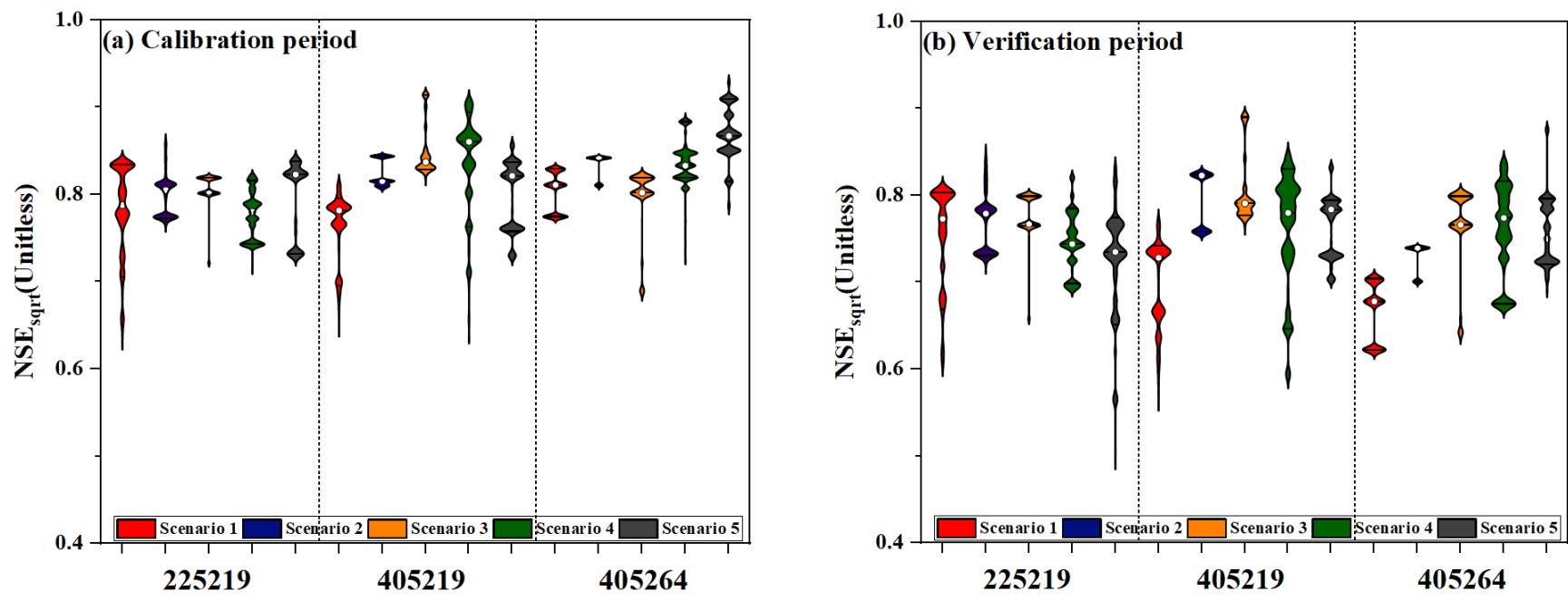

Figure 5. NSE$_{sqrt}$ for each of the five scenarios for each catchment during (a) the calibration period (wet years) and (b) the verification period (dry years). The white dots represent the median estimates of the results.

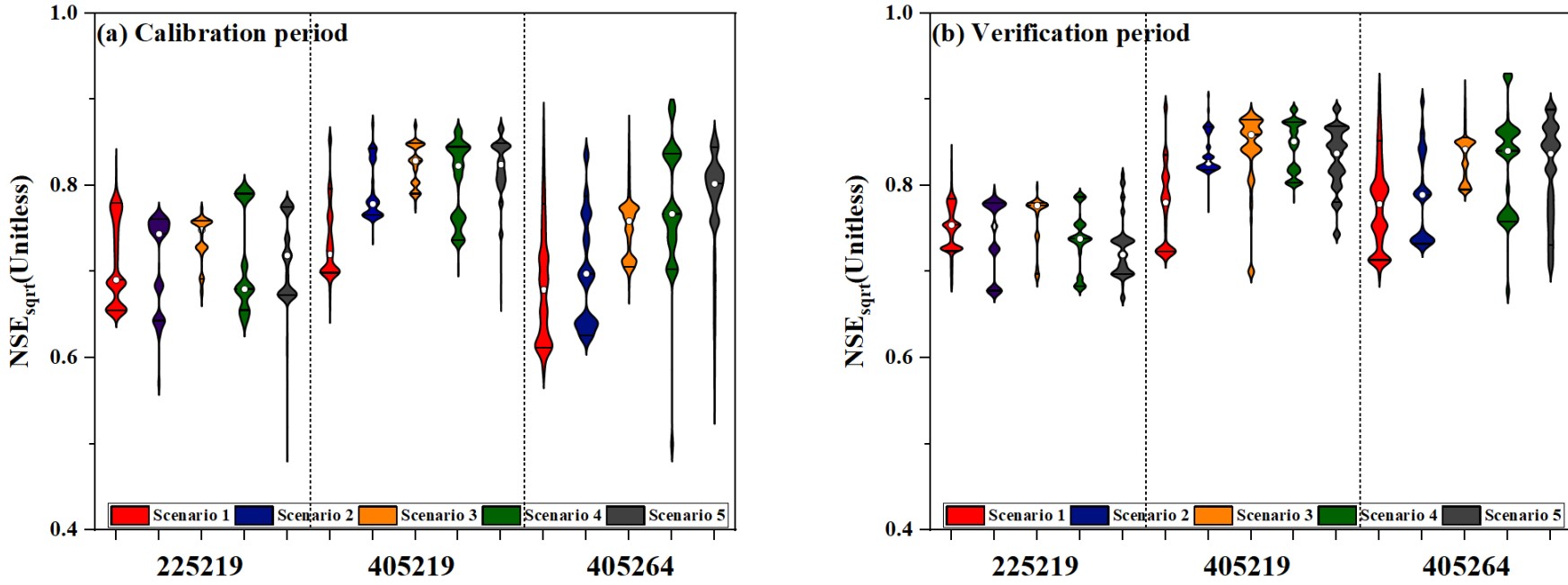

Figure 6. NSE$_{sqrt}$ for each of the five scenarios for each catchment during (a) the calibration period (dry years) and (b) the verification period (wet years). The white dots represent the median estimates of the results.

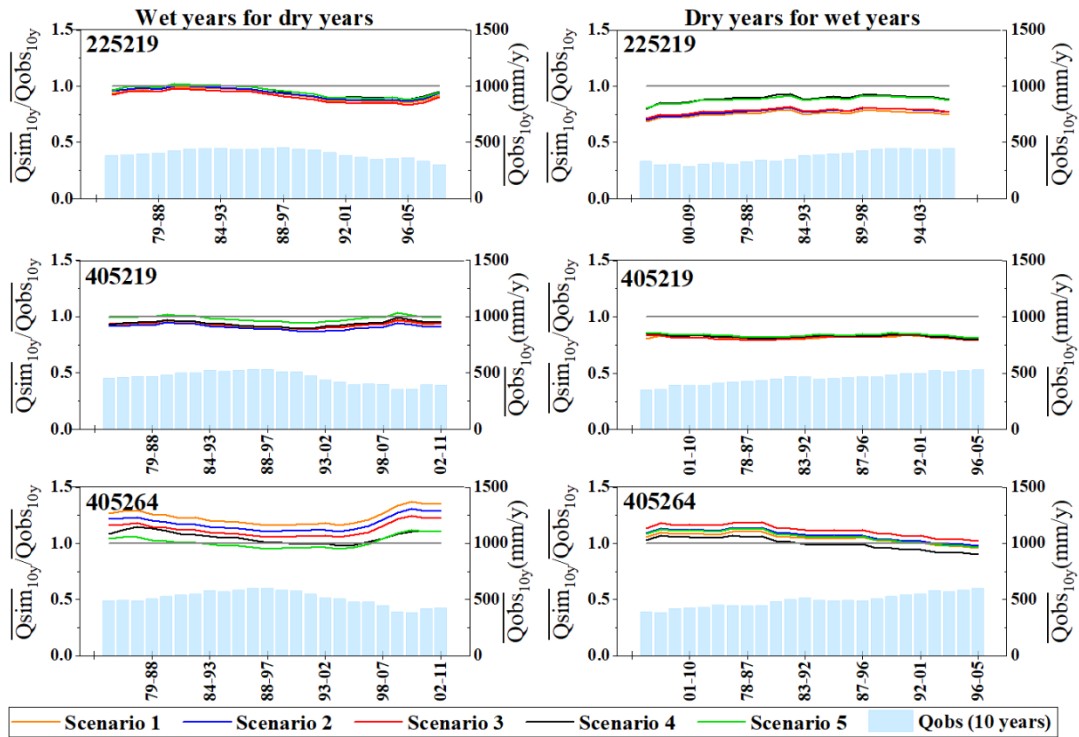

Figure 7. Long-term simulation BIAS of $Q_{median}$ for five scenarios in all catchments. Simulation BIAS is plotted as a 10-year moving average, and 10-year moving average streamflows are plotted for reference. The left-hand three graphs are calibrated in the wet years and then verified in the dry years, while the opposite sequence applies to the right-hand graphs.

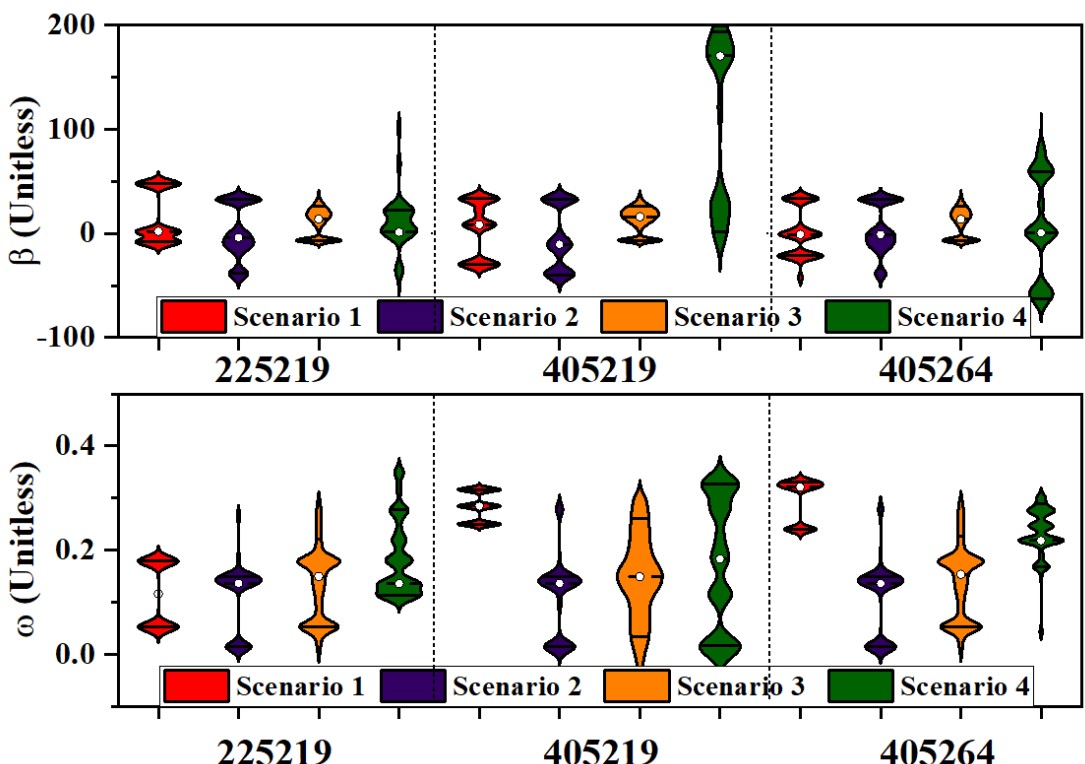


Figure 8. Posterior distributions of the regression parameters ($\beta$ and $\omega$) for the production storage capacity ($\theta_1$) for the four model scenarios in each catchment when calibrated in the wet years and verified in the dry years. The solid horizontal lines within the violin plots denote the 25[th] and 75[th] percentiles of the posterior distribution, while the white dots denote median estimates.


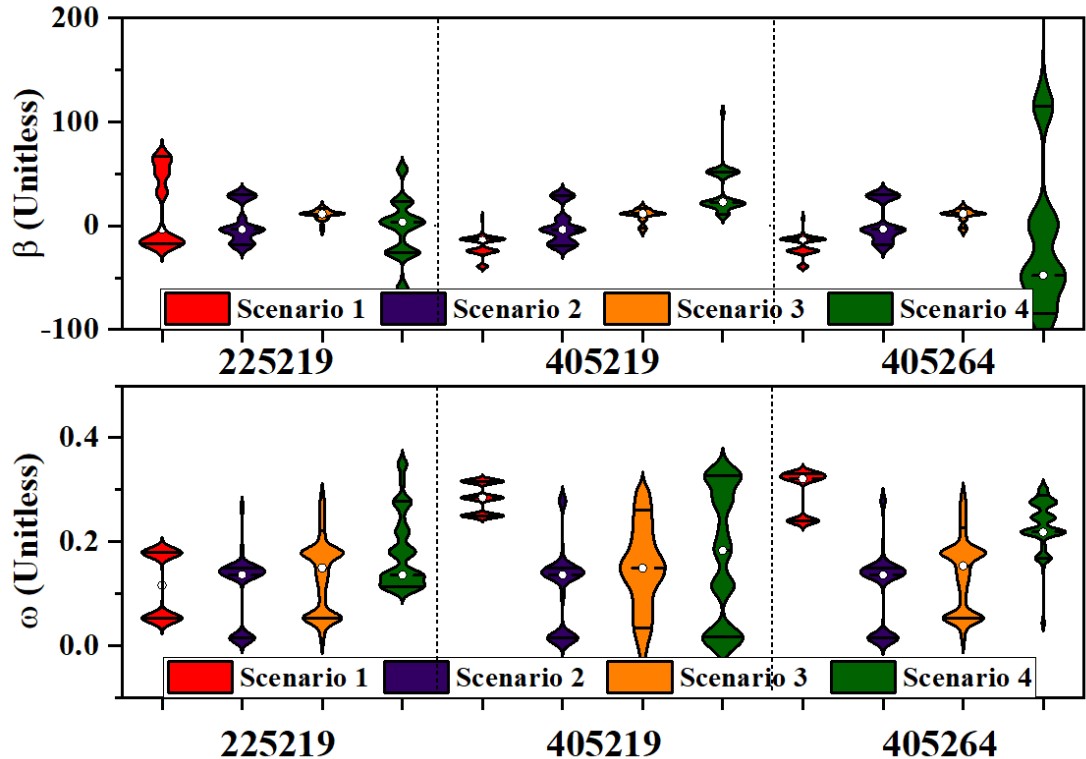


Figure 9. Posterior distributions of the regression parameters ($\beta$ and $\omega$) for the production storage capacity ($\theta_1$) for the four model scenarios in each catchment when calibrated in the dry years and verified in the wet years. The solid horizontal lines within the violin plots denote the 25th and 75th percentiles of the posterior distribution, while the white dots denote median estimates.