# Peer review of "Improving hydrological projection performance under contrasting climatic conditions using spatial coherence through a hierarchical Bayesian regression framework"

_Hydrology and Earth System Sciences, 2019_

## Referee Comment (RC1) · Anonymous Referee #1 · 22 Feb 2019

This paper analyzes the prediction performance of a lumped hydrological model using different time and spatial dependent parametrizations of one of its parameters.

There are several errors in the paper and points that should be explained better and I have a major concern regarding the results.

Comment on the results:

The value of omega looks strange to me. Assuming that the equation 1 you wrote is correct (and therefore it is a frequency and not a phase) and that the order of magnitude

of omega is of hundreds (like shown in figures 8 and 9), this mean that your parameter theta1 oscillates hundreds of times per time step. This looks unreal to me since the goal of having time-variant parameters is to represent long term (seasonal) oscillations. Therefore, either there is a problem with the unit of omega or your model is not doing what it was meant for. If omega is a phase (meaning theta1 = alpha + beta*sin(t + omega)) the value of omega makes more sense but theta1 would still complete an oscillations every 6.28 time steps (the time step is days, right?). Don't you also have a frequency that multiplies "t" and have a small value?

Detailed comments:

line 102-103: There is not a clear definition of pooling, complete pooling and hierarchical Bayesian. I would explain shortly what do they mean and which are the differences since then the paper only writes about hierarchical Bayesian.

line 152-153: It would be beneficial to explain shortly how the method works even if it was already used in other studies.

line 159: Maybe it is more appropriate to use "cross validation" instead. I suggest to avoid making a paragraph with just one sentence and remove paragraphs 2.1.1 and 2.1.2 putting all together in section 2.1.

chapter 2.3: It is not clear to me what do you do with the other parameters of the GR4J model (theta2, theta3, theta4). Do you keep them fixed or do you sample them? What is their effect on the final result?

line 199: The equation is different from the ones reported in Table 1.

line 201: You write that omega is the phase while in the equation 1 it is a frequency.

line 202: The combination alpha=beta=omega=0 makes theta 1 to be equal to 0, that indeed it is a constant value but probably it is not what you want.

chapter 2.3.2: What happens to alpha? You don't write about it anymore in the rest of

the paper. Do you keep it fixed or do you sample also it? What is its effect on the final result?

chapter 2.3.2: It is not clear to me if linking the parameters between catchments means sampling them from the same Gaussian distribution or there is another form of linking.

chapter 2.3.2: How do you sample omega and beta when they are not linked?

line 218: How do you choose the values of mu and sigma, the hyper-parameters of your model?

chapter 2.4.1: I wouldn't call "likelihood function" what actually is an objective function.

line 250: You are mixing an objective function with a prior distribution of the parameters. How do you account for the prior distribution of the parameters when they are not linked?

chapter 2.4.2: You don't say which settings of the sampling method you use (e.g. how many parameters you sample...)

chapter 3.2.1: The dataset that you get is unbalanced, since there are more wet years. Is it taken into account? Does it have an effect on the calibration?

chapter 3.2.3: Figures 7 and 8 are actually 8 and 9

Figures 5, 6, 8, 9: Since you want to show a probability distribution I wouldn't use a boxplot but, instead, I suggest to use a violin plot (e.g. https://seaborn.pydata.org/examples/grouped_violinplots.html)

Figures 8, 9: Why do you change the colors between beta and omega? This makes the plot more difficult to read.

---

## Referee Comment (RC2) · Anonymous Referee #2 · 22 Feb 2019

General Comments

The study of Pan et al. tests a Hierarchical Bayesian framework to incorporate time and spatial variability in model parameters. Specifically, the method was tested for the GR4J-model in three Australian catchments. Four modelling scenarios were tested, and one base scenario was formulated. The study shows that including spatially and temporally variable parameters improves model performance and reduces uncertainty.

The article shows interesting work, which could be a nice contribution to the field.

[Figure]

Generally, the article needs some more explanations on the method, but there is also some incomplete reasoning. Hence, there are several issues I'd like to address.

Specific comments

Key of the article is the hierarchical framework, but the authors may want to work on the explanation of the method. It is especially not clear to me how the hyper-parameters are determined, and how the catchment-specific values follow from that. Are the hyper-parameters estimated in SCEM-UA? Or are these pre-defined? The gaussian distributions are defined by the authors as prior distributions, and that makes me assume that the model parameter theta is determined in SCEM-UA starting from this prior distribution, whereas the remaining model parameters are either kept fixed or sampled from a uniform distribution and independently for each catchment. Is that correct? Because if that is the case, the hyper-parameters (and hence the distribution) are determined in advance, so what are these based on? Besides, the choice of a gaussian distribution may seem a logical first guess, but it remains an arbitrary choice. So what is the reasoning behind this choice? In addition, the choice of the prior distribution may lead to some circular reasoning. When spatial coherence is used, the variation in performance goes down, but is this not just an artefact of the pre-defined gaussian distribution? In other words, if the prior distribution is set narrower, the resulting posterior distribution will probably be narrower as well. I believe it is therefore crucial to report also the prior ranges (or fixed values) for especially the (time-invariant) theta-parameter, but also all other model parameters.

I also wonder how valid it is to assume the catchments are similar. The authors state on p15.L314, that the catchments satisfy the homogeneity assumption. What is this assumption and how do they satisfy this assumption? A clear description of the catchments may be needed to defend that the catchments are the same. Just looking at the DEM and the annual values of rainfall and runoff (Table 2) give me the idea that the Big catchment (405264) behaves fundamentally different compared to the other two. This catchment also reached much higher performances in calibration (Fig.5 and 6) when

no spatial coherence is used, and also shows different results in the BIAS comparison (Figure 7).

Sometimes, the conclusions and statements of the authors do not seem to be strongly supported by the data as shown. The boxplots with performances (Figures 5, 6) show relatively similar performances, and, to be honest, a clear pattern is not very obvious. In addition, the authors tend to generalize in some cases findings that mainly apply to just two of the three catchments (see also my minor comments). I believe additional analyses may be needed to support the conclusions more, for example a statistical test to check if the distributions are significantly different. Or the addition of other, multiple performance measures, to assess the performance over multiple aspects (high flows, low flows etc.). Further, all beta-values plot around zero in Figure 8, basically pointing at the absence of a clear trend. Is this indeed true? It would be interesting to show the timeseries of the parameter. The absence of a trend may explain the similar performances for all scenarios, and especially also why the time-varying scenarios do not outperform the others clearly. Besides, when beta is around zero, there is no point of looking at omega, as this does not do much in that case.

Concluding, the authors may need to clarify more what they did and how they arrive at several conclusions. I hope the authors find my comments useful, and I look forward to a revised manuscript.

Technical corrections

P.7. section 2.1.1. Please elaborate on how the dry periods are defined.

P8. Section 2.1.2. Why add this paragraph when you only refer to section 2.5?

P10. L210 Do you mean Eq. 1?

P10.L210 ...expected to the same... → expected to be the same

P12. L50. Please define N and n
P12.L258. Which parameters are optimized in SCEM-UA?

P15.L326. Please explain how I can see this from Figure 4, except for the pre-defined red colour. Is this where the black line crosses the axis? Why are the first years not considered?

P16.L339-340. Are these references in the right place? You describe your own results, shouldn't you refer to one of the figures?

P17.L355-357. This is, as far as I can see, not true for all catchments. Catchments 225219 and 405264 have a higher median, but the variation is less for 225219.

P17.L362. As far as I can see, it has only the highest median value for catchment 225219.

P18.L375. The performances in the verification period seem higher to me? What do you mean calibrated performances were inferior?

P.18L375-377. This is not true for catchment 225219

P18.L379. The ranges seem not very different between scenarios 4 and 5, only slightly.

P18.L379-380. It's not very obvious that scenario 3 has a higher median performance for catchment 405264

P18.L382 This is not very obvious to me

P18.L394. Compared –> comparing

P20.L438. Is omega for scenario 4 not the lowest in all cases? Or do you mean the absolute values?

Figure 2. Please define all symbols and abbreviations in the figure.

Figure 5,6: I would suggest to plot the boxes for calibration and verification next to each other. It's easier to see whether there is an improvement or not. Please also add the units (also when a unitless number is presented)

Figure 7. Please make the labels and text bigger.

Figure 8, 9. Maybe use the same colors for the scenarios in both plots. What are the units of beta and omega?

---

## Author Comment (AC1) · 1 Apr 2019

This paper analyzes the prediction performance of a lumped hydrological model using different time and spatial dependent parametrizations of one of its parameters. There are several errors in the paper and points that should be explained better and I have a major concern regarding the results. Comment on the results: A1: The value of omega looks strange to me. Assuming that the equation 1 you wrote is correct (and therefore it is a frequency and not a phase) and that the order of magnitude of omega is of hundreds (like shown in figures 8 and 9), this mean that your parameter theta1 oscillates hundreds of times per time step. This looks unreal to me since the goal of having time-variant parameters is to represent long term (seasonal) oscillations. Therefore, either there is a problem with the unit of omega or your model is not doing what it was meant for. If omega is a phase (meaning theta1 = alpha + beta*sin(t + omega)) the value of omega makes more sense but theta1 would still complete an oscillations every 6.28 time steps (the time step is days, right?). Don't you also have a frequency that multiplies "t" and have a small value? Reply: We apologize for our mistakes. Omega represents frequency rather than phase. It will be revised accordingly in the revised manuscript. We have carefully checked the results of regression parameter Omega and found that the Figures 8 and 9 in the manuscript of Omega should be modified as the attachments: See the attachment Figure 8. Posterior distributions of the regression parameters ($\beta$ and $\omega$) for the production storage capacity ($\theta1$) for the four modeling scenarios in all the 3 studied catchments. In this figure, parameters were calibrated in the non-dry period while verified in the dry period. The solid horizontal lines within the violin plots denote the 25th and 75th percentiles of the posterior distribution, while the dash line denotes median estimates. See the attachment Figure 9. Posterior distributions of the regression parameters ($\beta$ and $\omega$) for the production storage capacity ($\theta1$) for the four model scenarios in all 3 studied catchments. In this figure, parameters were calibrated in the dry period while verified in the non-dry period. The solid horizontal lines within the violin plots denote the 25th and 75th percentiles of the posterior distribution, while the dash line denotes median estimates. For the first four scenarios as shown in Figure 8, the average median estimates of regression parameter $\omega$ of the 3 catchments are 0.24, 0.14, 0.15, and 0.18, respectively., and that in Figure 9 are 0.15, 0.26, 0.23, and 0.17 respectively in Figure 9. Thus, the phase of the sine term could be derived based on the regression parameter $\omega$. The mean phase of model parameter Seta1 for each scenario is 26.2, 46.3, 41.9 and 35.2 in Figure 8, respectively. It is 42.9, 24.1, 27.4 and 38.0 in Figure 9, respectively.

Detailed comments: A2: line 102-103: There is not a clear definition of pooling, complete pooling and hierarchical Bayesian. I would explain shortly what do they mean and

which are the differences since then the paper only writes about hierarchical Bayesian. Reply: Thank you for your comments. The following explanations (in blue) about the pooling, complete pooling and hierarchical Bayesian will be added in the revised manuscript. In general, there are three methods to consider the spatial coherence between different catchments in parameter estimation. The first one is no pooling, which means every catchment is modeled independently, and all parameters are catchment-specific. The second one is complete pooling, which means parameters are considered to be common across all catchments. The third/last one is hierarchical Bayesian (HB) framework, also known as partial pooling, which means some parameters are allowed to vary by catchments and some parameters are assumed to be drown from a common hyper-distribution across the region that consists of different catchments.

A3: line 152-153: It would be beneficial to explain shortly how the method works even if it was already used in other studies. Reply: Thank you for your comment. Definition of dry period is explained in the following paragraph and will be added in the revised manuscript: Saft et al. (2015) tested several algorithms for dry period delineation, which considered different combinations of dry run length, dry run anomaly and various boundary criteria, and found that the identification results of dry period by one of the algorithms showed marginal dependence on the algorithm and the main results were robust to different algorithms. The detailed processes could be found on Saft et al. (2015) and also are as follows. Firstly, the annual rainfall data were calculated relative to the annual mean, and the anomaly series was divided by the mean annual rainfall and smoothed with a 3 year moving window. Secondly, the first year of the drought remained the start of the first 3 year negative anomaly period. Thirdly, the exact end date of the dry period was determined through analysis of the unsmoothed anomaly data from the last negative 3 year anomaly. The end year was identified as the last year of this 3 year period unless: (i) there was a year with a positive anomaly >15

A4: line 159: Maybe it is more appropriate to use "cross validation" instead. I suggest to avoid making a paragraph with just one sentence and remove paragraphs 2.1.1 and

2.1.2 putting all together in section 2.1. Reply: Thanks. (1) Follow the Referee's comment, the phrase "Verification method will be modified as "Cross validation". (2) Follow the Referee's suggestion, paragraph 2.1.1 and 2.1.2 will be put together in section 2.1, and the sub-titles of section 2.1.1 and 2.1.2 will be deleted in the revised manuscript.

A5: chapter 2.3: It is not clear to me what do you do with the other parameters of the GR4J model (theta2, theta3, theta4). Do you keep them fixed or do you sample them? What is their effect on the final result? Reply: Thank you for your comment. (1) All other model parameters $(\text{theta}_2, theta_3 and theta_4, except theta_1) are not fixed, but sampled simultaneously with regression parameter alpha, beta and ga$ $parameters mu_2, sigma_2, mu_3 and sigma_3 in the SCEM-UA algorithm. In actual calculation process, we would set a large varia$ $Rubin convergence value of 1.2 (Gelman et al., 2013) would be selected as the posterior probability distribution of parameters. Mor$ $varying while other model parameters are temporal invariant.$

A6: line 199: The equation is different from the ones reported in Table 1. Reply: We apologize for our mistakes. The fault equations in Table 1 have been revised as equation 1 in the revised manuscript.

A7: line 201: You write that omega is the phase while in the equation 1 it is a frequency. Reply: Thank you for pointing out this mistake. The Omega represents the frequency rather than the phase (see response to comment A1). The statement in line 201 is wrong and will be modified in the revised manuscript.

A8: line 202: The combination alpha=beta=omega=0 makes theta 1 to be equal to 0, that indeed it is a constant value but probably it is not what you want. Reply: Thanks. According to the definition of the GR4J model (Perrin et al., 2003), $\text{Theta}_1 represents the primary storage of water in the catchment and must be a positive value. Thus, in the first four scenarios, in$ $0), the combination of Alpha = beta = omega = $ $0 would be excluded first, and other combinations that made theta_1 equal to zero would be excluded too.$

A9: chapter 2.3.2: What happens to alpha? You don't write about it anymore in the rest of the paper. Do you keep it fixed or do you sample also it? What

is its effect on the final result? Reply: Thanks. (1) The alpha represents the constant term in equation 1. Changes in alpha lead to consistent changes in $theta_1$ $across the whole time series, which doesn't result in temporal variations of model parameter theta_1. In addition, one object$ $parameters mu_2, sigma_2, mu_3 and sigma_3, other regression parameters beta and omega (if present), and model parameters thet$ $UA algorithm.$

A10: chapter 2.3.2: It is not clear to me if linking the parameters between catchments means sampling them from the same Gaussian distribution or there is another form of linking. Reply: We apologize for the misunderstanding. The link is that regression parameter beta(omega) of different catchments is assumed to sample their values in the same Gaussian distribution. This kind of links have been widely used in the field of extreme event analysis, such as Sun et al (2015, 2016), Lima et al (2009) and Bracken et al (2018).

A11: chapter 2.3.2: How do you sample omega and beta when they are not linked? Reply: Thanks. The omega is not linked in scenario 1, while beta is not linked in scenario 2. In scenario 4, both omega and beta are not linked. Spatially irrelevant parameters would be sampled and derived as independent variables. For example, in scenario 4, the omega and beta of different catchments are not linked, thus values of mega and beta of each catchment are calibrated from corresponding catchment inputs. In scenario 1, regression parameter $\beta(c)=N(\mu_3, \sigma^2), which means that beta is shared with linked catchments, while independent regression parameters \omega 1$- 1, $\omega$1-2, and $\omega$1-3 are used to represent the frequency of model parameter $theta_1$ $in different catchments. The name of all unknown quantities in different scenarios could be found in the supplementary m$

A12: line 218: How do you choose the values of mu and sigma, the hyper-parameters of your model? Reply: Thanks. The posterior distributions of all unknown quantities, including model parameters $theta_2, theta_3 and theta_4, and regression parameter alpha, beta and gamma, and hyper-$ $parameters mu_2, sigma_2, mu_3 and sigma_3 are derived simultaneously through the SCEM-$ $UA algorithm. In actual calculation process, we would set a large variation interval for each unknown quantity first, parameter$

$Rubin convergence value of 1.2 (Gelman et al., 2013) would be selected as the posterior probability distribution of parameters.$

A13: chapter 2.4.1: I wouldn't call "likelihood function" what actually is an objective function. Reply: Thanks. As suggested, the "likelihood function" will be modified as "objective function" in the revised manuscript.

A14: line 250: You are mixing an objective function with a prior distribution of the parameters. How do you account for the prior distribution of the parameters when they are not linked? Reply: Thanks. The objective function of Eq.1 will be modified as follows: Please see the supplementary material (Line 31). where $theta_1, theta_2, theta_3 and theta_4 refer to four model parameters. The objective function of Eq.5 will be modified as follows : Please see the supplementary material (Line 32). where the number of catchments in the region is represented by C; c represents th$

A15: chapter 2.4.2: You don't say which settings of the sampling method you use (e.g. how many parameters you sample. . .) Reply: Thanks. The sampling method used in this paper is the SCEM-UA algorithm. The detailed description of the settings of SCEM-UA algorithm will be added in the revised manuscript: Convergence is assessed by evolving three parallel chains with 30000 random samples, while verifying that the posterior distribution of parameters results in a value smaller than a Gelman-Rubin convergence value of 1.2 (Gelman et al., 2013). The number of unknown quantities in different scenarios are as follows: 15 in scenario 1 and scenario 2, 13 in scenario 3 and 18 in scenario 4.

A16: chapter 3.2.1: The dataset that you get is unbalanced, since there are more wet years. Is it taken into account? Does it have an effect on the calibration? Reply: Thank you for pointing out this situation. (1) Generally, calibration data should be longer than 3-6 years for daily hydrological modeling in order to get robust results (Perrin et al., 2003, Coron et al., 2012). Thus, data from both dry period (15 years) and wet period (21 years) were used for model calibration to meet this requirement. (2) Generally, a longer time series may improve the robustness of hydrological predictions. However, we tested the calibration performance with different lengths of records (> 10 years) in

dry and non-dry periods and found that their results are almost the same. Therefore, we used both the length of 15 years of dry and 10 years non-dry periods into calibration in order to utilize all available data.

A17: chapter 3.2.3: Figures 7 and 8 are actually 8 and 9. Reply: Thanks. Changes will be made as suggested.

A18: Figures 5, 6, 8, 9: Since you want to show a probability distribution I wouldn't use a boxplot but, instead, I suggest to use a violin plot (e.g.https://seaborn.pydata.org/examples/grouped$_{v}$iolinplots.html)$Reply$ : $Thankyouforyoursuggestions.Figures 8 and 9 will be modified as violin plot in the revised manuscript, which also could be found$ $Please see the attachment Figure 5(a) Please see the attachment Figure 5(b) Figure 5. NSE sqrt for each of the five scenarios for$ $dry period) and (b) the verification period (dry period). Please see the attachment Figure 6(a) Please see the attachment Figure 6($ $dry period).$

A19: Figures 8, 9: Why do you change the colors between beta and omega? This makes the plot more difficult to read. Reply: Thanks. The same color will be used to the same parameter consistently in all figures. Changes will be made as suggested in the revised figures. Please refer to response to comment A1 by Referee 1.

Please also note the supplement to this comment:
https://www.hydrol-earth-syst-sci-discuss.net/hess-2019-6/hess-2019-6-AC1-supplement.pdf

———————————————

[Figure]

**Fig. 1.** Figure 8 Posterior distributions of the regression parameters ($\beta$ and $\omega$) for the production storage capacity ($\theta1$) for the four modeling scenarios in all the 3 studied catchments. In this figure, param

**Fig. 2.** Figure 9 Posterior distributions of the regression parameters ($\beta$ and $\omega$) for the production storage capacity ($\theta 1$) for the four model scenarios in all 3 studied catchments. In this figure, parameters we

(a) Calibration period

NSE$_{sqrt}$(Unitless)

1.0

0.8

0.6

0.4

Scenario 1 | Scenario 2 | Scenario 3 | Scenario 4 | Scenario 5

225219          405219          405264

**Fig. 3.** Figure 5(a) NSEsqrt for each of the five scenarios for each catchment during (a) the calibration period (non-dry period) and (b) the verification period (dry period).

HESSD

Interactive
comment

Legend: Scenario 1 (red), Scenario 2 (blue), Scenario 3 (orange), Scenario 4 (green), Scenario 5 (grey).

(b) Varification period

**Fig. 4.** Figure 5(b) NSEsqrt for each of the five scenarios for each catchment during (a) the calibration period (non-dry period) and (b) the verification period (dry period).
(a) Calibration period

NSE$_{sqrt}$(Unitless)

Scenario 1 | Scenario 2 | Scenario 3 | Scenario 4 | Scenario 5

225219   405219   405264

**Fig. 5.** Figure 6(a) NSEsqrt for each of the five scenarios for each catchment during (a) the calibration period (dry period) and (b) the verification period (non-dry period).

**(b) Varification period**

NSE$_{sqrt}$(Unitless)

1.0

0.8

0.6

0.4

| Scenario 1 | Scenario 2 | Scenario 3 | Scenario 4 | Scenario 5 |

225219          405219          405264

**Fig. 6.** Figure 6(b) NSEsqrt for each of the five scenarios for each catchment during (a) the calibration period (dry period) and (b) the verification period (non-dry period).

**Supplement:**

This paper analyzes the prediction performance of a lumped hydrological model using different time and spatial dependent parametrizations of one of its parameters. There are several errors in the paper and points that should be explained better and I have a major concern regarding the results.

Comment on the results:

A1: The value of omega looks strange to me. Assuming that the equation 1 you wrote is correct (and therefore it is a frequency and not a phase) and that the order of magnitude of omega is of hundreds (like shown in figures 8 and 9), this mean that your parameter theta1 oscillates hundreds of times per time step. This looks unreal to me since the goal of having time-variant parameters is to represent long term (seasonal) oscillations. Therefore, either there is a problem with the unit of omega or your model is not doing what it was meant for. If omega is a phase (meaning theta1 = alpha + beta*sin(t + omega)) the value of omega makes more sense but theta1 would still complete an oscillations every 6.28 time steps (the time step is days, right?). Don't you also have a frequency that multiplies "t" and have a small value?

**Reply:**

(1) We apologize for our mistakes. $\omega$ represents frequency rather than phase. It will be revised accordingly in the revised manuscript.

(2) We have carefully checked the results of regression parameter $\omega$ and found that the Figures 8 and 9 in the manuscript of $\omega$ should be modified as the attachmments:

[Figure]

Figure 8. Posterior distributions of the regression parameters ($\beta$ and $\omega$) for the production storage capacity ($\theta_1$) for the four modeling scenarios in all the 3 studied catchments. In this figure, parameters were calibrated in the non-dry period while verified in the dry period. The solid horizontal lines within the violin plots denote the 25th and 75th percentiles of the posterior distribution, while the dash line denotes median estimates.

[Figure]

Figure 9. Posterior distributions of the regression parameters ($\beta$ and $\omega$) for the production storage capacity ($\theta_1$) for the four model scenarios in all 3 studied catchments. In this figure, parameters were calibrated in the dry period while verified in the non-dry period. The solid horizontal lines within the violin plots denote the 25th and 75th

percentiles of the posterior distribution, while the dash line denotes median estimates.

For the first four scenarios as shown in Figure 8, the average median estimates of regression parameter ω of the 3 catchments are 0.24, 0.14, 0.15, and 0.18, respectively., and that in Figure 9 are 0.15, 0.26, 0.23, and 0.17 respectively in Figure 9. Thus, the phase of the sine term could be derived based on the regression parameter ω. The mean phase of model parameter $\theta_1$ for each scenario is 26.2, 46.3, 41.9 and 35.2 in Figure 8, respectively. It is 42.9, 24.1, 27.4 and 38.0 in Figure 9, respectively.

**Detailed comments:**
A2: line 102-103: There is not a clear definition of pooling, complete pooling and hierarchical Bayesian. I would explain shortly what do they mean and which are the differences since then the paper only writes about hierarchical Bayesian.

**Reply:** Thank you for your comments. The following explanations (in blue) about the pooling, complete pooling and hierarchical Bayesian will be added in the revised manuscript.

In general, there are three methods to consider the spatial coherence between different catchments in parameter estimation. The first one is no pooling, which means every catchment is modeled independently, and all parameters are catchment-specific. The second one is complete pooling, which means parameters are considered to be common across all catchments. The third/last one is hierarchical Bayesian (HB) framework, also known as partial pooling, which means some parameters are allowed to vary by catchments and some parameters are assumed to be drown from a common hyper-distribution across the region that consists of different catchments.

A3: line 152-153: It would be beneficial to explain shortly how the method works even if it was already used in other studies.

**Reply:** Thank you for your comment. Definition of dry period is explained in the

following paragraph and will be added in the revised manuscript:

Saft et al. (2015) tested several algorithms for dry period delineation, which considered different combinations of dry run length, dry run anomaly and various boundary criteria, and found that the identification results of dry period by one of the algorithms showed marginal dependence on the algorithm and the main results were robust to different algorithms. The detailed processes could be found on Saft et al. (2015) and also are as follows.

Firstly, the annual rainfall data were calculated relative to the annual mean, and the anomaly series was divided by the mean annual rainfall and smoothed with a 3 year moving window. Secondly, the first year of the drought remained the start of the first 3 year negative anomaly period. Thirdly, the exact end date of the dry period was determined through analysis of the unsmoothed anomaly data from the last negative 3 year anomaly. The end year was identified as the last year of this 3 year period unless: (i) there was a year with a positive anomaly >15% of the mean, in which case the end year is set to the year prior to that year; or (ii) if the last two years have slightly positive anomalies (but each <15% of the mean), in which case the end year is set to the first year of positive anomaly; (iii) To ensure that the dry periods are sufficiently long and severe, in the subsequent analysis, the author use dry periods with the following characteristics: length$\geq$ 7 years; mean dry period anomaly<25%.

A4: line 159: Maybe it is more appropriate to use "cross validation" instead. I suggest to avoid making a paragraph with just one sentence and remove paragraphs 2.1.1 and 2.1.2 putting all together in section 2.1.

**Reply:** Thanks.

(1) Follow the Referee's comment, the phrase "Verification method will be modified as "Cross validation".

(2) Follow the Referee's suggestion, paragraph 2.1.1 and 2.1.2 will be put together in section 2.1, and the sub-titles of section 2.1.1 and 2.1.2 will be deleted in the revised manuscript.

A5: chapter 2.3: It is not clear to me what do you do with the other parameters of the GR4J model (theta2, theta3, theta4). Do you keep them fixed or do you sample them? What is their effect on the final result?

**Reply:** Thank you for your comment.

(1) All other model parameters ($\theta_2$, $\theta_3$, and $\theta_4$, except $\theta_1$) are not fixed, but sampled simultaneously with regression parameter $\alpha$, $\beta$ and $\omega$ (if present), and hyper-parameters $\mu_2$, $\sigma_2$, $\mu_3$ and $\sigma_3$ in the SCEM-UA algorithm. In actual calculation process, we would set a large variation interval for each unknown quantity first, parameters would converge to a small interval in MCMC calculation process, the final parameter samples that satisfy the requirement that a GR value must be smaller than a Gelman-Rubin convergence value of 1.2 (Gelman et al., 2013) would be selected as the posterior probability distribution of parameters. More information will be added in the revised manuscript.

(2) Previous studies on GR4J model showed that $\theta_2$, $\theta_3$, and $\theta_4$ are less sensitive than $\theta_1$ under changing climate (Perrin et al., 2003;Renard et al., 2011;Westra et al., 2014). Therefore, we think that it is reasonable to assume that $\theta_1$ is time-varying while other model parameters are temporal invariant.

A6: line 199: The equation is different from the ones reported in Table 1.

**Reply:** We apologize for our mistakes. The fault equations in Table 1 have been revised as equation 1 in the revised manuscript.

A7: line 201: You write that omega is the phase while in the equation 1 it is a frequency.

**Reply:** Thank you for pointing out this mistake. The $\omega$ represents the frequency rather than the phase (see response to comment A1). The statement in line 201 is wrong and will be modified in the revised manuscript.

A8: line 202: The combination alpha=beta=omega=0 makes theta 1 to be equal to 0, that indeed it is a constant value but probably it is not what you want.

**Reply:** Thanks. According to the definition of the GR4J model (Perrin et al., 2003), $\theta_1$ represents the primary storage of water in the catchment and must be a positive value. Thus, in the first four scenarios, in order to avoid this situation ($\theta_1=0$), the combination of $\alpha=\beta=\omega=0$ would be excluded first, and other combinations that made $\theta_1$ equal to zero would be excluded too.

A9: chapter 2.3.2: What happens to alpha? You don't write about it anymore in the rest of the paper. Do you keep it fixed or do you sample also it? What is its effect on the final result?

**Reply:** Thanks.

(1) The $\alpha$ represents the constant term in equation 1. Changes in $\alpha$ lead to consistent changes in $\theta_1$ across the whole time series, which doesn't result in temporal variations of model parameter $\theta_1$. In addition, one objective of this study is to explore the potential temporal variation of $\theta_1$; thus, the regression parameter $\alpha$ is not our focus.

(2) Regression parameter $\alpha$ is not fixed in advance but is sampled as same as

other unknown quantities. The posterior distribution of $\alpha$ is derived out simultaneously with hyper-parameters $\mu_2$, $\mu_3$, $\sigma_2$ and $\sigma_3$, other regression parameters $\beta$ and $\omega$ (if present), and model parameters $\theta_2$, $\theta_3$ and $\theta_4$ in the SCEM-UA algorithm.

A10: chapter 2.3.2: It is not clear to me if linking the parameters between catchments means sampling them from the same Gaussian distribution or there is another form of linking.

**Reply:** We apologize for the misunderstanding. The link is that regression parameter $\beta(\omega)$ of different catchments is assumed to sample their values in the same Gaussian distribution. This kind of links have been widely used in the field of extreme event analysis, such as Sun et al (2015, 2016), Lima et al (2009) and Bracken et al (2018).

A11: chapter 2.3.2: How do you sample omega and beta when they are not linked?

**Reply:** Thanks. The $\omega$ is not linked in scenario 1, while $\beta$ is not linked in scenario 2. In scenario 4, both $\omega$ and $\beta$ are not linked. Spatially irrelevant parameters would be sampled and derived as independent variables. For example, in scenario 4, the $\omega$ and $\beta$ of different catchments are not linked, thus values of $\omega$ and $\beta$ of each catchment are calibrated from corresponding catchment inputs. In scenario 1, regression parameter $\beta(c) = N(\mu_3, \sigma^2)$, which means that $\beta$ is shared with linked catchments, while independent regression parameters $\omega_{1-1}$, $\omega_{1-2}$, and $\omega_{1-3}$ are used to represent the frequency of model parameter $\theta_1$ in different catchments. The name of all unknown quantities in different scenarios could be found in the supplementary material (at the end of this reply) , and these tables will be added in the revised manuscript.

The prior ranges of all unknown quantities in different scenarios have been added in the supplementary material.

A12: line 218: How do you choose the values of mu and sigma, the hyper-parameters of your model?

**Reply:** Thanks. The posterior distributions of all unknown quantities, including model parameters $\theta_2$, $\theta_3$ and $\theta_4$, and regression parameters $\alpha$, $\beta$ and $\omega$, and hyper-parameters $\mu_2, \mu_3$, $\sigma_2$ and $\sigma_3$ are derived simultaneously through the SCEM-UA algorithm. In actual calculation process, we would set a large variation interval for each unknown quantity first, parameters would converge to a small interval in MCMC calculation process, the final parameter samples that satisfy the requirement that a GR value must be smaller than a Gelman-Rubin convergence value of 1.2 (Gelman et al., 2013) would be selected as the posterior probability distribution of parameters.

A13: chapter 2.4.1: I wouldn't call "likelihood function" what actually is an objective function.

**Reply:** Thanks. As suggested, the "likelihood function" will be modified as "objective function" in the revised manuscript.

A14: line 250: You are mixing an objective function with a prior distribution of the parameters. How do you account for the prior distribution of the parameters when they are not linked?

**Reply:** Thanks.

(1) The objective function of Eq.1 will be modified as follows:

$$\varepsilon_c\left[\theta_1,\theta_2,\theta_3,\theta_4\right] = -RMSE\left[\sqrt{Q}\right]\left(1+\left|1+BIAS\right|\right)$$

where $\theta_1,\theta_2,\theta_3,\theta_4$ refer to four model parameters.

(2) The objective function of Eq.5 will be modified as follows:

$$Scenario\ 1:\ \Lambda=\prod_{c=1}^{C}\varepsilon_c\left[\theta_1(t,c),\theta_2(c),\theta_3(c),\theta_4(c)\big|\alpha(c),\beta,\omega(c)\right]\bullet f_N\left(\beta\big|\mu_2,\sigma_2\right)$$

$$Scenario\ 2:\ \Lambda=\prod_{c=1}^{C}\varepsilon_c\left[\theta_1(t,c),\theta_2(c),\theta_3(c),\theta_4(c)\big|\alpha(c),\beta(c),\omega\right]\bullet f_N\left(\omega\big|\mu_3,\sigma_3\right)$$

$$Scenario\ 3:\ \Lambda=\prod_{c=1}^{C}\varepsilon_c\left[\theta_1(t,c),\theta_2(c),\theta_3(c),\theta_4(c)\big|\alpha(c),\beta,\omega\right]\bullet\prod_{n=1}^{2}f_N\left(\beta,\omega\big|\mu_2,\sigma_2,\mu_3,\sigma_3\right)$$

$$Scenario\ 4:\ \Lambda=\prod_{c=1}^{C}\varepsilon_c\left[\theta_1(t,c),\theta_2(c),\theta_3(c),\theta_4(c)\right]$$

$$Scenario\ 5:\ \Lambda=\prod_{c=1}^{C}\varepsilon_c\left[\theta_1(c),\theta_2(c),\theta_3(c),\theta_4(c)\right]$$

where the number of catchments in the region is represented by C; $c$ represents the specific catchment; the $t$ is the time step.

A15: chapter 2.4.2: You don't say which settings of the sampling method you use (e.g. how many parameters you sample. . .)

**Reply:** Thanks. The sampling method used in this paper is the SCEM-UA algorithm. The detailed description of the settings of SCEM-UA algorithm will be added in the revised manuscript:

(1) Convergence is assessed by evolving three parallel chains with 30000 random samples, while verifying that the posterior distribution of parameters results in a value smaller than a Gelman-Rubin convergence value of 1.2 (Gelman et al., 2013).

(2) The number of unknown quantities in different scenarios are as follows: 15 in scenario 1 and scenario 2, 13 in scenario 3 and 18 in scenario 4.

A16: chapter 3.2.1: The dataset that you get is unbalanced, since there are more wet years. Is it taken into account? Does it have an effect on the calibration?

**Reply:** Thank you for pointing out this situation.

(1) Generally, calibration data should be longer than 3-6 years for daily hydrological modeling in order to get robust results (Perrin et al., 2003, Coron et al.,

2012). Thus, data from both dry period (15 years) and wet period (21 years) were used for model calibration to meet this requirement.

(2) Generally, a longer time series may improve the robustness of hydrological predictions. However, we tested the calibration performance with different lengths of records (> 10 years) in dry and non-dry periods and found that their results are almost the same. Therefore, we used both the length of 15 years of dry and 10 years non-dry periods into calibration in order to utilize all available data.

A17: chapter 3.2.3: Figures 7 and 8 are actually 8 and 9.

**Reply:** Thanks. Changes will be made as suggested.

A18: Figures 5, 6, 8, 9: Since you want to show a probability distribution I wouldn't use a boxplot but, instead, I suggest to use a violin plot (e.g.https://seaborn.pydata.org/examples/grouped_violinplots.html)

**Reply:** Thank you for your suggestions.

(1) Figures 8 and 9 will be modified as violin plot in the revised manuscript, which also could be found in response to comment A1 by Referee #1.

1     (2) Figures 5 and 6 will be revised as violin plot in the revised manuscript, which also could be found as follows:

[Figure]

2     Figure 5. $NSE_{sqrt}$ for each of the five scenarios for each catchment during (a) the calibration period (non-dry period) and (b) the verification period
3     (dry period).

[Figure]

Figure 6. NSE$_{sqrt}$ for each of the five scenarios for each catchment during (a) the calibration period (dry period) and (b) the verification period (non-dry period).

A19: Figures 8, 9: Why do you change the colors between beta and omega? This makes the plot more difficult to read.

**Reply:** Thanks. The same color will be used to the same parameter consistently in all figures. Changes will be made as suggested in the revised figures. Please refer to response to comment A1 by Referee #1.

**Supplement:**

**Table S1 The prior ranges of all unknown quantities in different scenarios**

**(1) Calibration in non-dry period and verification in dry period:**

Scenario 1:

| $\theta_{2\text{-}1}$ | $\theta_{2\text{-}1}$ | $\theta_{2\text{-}3}$ | $\mu_2$ | $\sigma_3$ | $\theta_{3\text{-}1}$ | $\theta_{4\text{-}1}$ | $\alpha_{1\text{-}1}$ | $\omega_{1\text{-}1}$ | $\theta_{3\text{-}2}$ | $\theta_{4\text{-}2}$ | $\alpha_{1\text{-}2}$ | $\omega_{1\text{-}2}$ | $\theta_{3\text{-}3}$ | $\theta_{4\text{-}3}$ | $\alpha_{1\text{-}3}$ | $\omega_{1\text{-}3}$ |
|---|---|---|---|---|---|---|---|---|---|---|---|---|---|---|---|---|
| -10 | -10 | -10 | -100 | 0 | 0.1 | 1 | 1 | 0.0001 | 0.1 | 0.5 | 100 | 0.0001 | 0.1 | 0.1 | 1 | 0.0001 |
| 10 | 10 | 10 | 100 | 6 | 200 | 10 | 600 | 0.4 | 300 | 20 | 1000 | 0.4 | 300 | 20 | 500 | 0.4 |

Scenario 2:

| $\theta_{2\text{-}1}$ | $\theta_{2\text{-}1}$ | $\theta_{2\text{-}3}$ | $\mu_3$ | $\sigma_3$ | $\theta_{3\text{-}1}$ | $\theta_{4\text{-}1}$ | $\alpha_{1\text{-}1}$ | $\beta_{1\text{-}1}$ | $\theta_{3\text{-}2}$ | $\theta_{4\text{-}2}$ | $\alpha_{1\text{-}2}$ | $\beta_{1\text{-}2}$ | $\theta_{3\text{-}3}$ | $\theta_{4\text{-}3}$ | $\alpha_{1\text{-}3}$ | $\beta_{1\text{-}3}$ |
|---|---|---|---|---|---|---|---|---|---|---|---|---|---|---|---|---|
| -6 | -6 | -6 | -0.4 | 0 | 1 | 0.5 | 1 | -300 | 1 | 0.1 | 100 | -300 | 0.1 | 2 | 1 | -200 |
| -6 | -6 | -6 | 0.4 | 0.1 | 500 | 10 | 600 | 300 | 300 | 20 | 600 | 500 | 400 | 20 | 800 | 300 |

Scenario 3:

| $\theta_{2\text{-}1}$ | $\theta_{2\text{-}1}$ | $\theta_{2\text{-}3}$ | $\mu_2$ | $\sigma_2$ | $\mu_3$ | $\sigma_3$ | $\theta_{3\text{-}1}$ | $\theta_{4\text{-}1}$ | $\alpha_{1\text{-}1}$ | $\theta_{3\text{-}2}$ | $\theta_{4\text{-}2}$ | $\alpha_{1\text{-}2}$ | $\theta_{3\text{-}3}$ | $\theta_{4\text{-}3}$ | $\alpha_{1\text{-}3}$ |
|---|---|---|---|---|---|---|---|---|---|---|---|---|---|---|---|
| -5 | -5 | -5 | -200 | 0 | -0 | 0 | 1 | 0.5 | 1 | 1 | 0.1 | 100 | 1 | 0.5 | 100 |
| 5 | 5 | 5 | 100 | 8 | 0.4 | 0.1 | 120 | 10 | 500 | 300 | 20 | 500 | 250 | 20 | 600 |

Scenario 4:

| $\theta_{2\text{-}1}$ | $\theta_{3\text{-}1}$ | $\theta_{4\text{-}1}$ | $\alpha_{1\text{-}1}$ | $\beta_{1\text{-}1}$ | $\omega_{1\text{-}1}$ | $\theta_{2\text{-}2}$ | $\theta_{3\text{-}2}$ | $\theta_{4\text{-}2}$ | $\alpha_{1\text{-}2}$ | $\beta_{1\text{-}2}$ | $\omega_{1\text{-}2}$ | $\theta_{2\text{-}3}$ | $\theta_{3\text{-}3}$ | $\theta_{4\text{-}3}$ | $\alpha_{1\text{-}3}$ | $\beta_{1\text{-}3}$ | $\omega_{1\text{-}3}$ |
|---|---|---|---|---|---|---|---|---|---|---|---|---|---|---|---|---|---|
| -10 | 1 | 0.1 | 1 | -300 | 0.0001 | -10 | 1 | 0.1 | 0 | -300 | 0 | -10 | 1 | 0.1 | 0 | -300 | 0.0001 |
| 10 | 500 | 10 | 800 | 300 | 0.4 | 10 | 500 | 10 | 800 | 300 | 0.4 | 10 | 500 | 10 | 800 | 300 | 0.4 |

**(2) Calibration in dry period and verification in dry period:**

Scenario 1:

| $\theta_{2-1}$ | $\theta_{2-2}$ | $\theta_{2-3}$ | $\mu_2$ | $\sigma_2$ | $\theta_{3-1}$ | $\theta_{4-1}$ | $\alpha_{1-1}$ | $\omega_{1-1}$ | $\theta_{3-2}$ | $\theta_{4-2}$ | $\alpha_{1-2}$ | $\omega_{1-2}$ | $\theta_{3-3}$ | $\theta_{4-3}$ | $\alpha_{1-3}$ | $\omega_{1-3}$ |
|---|---|---|---|---|---|---|---|---|---|---|---|---|---|---|---|---|
| -10 | -10 | -10 | -60 | 0 | 1 | 0.5 | 1 | 0 | 1 | 0.5 | 1 | 0 | 1 | 0.1 | 1 | 0 |
| 10 | 10 | 10 | 60 | 6 | 300 | 10 | 600 | 0.4 | 300 | 20 | 600 | 0.4 | 300 | 15 | 600 | 0.4 |

Scenario 2:

| $\theta_{2-1}$ | $\theta_{2-2}$ | $\theta_{2-3}$ | $\mu_3$ | $\sigma_3$ | $\theta_{3-1}$ | $\theta_{4-1}$ | $\alpha_{1-1}$ | $\beta_{1-1}$ | $\theta_{3-2}$ | $\theta_{4-2}$ | $\alpha_{1-2}$ | $\beta_{1-2}$ | $\theta_{3-3}$ | $\theta_{4-3}$ | $\alpha_{1-3}$ | $\beta_{1-3}$ |
|---|---|---|---|---|---|---|---|---|---|---|---|---|---|---|---|---|
| -10 | -10 | -10 | 0.0001 | 0 | 1 | 0.5 | 1 | -300 | 1 | 0.1 | 1 | -400 | 0.1 | 0.5 | 1 | -400 |
| 10 | 10 | 10 | 0.4 | 0.1 | 200 | 15 | 500 | 400 | 300 | 20 | 600 | 500 | 140 | 20 | 600 | 400 |

Scenario 3:

| $\theta_{2-1}$ | $\theta_{2-2}$ | $\theta_{2-3}$ | $\mu_2$ | $\sigma_2$ | $\mu_3$ | $\sigma_3$ | $\theta_{3-1}$ | $\theta_{4-1}$ | $\alpha_{1-1}$ | $\theta_{3-2}$ | $\theta_{4-2}$ | $\alpha_{1-2}$ | $\theta_{3-3}$ | $\theta_{4-3}$ | $\alpha_{1-3}$ |
|---|---|---|---|---|---|---|---|---|---|---|---|---|---|---|---|
| -10 | -10 | -10 | -80 | 0 | 0 | 0 | 1 | 0.5 | 1 | 1 | 0.1 | 1 | 1 | 0.1 | 1 |
| 10 | 10 | 10 | 80 | 6 | 0 | 0.1 | 200 | 10 | 500 | 400 | 20 | 600 | 400 | 20 | 600 |

Scenario 4:

| $\theta_{2-1}$ | $\theta_{3-1}$ | $\theta_{4-1}$ | $\alpha_{1-1}$ | $\beta_{1-1}$ | $\omega_{1-1}$ | $\theta_{2-2}$ | $\theta_{3-2}$ | $\theta_{4-2}$ | $\alpha_{1-2}$ | $\beta_{1-2}$ | $\omega_{1-2}$ | $\theta_{2-3}$ | $\theta_{3-3}$ | $\theta_{4-3}$ | $\alpha_{1-3}$ | $\beta_{1-3}$ | $\omega_{1-3}$ |
|---|---|---|---|---|---|---|---|---|---|---|---|---|---|---|---|---|---|
| -10 | 1 | 0.1 | 1 | -300 | 0.0001 | -10 | 1 | 0.1 | 1 | -300 | 0 | -10 | 1 | 0.1 | 1 | -300 | 0 |
| 10 | 500 | 10 | 800 | 300 | 0.4 | 10 | 500 | 10 | 800 | 300 | 0.4 | 10 | 500 | 10 | 800 | 300 | 0.4 |

Notes:

$\theta_{2-1}$, $\theta_{2-2}$ and $\theta_{2-3}$ refers to model parameter $\theta_2$ in catchment 225219, 405219 and 405264, respectively; $\theta_{3-1}$, $\theta_{3-2}$ and $\theta_{3-3}$ refer to model parameter $\theta_3$ in catchment 225219, 405219 and 405264, respectively; $\theta_{4-1}$, $\theta_{4-2}$ and $\theta_{4-3}$ refers to model parameter $\theta_4$ in catchment 225219, 405219 and 405264, respectively; $\mu_2$, $\sigma_2$, $\mu_3$ and $\sigma_3$ represent four hyper-parameters; $\alpha_{1-1}$, $\alpha_{1-2}$ and $\alpha_{1-3}$ refer to regression parameter $\alpha$ in catchment 225219, 405219 and

26  405264, respectively; $\beta_{1\text{-}1}$, $\beta_{1\text{-}2}$ and $\beta_{1\text{-}3}$ refer to regression parameter $\beta$ in catchment 225219, 405219 and 405264, respectively; $\omega_{1\text{-}1}$, $\omega_{1\text{-}2}$ and $\omega_{1\text{-}3}$

27  refer to regression parameter $\omega$ in catchment 225219, 405219 and 405264, respectively.

28

29

30

31  $$\varepsilon_c\left[\theta_1,\theta_2,\theta_3,\theta_4\right] = -RMSE\left[\sqrt{Q}\right]\left(1+\left|1+BIAS\right|\right)$$

32

$$Scenario\ 1:\ \Lambda = \prod_{c=1}^{C}\varepsilon_c\left[\theta_1(t,c),\theta_2(c),\theta_3(c),\theta_4(c)\middle|\alpha(c),\beta,\omega(c)\right]\bullet f_N\left(\beta\middle|\mu_2,\sigma_2\right)$$

$$Scenario\ 2:\ \Lambda = \prod_{c=1}^{C}\varepsilon_c\left[\theta_1(t,c),\theta_2(c),\theta_3(c),\theta_4(c)\middle|\alpha(c),\beta(c),\omega\right]\bullet f_N\left(\omega\middle|\mu_3,\sigma_3\right)$$

33  $$Scenario\ 3:\ \Lambda = \prod_{c=1}^{C}\varepsilon_c\left[\theta_1(t,c),\theta_2(c),\theta_3(c),\theta_4(c)\middle|\alpha(c),\beta,\omega\right]\bullet \prod_{n=1}^{2} f_N\left(\beta,\omega\middle|\mu_2,\sigma_2,\mu_3,\sigma_3\right)$$

$$Scenario\ 4:\ \Lambda = \prod_{c=1}^{C}\varepsilon_c\left[\theta_1(t,c),\theta_2(c),\theta_3(c),\theta_4(c)\right]$$

$$Scenario\ 5:\ \Lambda = \prod_{c=1}^{C}\varepsilon_c\left[\theta_1(c),\theta_2(c),\theta_3(c),\theta_4(c)\right]$$

---

## Author Comment (AC2) · 1 Apr 2019

General Comments The study of Pan et al. tests a Hierarchical Bayesian framework to incorporate time and spatial variability in model parameters. Specifically, the method was tested for the GR4J-model in three Australian catchments. Four modelling scenarios were tested, and one base scenario was formulated. The study shows that including spatially and temporally variable parameters improves model performance and reduces uncertainty. The article shows interesting work, which could be a nice contribution to the field. Generally, the article needs some more explanations on the method,

but there is also some incomplete reasoning. Hence, there are several issues I'd like to address. Specific comments B1: Key of the article is the hierarchical framework, but the authors may want to work on the explanation of the method. It is especially not clear to me how the hyper-parameters are determined, and how the catchment-specific values follow from that. Are the hyperparameters estimated in SCEM-UA? Or are these pre-defined? The gaussian distributions are defined by the authors as prior distributions, and that makes me assume that the model parameter theta is determined in SCEM-UA starting from this prior distribution, whereas the remaining model parameters are either kept fixed or sampled from a uniform distribution and independently for each catchment. Is that correct? Because if that is the case, the hyper-parameters (and hence the distribution) are determined in advance, so what are these based on? Besides, the choice of a gaussian distribution may seem a logical first guess, but it remains an arbitrary choice. So what is the reasoning behind this choice? In addition, the choice of the prior distribution may lead to some circular reasoning. When spatial coherence is used, the variation in performance goes down, but is this not just an artefact of the pre-defined gaussian distribution? In other words, if the prior distribution is set narrower, the resulting posterior distribution will probably be narrower as well. I believe it is therefore crucial to report also the prior ranges (or fixed values) for especially the (time-invariant) theta-parameter, but also all other model parameters. Reply: Thank you for your comment. Since that several sub-comments have been included in comment B1, for clarification, a point by point response to these sub-comments is made as follows. For example, B1S1 refers to the first sub-comment in B1.

B1S1: Key of the article is the hierarchical framework, but the authors may want to work on the explanation of the method. It is especially not clear to me how the hyper-parameters are determined, and how the catchment-specific values follow from that. Are the hyperparameters estimated in SCEM-UA? Or are these pre-defined? Reply: We apologize for this oversight. (1) All the hyper-parameters are not determined in advance. The hyper-parameters are sampled and determined with other unknown quantities simultaneously in the SCEM-UA algorithm. Actually, all other model parameters

(theta$_2$, $theta_3 and theta_4$, $except theta_1$) $are sampled simultaneously with regression parameters$ ($alpha, beta and omega (if pres$

$parameters (mu_2, sigma_2, mu_3 and sigma_3) in the SCEM$ −

$UA algorithm. In actual calculation process, we would set a large variation interval for each unknown quantity first, parameter$

$Rubin convergence value of 1.2 (Gelman et al., 2013) would be selected as the posterior probability distribution of parameters. (2) T$

is shared with linked catchments, while independent regression parameters $\omega1$-1, $\omega1$-2, and $\omega1$-3 are used to represent the frequency of model parameter theta$_1$ $in different catchments. The name of all unknown quantities in different scenarios could be found in the supplementary m$

B1S2: The gaussian distributions are defined by the authors as prior distributions, and that makes me assume that the model parameter theta is determined in SCEM-UA starting from this prior distribution, whereas the remaining model parameters are either kept fixed or sampled from a uniform distribution and independently for each catchment. Is that correct? Because if that is the case, the hyper-parameters (and hence the distribution) are determined in advance, so what are these based on? Besides, the choice of a gaussian distribution may seem a logical first guess, but it remains an arbitrary choice. So what is the reasoning behind this choice? In addition, the choice of the prior distribution may lead to some circular reasoning. When spatial coherence is used, the variation in performance goes down, but is this not just an artefact of the pre-defined gaussian distribution? In other words, if the prior distribution is set narrower, the resulting posterior distribution will probably be narrower as well. I believe it is therefore crucial to report also the prior ranges (or fixed values) for especially the (time-invariant) theta-parameter, but also all other model parameters. Reply: Thank you. (1) The Gaussian distribution is one of widely used distributions for describing the process level within the HB framework and has been applied in many previous studies, such as Sun et al (2015, 2016) and Chen et al (2014). The choice of the distribution is not our key point, so we just adopt a typical one from historical literatures. (2) Only the structure of Gaussian distribution and ranges of all unknown quantities were fixed in advance. The hyper-parameters and the prior distributions were obtained from Sun et al (2015,2016). All unknown quantities would be derived in the SCEM-UA algorithm. In addition, as illustrated in response to comment A14 by Referee 1, convergence for the

SCEM-UA algorithm is assessed by evolving three parallel chains with 30000 random samples, combined with the additional large prior ranges for all unknown quantities, is enough for ensuring a reliable result of all parameters. (3) The prior ranges of all unknown quantities in different scenarios are added in the supplementary material.

B2: I also wonder how valid it is to assume the catchments are similar. The authors state on p15.L314, that the catchments satisfy the homogeneity assumption. What is this assumption and how do they satisfy this assumption? A clear description of the catchments may be needed to defend that the catchments are the same. Just looking at the DEM and the annual values of rainfall and runoff (Table 2) give me the idea that the Big catchment (405264) behaves fundamentally different compared to the other two. This catchment also reached much higher performances in calibration (Fig.5 and 6) when no spatial coherence is used, and also shows different results in the BIAS comparison (Figure 7). Sometimes, the conclusions and statements of the authors do not seem to be strongly supported by the data as shown. The boxplots with performances (Figures 5, 6) show relatively similar performances, and, to be honest, a clear pattern is not very obvious. In addition, the authors tend to generalize in some cases findings that mainly apply to just two of the three catchments (see also my minor comments). I believe additional analyses may be needed to support the conclusions more, for example a statistical test to check if the distributions are significantly different. Or the addition of other, multiple performance measures, to assess the performance over multiple aspects (high flows, low flows etc.). Further, all beta-values plot around zero in Figure 8, basically pointing at the absence of a clear trend. Is this indeed true? It would be interesting to show the timeseries of the parameter. The absence of a trend may explain the similar performances for all scenarios, and especially also why the time-varying scenarios do not outperform the others clearly. Besides, when beta is around zero, there is no point of looking at omega, as this does not do much in that case. Concluding, the authors may need to clarify more what they did and how they arrive at several conclusions. I hope the authors find my comments useful, and I look forward to a revised manuscript. Reply: Thank you for your insightful comments.

B2S1: I also wonder how valid it is to assume the catchments are similar. The authors state on p15.L314, that the catchments satisfy the homogeneity assumption. What is this assumption and how do they satisfy this assumption? A clear description of the catchments may be needed to defend that the catchments are the same. Just looking at the DEM and the annual values of rainfall and runoff (Table 2) give me the idea that the Big catchment (405264) behaves fundamentally different compared to the other two. This catchment also reached much higher performances in calibration (Fig.5 and 6) when no spatial coherence is used, and also shows different results in the BIAS comparison (Figure 7). Reply: We apologize for our mistakes. (1) The homogeneity assumption will be deleted in the revised manuscript, because it is the spatial coherence of adjacent catchments that has been used as effective information to restrict the prediction uncertainties, so the homogeneity assumption of different catchments is not necessary. The studied catchments do have several similar characteristics: i) the average slope is similar, that is, catchment 225219 is 12.8, catchment 405219 is 10.7 and catchment 405264 is 9.7; ii) as shown in Table 2, these catchments have similar climatic conditions including mean annual potential evapotranspiration and rainfall patterns; iii) these catchments have experienced the same prolonged drought, and have similar amplitude of variation of rainfall and runoff between non-dry and dry periods.

B2S2: Sometimes, the conclusions and statements of the authors do not seem to be strongly supported by the data as shown. The boxplots with performances (Figures 5, 6) show relatively similar performances, and, to be honest, a clear pattern is not very obvious. Reply: We agree with the Referee that Figure 5 and 6 showed similar pattern in terms of the ranked orders of NSE amongst four scenarios. In addition, we have made point to point responses to all the ambiguous sentences with objective descriptions in the following technical comments raised by Referee 2.

B2S3: In addition, the authors tend to generalize in some cases findings that mainly apply to just two of the three catchments (see also my minor comments). I believe additional analyses may be needed to support the conclusions more, for example a
statistical test to check if the distributions are significantly different. Or the addition of other, multiple performance measures, to assess the performance over multiple aspects (high flows, low flows etc.). Reply: Thank you for your helpful comment. More information about the additional performance measures will be added in the revised manuscript. (1) Firstly, two performance measures based on high flows (i.e., mean annual maximum flow) and low flows (i.e., mean annual minimum flow) will be used to evaluate the high and low flows. The following paragraph will be added in section 2.5 in the revised manuscript. The fourth and fifth criteria are the Mean annual maximum flow (MaxF, mm/d) and Mean annual minimum flow (MinF, mm/d), which are used to qualify the performance of the high flows and low flows. These criteria are self-explanatory and have been used in many studies to assess the magnitude of maximum and minimum levels of flows (Ekstrom et al., 2018). Secondly, the following paragraph about the results of these measures will be added as Tables 6 and 7 in the revised manuscript. Table 6. Comparison of the projection performance during the verification period regarding the mean annual maximum flow (MaxF, mm/d) and mean annual minimum flow (MinF, mm/d) when model parameters were calibrated in the non-dry period and verified in the dry period.

Mean annual maximum flow Mean annual minimum flow 225219 405219 405264 225219 405219 405264 Observed 10.58 11.98 9.23 0.050 0.093 0.17 Scenario 1 13.30 5.64 6.68 0.050 0.045 0.13 Scenario 2 9.04 10.23 7.30 0.054 0.060 0.14 Scenario 3 10.91 7.66 9.75 0.041 0.092 0.16 Scenario 4 5.91 5.42 9.54 0.089 0.089 0.15 Scenario 5 5.07 6.03 7.98 0.086 0.086 0.12 Note: 1. The data in 1976 has been used for model warm-up to reduce the impact of the initial soil moisture conditions during the calibration period, and is not counted in the table; 2. The scenarios with bold values are labeled as the best scenario for projecting the streamflow during the verification periods.

Table 7. Comparison of the projection performance during the verification period associated with the Mean annual maximum flow (MaxF, mm/d) and Mean annual maximum

flow (MinF, mm/d) when model parameters were calibrated in the dry period and veri-fied in the non-dry period.

Mean annual maximum flow ãĂĂ Mean annual minimum flow 225219 405219 405264 ãĂĂ 225219 405219 405264 Observed 10.73 12.06 8.94 ãĂĂ 0.03 0.09 0.19 Scenario 1 12.40 6.87 12.90 0.03 0.04 0.09 Scenario 2 12.42 5.52 10.30 0.02 0.06 0.09 Scenario 3 10.95 10.67 8.37 0.03 0.05 0.10 Scenario 4 11.98 9.85 12.34 0.03 0.05 0.10 Sce-nario 5 14.19 9.45 11.97 ãĂĂ 0.02 0.05 0.10 Note: 1. The data in 1997 has been used for model warm-up to reduce the impact of the initial soil moisture conditions during the calibration period, and is not counted in the table; 2. The scenarios with bold values are labeled as the best scenario for projecting the streamflow during the verification periods. Thirdly, discussions of the results of these measures will be added in P21-22. L445-463 in the revised manuscript, which are as follows. Tables 6 and 7 illustrate the performance of high and low flows during the verification period in terms of MaxF and MinF estimates for the median projected streamflows in both DSST schemes. As shown in table 7, for the projection of high flow part, scenario 3 exhibits the best per-formance in all catchments among five scenarios under the scheme of calibrating in the dry period and verifying in the non-dry period. For the projection performance in the other DSST scheme (Table 6), scenario 3 has the best projection performance in high flow part in catchment 225219 and is the second best scenario in the other two catchments. It indicates that the incorporation of spatial coherence of both regression parameters omega and beta successfully improves the projection performance in the high flow part. As for the projection of the low flow part, the discrepancy between the results of different scenarios and the observed low flows is not obvious. Further-more, scenario 3 shows the best projected performance in two catchments (405219 and 405264) in the scheme of calibrating in dry period and verifying in non-dry period, and is the best scenario in catchment 405264 in the scheme of calibrating in non-dry period and verifying in dry period. In addition, scenario 3 is the second best option in catchment 225219 and 405219 under the scheme of calibrating in non-dry period and verifying in dry period. Combined with the projection performance of both high

and low flows, scenario 3 achieves its superior projection performance mainly by the improvement in the prediction of high flow parts.

B2S4: Further, all beta-values plot around zero in Figure 8, basically pointing at the absence of a clear trend. Is this indeed true? It would be interesting to show the timeseries of the parameter. The absence of a trend may explain the similar performances for all scenarios, and especially also why the time-varying scenarios do not outperform the others clearly. Besides, when beta is around zero, there is no point of looking at omega, as this does not do much in that case. Concluding, the authors may need to clarify more what they did and how they arrive at several conclusions. I hope the authors find my comments useful, and I look forward to a revised manuscript. Reply: We apologize for the mistakes in Figure 8 and 9. (1) As response to comment A1 by Referee 1, Figures 8 and 9 will be redrawn as follows: Please see the attacment Figure 8. Posterior distributions of the regression parameters ($\beta$ and $\omega$) for the production storage capacity ($\theta 1$) for the four model scenarios in each catchment when calibrated in the non-dry period and verified in the dry period. The solid horizontal lines within the violin plots denote the 25th and 75th percentiles of the posterior distribution, while the dotted line denotes median estimates. Please see the attacment Figure 9. Posterior distributions of the regression parameters ($\beta$ and $\omega$) for the production storage capacity ($\theta 1$) for the four model scenarios in each catchment when calibrated in the dry period and verified in the non-dry period. The solid horizontal lines within the violin plots denote the 25th and 75th percentiles of the posterior distribution, while the dotted line denotes median estimates. (2) As discussed in the section 3.2.2, model parameter theta$_1$ $is\ time-varying\ and\ no\ spatial\ coherence\ is\ considered (beta = 0)\ in\ scenario 4, while\ theta_1\ is\ stationary\ and\ of\ course\ no\ spatial\ coherence\ is\ included\ in\ scenario 5 (beta = 0). Scenario 4, had\ a\ higher\ median\ NSE sqrt performance\ than\ that\ of\ scenario 5\ in\ five\ of\ six\ options (except\ catchment 405219 in\ varying scheme\ for\ improving\ the\ model\ performance. Compared\ with\ scenario 5, the\ introduction\ of\ additional\ regression\ par\ varying\ model\ parameter\ theta_1\ would\ change\ its\ values\ in\ each\ time\ step, the\ regression\ parameter\ beta, as\ the\ amplitude\ of\ the\ s\ 39.20\ in\ scenario 4\ in\ the\ scheme\ of\ calibrating\ in\ the\ non-dry\ period\ and\ verifying\ in\ the\ dry\ period. (4) After\ calculation, param\ varying\ one. Technical\ corrections B3: P.7. section 2.1.1. Please\ elaborate\ on\ how\ the\ dry\ periods\ are\ defined? Reply:$

B4: P8. Section 2.1.2. Why add this paragraph when you only refer to section 2.5? Reply: We apologize for this misunderstanding. This paragraph will be modified as follows: In the DSST method, the model parameters calibrated in the non-dry period were evaluated in the dry period, and vice versa. In addition, criteria, i.e, NSEsqrt, BIAS and DIC illustrated in the section 2.5, were used to evaluated the performance of the calibrated parameters for different transfer schemes.

B5: P10. L210 Do you mean Eq. 1? Reply: We are sorry for this oversight. The phase "Eq.2" will be revised as "Eq.1" in the revised manuscript.

B6: P10.L210 ...expected to the same. . . ! expected to be the same Reply: Thanks. Change will be made as suggested.

B7: P12. L50. Please define N and n Reply: Thanks. N refers to the Gaussian distribution and n represents the number of regression parameters that are spatially coherent. The definitions of N and n will be added in the revised manuscript.

B8: P12.L258. Which parameters are optimized in SCEM-UA? Reply: Thanks. All unknown quantities of different scenarios that needed to be optimized in SCEM-UA have been added in the supplementary material.

B9: P15.L326. Please explain how I can see this from Figure 4, except for the pre-defined red colour. Is this where the black line crosses the axis? Why are the first years not considered? Reply: Thanks. The bars in blue and red colors in Figure 4 represent annual rainfall anomalies during the non-dry and dry periods, respectively. The black line is annual anomaly of rainfall smoothed with the 3-year moving window. The start of the dry period is defined as the start of first 3-year consecutive negative anomaly period based on Saft et al (2015). According to the definition of dry period, the start of the dry period is not the place where the black line crosses the axis. Because the years near the cross point have positive rainfall anomaly. Similar comment is also
raised by the Referee 1 (see comment A3 and our corresponding response). In the revised manuscript, the definition of the dry periods will be added.

B10: P16.L339-340. Are these references in the right place? You describe your own results, shouldn't you refer to one of the figures? Reply: Thank you for your comment. These references will be deleted, and the sentence will be modified as follows: As shown in Figures 5(a), 6(a) and 7, the calibrated model parameters yielded good simulation performance over the calibrated periods for all criteria.

B11: P17.L355-357. This is, as far as I can see, not true for all catchments. Catchments 225219 and 405264 have a higher median, but the variation is less for 225219. Reply: We apologize for this mistake. In figure 5(b), the variation of NSEsqrt in scenario 4 is less than that in scenario 5. The phrase of the comparison of variation will be deleted in the revised manuscript, because in this sentence we focus on the advantage of scenario 4, i.e., the improvement in median NSEsqrt performance. This sentence will be modified as follows: Scenario 4 had a higher median NSEsqrt performance than scenario 5 in catchments 225219 and 405264, and was slightly inferior than the latter in catchment 405219, which indicates the validity of the time-varying scheme for improving the model performance.

B12: P17.L362. As far as I can see, it has only the highest median value for catchment 225219. Reply: We apologize for our mistakes. This sentence will be modified as follows: In the DSST scheme of calibrating in the dry period and verifying in the non-dry period, scenario 3, which both considered spatial coherence of regression parameters beta and omega between different catchments, exhibited the highest median NSEsqrt for all catchments, had the smallest fluctuation range in two catchments (405219 and 405264) and is the second smallest scenario in catchment 22519 during the verification period. In the other DSST scheme, scenario 3 exhibited the smallest fluctuation range of NSEsqrt estimate for all catchments, showed the highest median value in catchment 225219, and was the second best scenario in the other two catchments (405219 and 405264) during the verification period.

B13: P18.L375. The performances in the verification period seem higher to me? What do you mean calibrated performances were inferior? Reply: We are sorry for this misunderstanding. In the scheme of calibration in dry period and verification in non-dry period, it is true that the NSE sqrt during the verification period is higher than that in the calibration period. However, the projection performance calibrated using a contrasting climatic condition was inferior to the simulation performance that was directly calibrated from the climatic condition, compared with Figure 5(a) and 6(b), or Figure 6(a) and 5(b). For example, the NSEsqrt performance in Figure 6(b) is inferior to which in Figure 5(a). In the other words, for the non-dry period: ; for the dry period: . This sentence will be modified as follows in the revised manuscript: "However, the projection performance calibrated using a contrasting climatic condition was inferior to the simulation performance that was directly calibrated from the climatic condition, compared with Figure 5(a) and 6(b), or Figure 6(a) and 5(b). For example, the NSEsqrt performance in Figure 6(b) is inferior to which in Figure 5(a)." Hope the revision is clear to the referee and readers.

B14: P.18L375-377. This is not true for catchment 225219 Reply: We apologize for our mistakes. Follow the referee's comment, this sentence will be modified to specify the problem, which is as follows: By comparing scenarios in the calibration period, it was found that scenarios 4 and 5 exhibited the highest performance in two of three catchments (405219 and 405264), followed successively by scenario 3, scenario 2, and scenario 1.

B15: P18.L379. The ranges seem not very different between scenarios 4 and 5, only slightly. Reply: Thanks. This sentence will be modified as follows: During the verification period, the median NSEsqrt performance in scenario 4 was 0.80

B16: P18.L379-380. It's not very obvious that scenario 3 has a higher median performance for catchment 405264. Reply: Thanks. This sentence will be modified as follows: In catchment 405264, compared with scenarios 1, 2, 4 and 5, scenario 3 showed an 8.1

B17: P18.L382 This is not very obvious to me. Reply: Thank you. This sentence will be modified as follows: During the verification period, the median NSEsqrt performance in scenario 4 was 0.80

B18: P18.L394. Compared –> comparing Reply: Thanks. Change will be made as suggested.

B19: P20.L438. Is omega for scenario 4 not the lowest in all cases? Or do you mean the absolute values? Reply: We apologize for our mistakes. This should be regression parameter beta in this place rather than omega. The catchment average of the median estimates of beta in the first three scenarios are 2.78, -4.91, and 9.26 respectively, while that in the fourth scenario is much larger, reached at -39.20. Scenario 3, which considered both spatial coherence of regression parameters beta and omega, has the narrowest interval of beta for all catchments, followed successively by scenario 1 (only considered the spatial coherence of the regression parameter beta, scenario 2 (only parameter omega was spatially coherent), and scenario 4 (no parameter was spatially coherent). With regards to the regression parameter omega, which denotes the frequency of the sine function (in the lower figures of Figures 8 and 9), its median estimates and variation ranges in both four scenarios differ slightly. The former reached a catchment average of 0.19,0.20,0.19,0.17 for different scenarios.

B20: Figure 2. Please define all symbols and abbreviations in the figure. Reply: Thanks. All symbols and abbreviations in the Figure 2 will be defined in the revised manuscript. Please see the attachments-Figure 2 Figure 2. Schematic diagram of the GR4J rainfall-runoff model adopted from Perrin et al. (2003). In the figure, P and E refer to precipitation and evapotranspiration, respectively; En and Pn denote net precipitation and net evapotranspiration, respectively; Ps refers part of precipitation that fills the production store (i.e. S). The production store is determined as a function of the water level S in production store. The $\theta 1, \theta 2, \theta 3$, and $\theta 4$ denote model parameters. The Perc refers to the percolation leakage that is a function of production store S and parameter $\theta 1$. The Pr refers to total quantity of water that reaches the routing functions.

The UH1 and UH2 denote two unit hydrographs. The Q1 and Q9 refer the corresponding output of the unit hydrographs, respectively; F indicates the groundwater exchange term; R is the level in the routing store. The Qr refers to the outflow of the routing store, Qd is a function of water exchange, and Q refers to the total streamflow.

B21: Figure 5,6: I would suggest to plot the boxes for calibration and verification next to each other. It's easier to see whether there is an improvement or not. Please also add the units (also when a unitless number is presented) Reply: Thanks. Changes will be made as suggested.

B22: Figure 7. Please make the labels and text bigger. Reply: Thanks. Changes will be made as suggested. The modified Figure 7 will be as follows: Please see the attachments-Figure 7 Figure 7. BIAS performance of Qmedian for five scenarios in all catchments. The BIAS is plotted as a 10-year moving average, and 10-year moving average streamflows are plotted for reference. The left-hand three graphs are calibrated in the non-dry period and then verified in the dry period, while the opposite sequence applies to the right-hand graphs.

B23: Figure 8, 9. Maybe use the same colors for the scenarios in both plots. What are the units of beta and omega? Reply: Thanks for thoughtful comment. Similar comment has been raised by Referee 1 (see comment A1). The same color for the scenarios in both plots will be used. Both beta and omega are unitless. Figures 8 and 9 will be modified as violin plots in the revised manuscript (see response to A1).

Please also note the supplement to this comment:
https://www.hydrol-earth-syst-sci-discuss.net/hess-2019-6/hess-2019-6-AC2-supplement.pdf

—————————————

[Figure]

[Figure]

**Fig. 1.** Figure 8 Posterior distributions of the regression parameters ($\beta$ and $\omega$) for the production storage capacity ($\theta 1$) for the four modeling scenarios in all the 3 studied catchments. In this figure, param

[Figure]

**Fig. 2.** Figure 9 Posterior distributions of the regression parameters ($\beta$ and $\omega$) for the production storage capacity ($\theta 1$) for the four model scenarios in all 3 studied catchments. In this figure, parameters we

[Figure]

**Fig. 3.** Figure 7. Long-term simulation BIAS of Qmedian for five scenarios in all catchments. Simulation BIAS is plotted as a 10-year moving average, and 10-year moving average stream-flows are plotted for refe

[Figure]

**Fig. 4.** Figure 2. Schematic diagram of the GR4J rainfall-runoff model adopted from Perrin et al. (2003). In the figure, P and E refer to precipitation and evapotranspiration, respectively; En and Pn denote ne

**Supplement:**

**General Comments**

The study of Pan et al. tests a Hierarchical Bayesian framework to incorporate time and spatial variability in model parameters. Specifically, the method was tested for the GR4J-model in three Australian catchments. Four modelling scenarios were tested, and one base scenario was formulated. The study shows that including spatially and temporally variable parameters improves model performance and reduces uncertainty. The article shows interesting work, which could be a nice contribution to the field. Generally, the article needs some more explanations on the method, but there is also some incomplete reasoning. Hence, there are several issues I'd like to address.

**Specific comments**

B1: Key of the article is the hierarchical framework, but the authors may want to work on the explanation of the method. It is especially not clear to me how the hyper-parameters are determined, and how the catchment-specific values follow from that. Are the hyperparameters estimated in SCEM-UA? Or are these pre-defined? The gaussian distributions are defined by the authors as prior distributions, and that makes me assume that the model parameter theta is determined in SCEM-UA starting from this prior distribution, whereas the remaining model parameters are either kept fixed or sampled from a uniform distribution and independently for each catchment. Is that correct? Because if that is the case, the hyper-parameters (and hence the distribution) are determined in advance, so what are these based on? Besides, the choice of a gaussian distribution may seem a logical first guess, but it remains an arbitrary choice. So what is the reasoning behind this choice? In addition, the choice of the prior distribution may lead to some circular reasoning. When spatial coherence is used, the variation in performance goes down, but is this not just an artefact of the pre-defined gaussian distribution? In other words, if the prior distribution is set narrower, the resulting posterior distribution will probably be narrower as well. I believe it is therefore crucial to report also the prior ranges (or fixed values) for especially the (time-invariant) theta-parameter, but also all other model parameters.

**Reply:** Thank you for your comment. Since that several sub-comments have been included in comment B1, for clarification, a point by point response to these sub-comments is made as follows. For example, B1S1 refers to the first sub-comment in B1.

B1S1: Key of the article is the hierarchical framework, but the authors may want to work on the explanation of the method. It is especially not clear to me how the hyper-parameters are determined, and how the catchment-specific values follow from that. Are the hyperparameters estimated in SCEM-UA? Or are these pre-defined?

**Reply:** We apologize for this oversight.

(1) All the hyper-parameters are not determined in advance. The hyper-parameters are sampled and determined with other unknown quantities simultaneously in the SCEM-UA algorithm. Actually, all other model parameters ($\theta_2$, $\theta_3$, and $\theta_4$, except $\theta_1$) are sampled simultaneously with regression parameters ($\alpha$, $\beta$ and $\omega$ (if present)) and hyper-parameters ($\mu_2$, $\sigma_2$, $\mu_3$ and $\sigma_3$) in the SCEM-UA algorithm. In actual calculation process, we would set a large variation interval for each unknown quantity first, parameters would converge to a small interval in MCMC calculation process, the final parameter samples that satisfy the requirement that a GR value must be smaller than a Gelman-Rubin convergence value of 1.2 (Gelman et al., 2013) would be selected as the posterior probability distribution of parameters.

(2) The $\omega$ not linked in scenario 1, while $\beta$ is not linked in scenario 2. In scenario 4, both $\omega$ and $\beta$ are not linked. Spatially irrelevant parameters would be sampled and derived as independent variables. For example, in scenario 4, regression parameters $\omega$ and $\beta$ of different catchments are not linked, thus values of $\omega$ and $\beta$ of each catchment are calibrated from corresponding catchment inputs. In scenario 1, regression parameter $\beta(c) = N(\mu_3, \sigma^2)$, which means that $\beta$ is shared with linked catchments, while independent regression parameters $\omega_{1-1}$, $\omega_{1-2}$, and $\omega_{1-3}$ are used to represent the frequency of model parameter $\theta_1$ in different catchments. The name of all unknown quantities in different scenarios could be found in the supplementary material.

B1S2: The gaussian distributions are defined by the authors as prior distributions, and that makes me assume that the model parameter theta is determined in SCEM-UA starting from this prior distribution, whereas the remaining model parameters are either kept fixed or sampled from a uniform distribution and independently for each catchment. Is that correct? Because if that is the case, the hyper-parameters (and hence the distribution) are determined in advance, so what are these based on? Besides, the choice of a gaussian distribution may seem a logical first guess, but it remains an arbitrary choice. So what is the reasoning behind this choice? In addition, the choice of the prior distribution may lead to some circular reasoning. When spatial coherence is used, the variation in performance goes down, but is this not just an artefact of the pre-defined gaussian distribution? In other words, if the prior distribution is set narrower, the resulting posterior distribution will probably be narrower as well. I believe it is therefore crucial to report also the prior ranges (or fixed values) for especially the (time-invariant) theta-parameter, but also all other model parameters.

**Reply:** Thank you.

(1) The Gaussian distribution is one of widely used distributions for describing the process level within the HB framework and has been applied in many previous studies, such as Sun et al (2015, 2016) and Chen et al (2014). The choice of the distribution is not our key point, so we just adopt a typical one from historical literatures.

(2) Only the structure of Gaussian distribution and ranges of all unknown quantities were fixed in advance. The hyper-parameters and the prior distributions were obtained from Sun et al (2015,2016). All unknown quantities would be derived in the SCEM-UA algorithm. In addition, as illustrated in response to comment A14 by Referee #1, convergence for the SCEM-UA algorithm is assessed by evolving three parallel chains with 30000 random samples, combined with the additional large prior ranges for all unknown quantities, is enough for ensuring a reliable result of all parameters.

(3) The prior ranges of all unknown quantities in different scenarios are added in the supplementary material.

B2: I also wonder how valid it is to assume the catchments are similar. The authors state on p15.L314, that the catchments satisfy the homogeneity assumption. What is this assumption and how do they satisfy this assumption? A clear description of the catchments may be needed to defend that the catchments are the same. Just looking at the DEM and the annual values of rainfall and runoff (Table 2) give me the idea that the Big catchment (405264) behaves fundamentally different compared to the other two. This catchment also reached much higher performances in calibration (Fig.5 and 6) when no spatial coherence is used, and also shows different results in the BIAS comparison (Figure 7).

Sometimes, the conclusions and statements of the authors do not seem to be strongly supported by the data as shown. The boxplots with performances (Figures 5, 6) show relatively similar performances, and, to be honest, a clear pattern is not very obvious. In addition, the authors tend to generalize in some cases findings that mainly apply to just two of the three catchments (see also my minor comments). I believe additional analyses may be needed to support the conclusions more, for example a statistical test to check if the distributions are significantly different. Or the addition of other, multiple performance measures, to assess the performance over multiple aspects (high flows, low flows etc.). Further, all beta-values plot around zero in Figure 8, basically pointing at the absence of a clear trend. Is this indeed true? It would be interesting to show the timeseries of the parameter. The absence of a trend may explain the similar performances for all scenarios, and especially also why the time-varying scenarios do not outperform the others clearly. Besides, when beta is around zero, there is no point of looking at omega, as this does not do much in that case.

Concluding, the authors may need to clarify more what they did and how they arrive at several conclusions. I hope the authors find my comments useful, and I look forward to a revised manuscript.

**Reply:** Thank you for your insightful comments.

B2S1: I also wonder how valid it is to assume the catchments are similar. The authors state on p15.L314, that the catchments satisfy the homogeneity assumption. What is this assumption and how do they satisfy this assumption? A clear description of the catchments may be needed to defend that the catchments are the same. Just looking at the DEM and the annual values of rainfall and runoff (Table 2) give me the idea that the Big catchment (405264) behaves fundamentally different compared to the other two. This catchment also reached much higher performances in calibration (Fig.5 and 6) when no spatial coherence is used, and also shows different results in the BIAS comparison (Figure 7).

**Reply:** We apologize for our mistakes.

(1) The homogeneity assumption will be deleted in the revised manuscript,

because it is the spatial coherence of adjacent catchments that has been used as effective

information to restrict the prediction uncertainties, so the homogeneity assumption of different catchments is not necessary. The studied catchments do have several similar characteristics: i) the average slope is similar, that is, catchment 225219 is 12.8, catchment 405219 is 10.7 and catchment 405264 is 9.7; ii) as shown in Table 2, these catchments have similar climatic conditions including mean annual potential evapotranspiration and rainfall patterns; iii) these catchments have experienced the same prolonged drought, and have similar amplitude of variation of rainfall and runoff between non-dry and dry periods.

B2S2: Sometimes, the conclusions and statements of the authors do not seem to be strongly supported by the data as shown. The boxplots with performances (Figures 5, 6) show relatively similar performances, and, to be honest, a clear pattern is not very obvious.

**Reply:** We agree with the Referee that Figure 5 and 6 showed similar pattern in terms of the ranked orders of NSE amongst four scenarios. In addition, we have made point to point responses to all the ambiguous sentences with objective descriptions in the following technical comments raised by Referee #2.

B2S3: In addition, the authors tend to generalize in some cases findings that mainly apply to just two of the three catchments (see also my minor comments). I believe additional analyses may be needed to support the conclusions more, for example a statistical test to check if the distributions are significantly different. Or the addition of other, multiple performance measures, to assess the performance over multiple aspects (high flows, low flows etc.).

**Reply:** Thank you for your helpful comment. More information about the additional performance measures will be added in the revised manuscript.

(1) Firstly, two performance measures based on high flows (i.e., mean annual

maximum flow) and low flows (i.e., mean annual minimum flow) will be used to evaluate the high and low flows. The following paragraph will be added in section 2.5 in the revised manuscript.

The fourth and fifth criteria are the Mean annual maximum flow (MaxF, mm/d) and Mean annual minimum flow (MinF, mm/d), which are used to qualify the performance of the high flows and low flows. These criteria are self-explanatory and have been used in many studies to assess the magnitude of maximum and minimum levels of flows (Ekstrom et al., 2018).

(1) Secondly, the following paragraph about the results of these measures will be added as Tables 6 and 7 in the revised manuscript.

Table 6. Comparison of the projection performance during the verification period regarding the mean annual maximum flow (MaxF, mm/d) and mean annual minimum flow (MinF, mm/d) when model parameters were calibrated in the non-dry period and verified in the dry period.

|  | Mean annual maximum flow | | | Mean annual minimum flow | | |
|---|---|---|---|---|---|---|
|  | 225219 | 405219 | 405264 | 225219 | 405219 | 405264 |
| Observed | 10.58 | 11.98 | 9.23 | 0.050 | 0.093 | 0.17 |
| Scenario 1 | 13.30 | 5.64 | 6.68 | **0.050** | 0.045 | 0.13 |
| Scenario 2 | 9.04 | **10.23** | 7.30 | 0.054 | 0.060 | 0.14 |
| Scenario 3 | **10.91** | 7.66 | 9.75 | 0.041 | **0.092** | **0.16** |
| Scenario 4 | 5.91 | 5.42 | **9.54** | 0.089 | 0.089 | 0.15 |
| Scenario 5 | 5.07 | 6.03 | 7.98 | 0.086 | 0.086 | 0.12 |

Note: 1. The data in 1976 has been used for model warm-up to reduce the impact of the initial soil moisture conditions during the calibration period, and is not counted in the table; 2. The scenarios with bold values are labeled as the best scenario for projecting the streamflow during the verification periods.

Table 7. Comparison of the projection performance during the verification period associated with the Mean annual maximum flow (MaxF, mm/d) and Mean annual maximum flow (MinF, mm/d) when model parameters were calibrated in the dry period and verified in the non-dry period.

|  | Mean annual maximum flow | Mean annual minimum flow |
|---|---|---|

|  | 225219 | 405219 | 405264 | 225219 | 405219 | 405264 |
|---|---|---|---|---|---|---|
| Observed | 10.73 | 12.06 | 8.94 | 0.03 | 0.09 | 0.19 |
| Scenario 1 | 12.40 | 6.87 | 12.90 | 0.03 | 0.04 | 0.09 |
| Scenario 2 | 12.42 | 5.52 | 10.30 | 0.02 | **0.06** | 0.09 |
| Scenario 3 | **10.95** | **10.67** | **8.37** | **0.03** | 0.05 | 0.10 |
| Scenario 4 | 11.98 | 9.85 | 12.34 | 0.03 | 0.05 | **0.10** |
| Scenario 5 | 14.19 | 9.45 | 11.97 | 0.02 | 0.05 | 0.10 |

Note: 1. The data in 1997 has been used for model warm-up to reduce the impact of the initial soil moisture conditions during the calibration period, and is not counted in the table; 2. The scenarios with bold values are labeled as the best scenario for projecting the streamflow during the verification periods.

(2) Thirdly, discussions of the results of these measures will be added in P21-22. L445-463 in the revised manuscript, which are as follows.

Tables 6 and 7 illustrate the performance of high and low flows during the verification period in terms of MaxF and MinF estimates for the median projected streamflows in both DSST schemes. As shown in table 7, for the projection of high flow part, scenario 3 exhibits the best performance in all catchments among five scenarios under the scheme of calibrating in the dry period and verifying in the non-dry period. For the projection performance in the other DSST scheme (Table 6), scenario 3 has the best projection performance in high flow part in catchment 225219 and is the second best scenario in the other two catchments. It indicates that the incorporation of spatial coherence of both regression parameters $\beta$ and $\omega$ successfully improves the projection performance in the high flow part. As for the projection of the low flow part, the discrepancy between the results of different scenarios and the observed low flows is not obvious. Furthermore, scenario 3 shows the best projected performance in two catchments (405219 and 405264) in the scheme of calibrating in dry period and verifying in non-dry period, and is the best scenario in catchment 405264 in the scheme

of calibrating in non-dry period and verifying in dry period. In addition, scenario 3 is the second best option in catchment 225219 and 405219 under the scheme of calibrating in non-dry period and verifying in dry period. Combined with the projection performance of both high and low flows, scenario 3 achieves its superior projection performance mainly by the improvement in the prediction of high flow parts.

B2S4: Further, all beta-values plot around zero in Figure 8, basically pointing at the absence of a clear trend. Is this indeed true? It would be interesting to show the timeseries of the parameter. The absence of a trend may explain the similar performances for all scenarios, and especially also why the time-varying scenarios do not outperform the others clearly. Besides, when beta is around zero, there is no point of looking at omega, as this does not do much in that case.

Concluding, the authors may need to clarify more what they did and how they arrive at several conclusions. I hope the authors find my comments useful, and I look forward to a revised manuscript.

**Reply:** We apologize for the mistakes in Figure 8 and 9.

(1) As response to comment A1 by Referee #1, Figures 8 and 9 will be redrawn as follows:

[Figure]

Figure 8. Posterior distributions of the regression parameters ($\beta$ and $\omega$) for the production storage capacity ($\theta_1$) for the four model scenarios in each catchment

when calibrated in the non-dry period and verified in the dry period. The solid horizontal lines within the violin plots denote the 25th and 75th percentiles of the posterior distribution, while the dotted line denotes median estimates.

[Figure]

Figure 9. Posterior distributions of the regression parameters (β and ω) for the production storage capacity ($\theta_1$) for the four model scenarios in each catchment when calibrated in the dry period and verified in the non-dry period. The solid horizontal lines within the violin plots denote the 25th and 75th percentiles of the posterior distribution, while the dotted line denotes median estimates.

(2) As discussed in the section 3.2.2, model parameter $\theta_1$ is time-varying and no spatial coherence is considered ($\beta \neq 0$) in scenario 4, while $\theta_1$ is stationary and of course no spatial coherence is included in scenario 5 ($\beta = 0$). Scenario 4, had a higher median NSE$_{sqrt}$ performance than that of scenario 5 in five of six options (except catchment 405219 in the first DSST scheme), which indicates the validity of the time-varying scheme for improving the model performance. Compared with scenario 5, the introduction of additional regression parameters ($\alpha, \beta$ and $\omega$) in scenario 4 at the same time amplified the model projection uncertainty in two of three catchments (225219 and 405264). However, the appropriate adoption of spatial coherence alleviates this problem. Scenario 3, which both considered spatial coherence of regression parameters

$\beta$ and $\omega$ between different catchments, exhibited the optimal median NSE$_{sqrt}$, DIC, and MaxF estimates in most options during the verification period, which illustrated the validity of the inclusion of the spatial coherence of regression parameters $\beta$ and $\omega$.

(3) The median estimates of $\beta$ in Figures 8 and 9 are not equal to zero. Because the facts that the adopted hydrological model is on a daily scale and the assumed time-varying model parameter $\theta_1$ would change its values in each time step, the regression parameter $\beta$, as the amplitude of the sine term, is supposed to have a small absolute value rather a large one. As shown in Figure 8, the catchment average of the median estimate of $\beta$ are 2.78 in scenario 1, -4.91 in scenario 2, 9.26 in scenario 3, and -39.20 in scenario 4 in the scheme of calibrating in the non-dry period and verifying in the dry period.

(4) After calculation, parameter $\beta$ has a deterministic posterior distribution (as shown in Figures 8 and 9) rather than a time-varying one.

**Technical corrections**

B3: P.7. section 2.1.1. Please elaborate on how the dry periods are defined.

**Reply:** Thank you. Please refer to response of comment A3 about the detailed definition of the dry period, which will be added in the revised manuscript.

B4: P8. Section 2.1.2. Why add this paragraph when you only refer to section 2.5?

**Reply:** We apologize for this misunderstanding. This paragraph will be modified as follows:

In the DSST method, the model parameters calibrated in the non-dry period were evaluated in the dry period, and vice versa. In addition, criteria, i.e, NSE$_{sqrt}$, BIAS and

DIC illustrated in the section 2.5, were used to evaluated the performance of the calibrated parameters for different transfer schemes.

B5: P10. L210 Do you mean Eq. 1?

**Reply:** We are sorry for this oversight. The phase "Eq.2" will be revised as "Eq.1" in the revised manuscript.

B6: P10.L210 ...expected to the same. . . ! expected to be the same

**Reply:** Thanks. Change will be made as suggested.

B7: P12. L50. Please define N and n

**Reply:** Thanks. $N$ refers to the Gaussian distribution and $n$ represents the number of regression parameters that are spatially coherent. The definitions of $N$ and $n$ will be added in the revised manuscript.

B8: P12.L258. Which parameters are optimized in SCEM-UA?

**Reply:** Thanks. All unknown quantities of different scenarios that needed to be optimized in SCEM-UA have been added in the supplementary material.

B9: P15.L326. Please explain how I can see this from Figure 4, except for the pre-defined red colour. Is this where the black line crosses the axis? Why are the first years not considered?

**Reply:** Thanks. The bars in blue and red colors in Figure 4 represent annual rainfall anomalies during the non-dry and dry periods, respectively. The black line is annual anomaly of rainfall smoothed with the 3-year moving window.

The start of the dry period is defined as the start of first 3-year consecutive negative anomaly period based on Saft et al (2015). According to the definition of dry period, the start of the dry period is not the place where the black line crosses the axis. Because

the years near the cross point have positive rainfall anomaly. Similar comment is also raised by the Referee #1 (see comment A3 and our corresponding response). In the revised manuscript, the definition of the dry periods will be added.

B10: P16.L339-340. Are these references in the right place? You describe your own results, shouldn't you refer to one of the figures?

**Reply:** Thank you for your comment. These references will be deleted, and the sentence will be modified as follows:

As shown in Figures 5(a), 6(a) and 7, the calibrated model parameters yielded good simulation performance over the calibrated periods for all criteria.

B11: P17.L355-357. This is, as far as I can see, not true for all catchments. Catchments 225219 and 405264 have a higher median, but the variation is less for 225219.

**Reply:** We apologize for this mistake. In figure 5(b), the variation of $NSE_{sqrt}$ in scenario 4 is less than that in scenario 5. The phrase of the comparison of variation will be deleted in the revised manuscript, because in this sentence we focus on the advantage of scenario 4, i.e., the improvement in median $NSE_{sqrt}$ performance. This sentence will be modified as follows:

Scenario 4 had a higher median $NSE_{sqrt}$ performance than scenario 5 in catchments 225219 and 405264, and was slightly inferior than the latter in catchment 405219, which indicates the validity of the time-varying scheme for improving the model performance.

B12: P17.L362. As far as I can see, it has only the highest median value for catchment 225219.

**Reply:** We apologize for our mistakes. This sentence will be modified as follows:

In the DSST scheme of calibrating in the dry period and verifying in the non-dry

period, scenario 3, which both considered spatial coherence of regression parameters $\beta$ and $\omega$ between different catchments, exhibited the highest median NSE$_{sqrt}$ for all catchments, had the smallest fluctuation range in two catchments (405219 and 405264) and is the second smallest scenario in catchment 22519 during the verification period. In the other DSST scheme, scenario 3 exhibited the smallest fluctuation range of NSE$_{sqrt}$ estimate for all catchments, showed the highest median value in catchment 225219, and was the second best scenario in the other two catchments (405219 and 405264) during the verification period.

B13: P18.L375. The performances in the verification period seem higher to me? What do you mean calibrated performances were inferior?

**Reply:** We are sorry for this misunderstanding. In the scheme of calibration in dry period and verification in non-dry period, it is true that the NSE $_{sqrt}$ during the verification period is higher than that in the calibration period. However, the projection performance calibrated using a contrasting climatic condition was inferior to the simulation performance that was directly calibrated from the climatic condition, compared with Figure 5(a) and 6(b), or Figure 6(a) and 5(b). For example, the NSE$_{sqrt}$ performance in Figure 6(b) is inferior to which in Figure 5(a). In the other words, for the non-dry period: $NSE_{sqrt,Figure\ 5(a)} > NSE_{sqrt,Figure\ 6(b)}$ ; for the dry period: $NSE_{sqrt,Figure\ 6(a)} > NSE_{sqrt,Figure\ 5(b)}$ .

This sentence will be modified as follows in the revised manuscript:

"However, the projection performance calibrated using a contrasting climatic condition was inferior to the simulation performance that was directly calibrated from the climatic condition, compared with Figure 5(a) and 6(b), or Figure 6(a) and 5(b). For

example, the NSE$_{sqrt}$ performance in Figure 6(b) is inferior to which in Figure 5(a)."

Hope the revision is clear to the referee and readers.

B14: P.18L375-377. This is not true for catchment 225219

**Reply:** We apologize for our mistakes. Follow the referee's comment, this sentence will be modified to specify the problem, which is as follows:

By comparing scenarios in the calibration period, it was found that scenarios 4 and 5 exhibited the highest performance in two of three catchments (405219 and 405264), followed successively by scenario 3, scenario 2, and scenario 1.

B15: P18.L379. The ranges seem not very different between scenarios 4 and 5, only slightly.

**Reply:** Thanks. This sentence will be modified as follows:

During the verification period, the median NSE$_{sqrt}$ performance in scenario 4 was 0.80% higher than scenario 5, however, the variation range in scenario 4 was 53% wider than the latter. In the DSST scheme of calibrating in the dry period and verifying in the non-dry period, scenario 3, which both considered spatial coherence of regression parameters $\beta$ and $\omega$ between different catchments, exhibited the highest median NSE$_{sqrt}$ for all catchments, had the smallest fluctuation range in two catchments (405219 and 405264) and is the second smallest scenario in catchment 22519 during the verification period.

B16: P18.L379-380. It's not very obvious that scenario 3 has a higher median performance for catchment 405264.

**Reply:** Thanks. This sentence will be modified as follows:

In catchment 405264, compared with scenarios 1, 2, 4 and 5, scenario 3 showed

an 8.1%, 6.7%, 0.2% and 0.6% improvement in the median NSE$_{sqrt}$ performance, respectively.

B17: P18.L382 This is not very obvious to me.

**Reply:** Thank you. This sentence will be modified as follows:

During the verification period, the median NSE$_{sqrt}$ performance in scenario 4 was 0.80% higher than scenario 5, however, the variation range in scenario 4 was 53% wider than the latter. In the DSST scheme of calibrating in the dry period and verifying in the non-dry period, scenario 3, which both considered spatial coherence of regression parameters $\beta$ and $\omega$ between different catchments, exhibited the highest median NSE$_{sqrt}$ for all catchments, had the smallest fluctuation range in two catchments (405219 and 405264) and is the second smallest scenario in catchment 22519 during the verification period.

B18: P18.L394. Compared –> comparing

**Reply:** Thanks. Change will be made as suggested.

B19: P20.L438. Is omega for scenario 4 not the lowest in all cases? Or do you mean the absolute values?

**Reply:** We apologize for our mistakes. This should be regression parameter $\beta$ in this place rather than $\omega$. The catchment average of the median estimates of $\beta$ in the first three scenarios are 2.78, -4.91, and 9.26 respectively, while that in the fourth scenario is much larger, reached at -39.20. Scenario 3, which considered both spatial coherence of regression parameters $\beta$ and $\omega$, has the narrowest interval of $\beta$ for all catchments, followed successively by scenario 1 (only considered the spatial coherence of the regression parameter $\beta$), scenario 2 (only parameter $\omega$ was spatially

coherent), and scenario 4 (no parameter was spatially coherent). With regards to the regression parameter $\omega$, which denotes the frequency of the sine function (in the lower figures of Figures 8 and 9), its median estimates and variation ranges in both four scenarios differ slightly. The former reached a catchment average of 0.19,0.20,0.19,0.17 for different scenarios.

B20: Figure 2. Please define all symbols and abbreviations in the figure.

**Reply:** Thanks. All symbols and abbreviations in the Figure 2 will be defined in the revised manuscript.

[Figure]

Figure 2. Schematic diagram of the GR4J rainfall-runoff model adopted from Perrin et al. (2003). In the figure, P and E refer to precipitation and evapotranspiration, respectively; $E_n$ and $P_n$ denote net precipitation and net evapotranspiration, respectively; Ps refers part of precipitation that fills the production store (i.e. S). The production store is determined as a function of the water level S in production store. The $\theta_1, \theta_2, \theta_3$, and $\theta_4$ denote model parameters. The *Perc* refers to the percolation leakage that is a function of production store *S* and parameter $\theta_1$. The Pr refers to total quantity of water that reaches the routing functions. The UH1 and UH2 denote two unit hydrographs. The $Q_1$ and $Q_9$ refer the corresponding output of the unit hydrographs, respectively; F indicates

the groundwater exchange term; R is the level in the routing store. The $Q_r$ refers to the outflow of the routing store, $Q_d$ is a function of water exchange, and Q refers to the total streamflow.

B21: Figure 5,6: I would suggest to plot the boxes for calibration and verification next to each other. It's easier to see whether there is an improvement or not. Please also add the units (also when a unitless number is presented)

**Reply:** Thanks. Changes will be made as suggested.

B22: Figure 7. Please make the labels and text bigger.

**Reply:** Thanks. Changes will be made as suggested. The modified Figure 7 will be as

follows:

[Figure]

Figure 7. BIAS performance of $Q_{median}$ for five scenarios in all catchments. The BIAS is plotted as a 10-year moving average, and 10-year moving average streamflows are plotted for reference. The left-hand three graphs are calibrated in the non-dry period and then verified in the dry period, while the opposite sequence applies to the right-hand graphs.

B23: Figure 8, 9. Maybe use the same colors for the scenarios in both plots. What are the units of beta and omega?

**Reply:** Thanks for thoughtful comment. Similar comment has been raised by Referee

**1 (see comment A1). The same color for the scenarios in both plots will be used. Both**

$\beta$ and $\omega$ are unitless. Figures 8 and 9 will be modified as violin plots in the revised manuscript (see response to A1).

**Supplement:**

**Table S1 The prior ranges of all unknown quantities in different scenarios**

**(1) Calibration in non-dry period and verification in dry period:**

Scenario 1:

| $\theta_{2-1}$ | $\theta_{2-1}$ | $\theta_{2-3}$ | $\mu_2$ | $\sigma_3$ | $\theta_{3-1}$ | $\theta_{4-1}$ | $\alpha_{1-1}$ | $\omega_{1-1}$ | $\theta_{3-2}$ | $\theta_{4-2}$ | $\alpha_{1-2}$ | $\omega_{1-2}$ | $\theta_{3-3}$ | $\theta_{4-3}$ | $\alpha_{1-3}$ | $\omega_{1-3}$ |
|---|---|---|---|---|---|---|---|---|---|---|---|---|---|---|---|---|
| -10 | -10 | -10 | -100 | 0 | 0.1 | 1 | 1 | 0.0001 | 0.1 | 0.5 | 100 | 0.0001 | 0.1 | 0.1 | 1 | 0.0001 |
| 10 | 10 | 10 | 100 | 6 | 200 | 10 | 600 | 0.4 | 300 | 20 | 1000 | 0.4 | 300 | 20 | 500 | 0.4 |

Scenario 2:

| $\theta_{2-1}$ | $\theta_{2-1}$ | $\theta_{2-3}$ | $\mu_3$ | $\sigma_3$ | $\theta_{3-1}$ | $\theta_{4-1}$ | $\alpha_{1-1}$ | $\beta_{1-1}$ | $\theta_{3-2}$ | $\theta_{4-2}$ | $\alpha_{1-2}$ | $\beta_{1-2}$ | $\theta_{3-3}$ | $\theta_{4-3}$ | $\alpha_{1-3}$ | $\beta_{1-3}$ |
|---|---|---|---|---|---|---|---|---|---|---|---|---|---|---|---|---|
| -6 | -6 | -6 | -0.4 | 0 | 1 | 0.5 | 1 | -300 | 1 | 0.1 | 100 | -300 | 0.1 | 2 | 1 | -200 |
| -6 | -6 | -6 | 0.4 | 0.1 | 500 | 10 | 600 | 300 | 300 | 20 | 600 | 500 | 400 | 20 | 800 | 300 |

Scenario 3:

| $\theta_{2-1}$ | $\theta_{2-1}$ | $\theta_{2-3}$ | $\mu_2$ | $\sigma_2$ | $\mu_3$ | $\sigma_3$ | $\theta_{3-1}$ | $\theta_{4-1}$ | $\alpha_{1-1}$ | $\theta_{3-2}$ | $\theta_{4-2}$ | $\alpha_{1-2}$ | $\theta_{3-3}$ | $\theta_{4-3}$ | $\alpha_{1-3}$ |
|---|---|---|---|---|---|---|---|---|---|---|---|---|---|---|---|
| -5 | -5 | -5 | -200 | 0 | -0 | 0 | 1 | 0.5 | 1 | 1 | 0.1 | 100 | 1 | 0.5 | 100 |
| 5 | 5 | 5 | 100 | 8 | 0.4 | 0.1 | 120 | 10 | 500 | 300 | 20 | 500 | 250 | 20 | 600 |

Scenario 4:

| $\theta_{2-1}$ | $\theta_{3-1}$ | $\theta_{4-1}$ | $\alpha_{1-1}$ | $\beta_{1-1}$ | $\omega_{1-1}$ | $\theta_{2-2}$ | $\theta_{3-2}$ | $\theta_{4-2}$ | $\alpha_{1-2}$ | $\beta_{1-2}$ | $\omega_{1-2}$ | $\theta_{2-3}$ | $\theta_{3-3}$ | $\theta_{4-3}$ | $\alpha_{1-3}$ | $\beta_{1-3}$ | $\omega_{1-3}$ |
|---|---|---|---|---|---|---|---|---|---|---|---|---|---|---|---|---|---|
| -10 | 1 | 0.1 | 1 | -300 | 0.0001 | -10 | 1 | 0.1 | 0 | -300 | 0 | -10 | 1 | 0.1 | 0 | -300 | 0.0001 |
| 10 | 500 | 10 | 800 | 300 | 0.4 | 10 | 500 | 10 | 800 | 300 | 0.4 | 10 | 500 | 10 | 800 | 300 | 0.4 |

**(2) Calibration in dry period and verification in dry period:**

Scenario 1**:**

| $\theta_{2-1}$ | $\theta_{2-2}$ | $\theta_{2-3}$ | $\mu_2$ | $\sigma_2$ | $\theta_{3-1}$ | $\theta_{4-1}$ | $\alpha_{1-1}$ | $\omega_{1-1}$ | $\theta_{3-2}$ | $\theta_{4-2}$ | $\alpha_{1-2}$ | $\omega_{1-2}$ | $\theta_{3-3}$ | $\theta_{4-3}$ | $\alpha_{1-3}$ | $\omega_{1-3}$ |
|---|---|---|---|---|---|---|---|---|---|---|---|---|---|---|---|---|
| -10 | -10 | -10 | -60 | 0 | 1 | 0.5 | 1 | 0 | 1 | 0.5 | 1 | 0 | 1 | 0.1 | 1 | 0 |
| 10 | 10 | 10 | 60 | 6 | 300 | 10 | 600 | 0.4 | 300 | 20 | 600 | 0.4 | 300 | 15 | 600 | 0.4 |

Scenario 2:

| $\theta_{2-1}$ | $\theta_{2-2}$ | $\theta_{2-3}$ | $\mu_3$ | $\sigma_3$ | $\theta_{3-1}$ | $\theta_{4-1}$ | $\alpha_{1-1}$ | $\beta_{1-1}$ | $\theta_{3-2}$ | $\theta_{4-2}$ | $\alpha_{1-2}$ | $\beta_{1-2}$ | $\theta_{3-3}$ | $\theta_{4-3}$ | $\alpha_{1-3}$ | $\beta_{1-3}$ |
|---|---|---|---|---|---|---|---|---|---|---|---|---|---|---|---|---|
| -10 | -10 | -10 | 0.0001 | 0 | 1 | 0.5 | 1 | -300 | 1 | 0.1 | 1 | -400 | 0.1 | 0.5 | 1 | -400 |
| 10 | 10 | 10 | 0.4 | 0.1 | 200 | 15 | 500 | 400 | 300 | 20 | 600 | 500 | 140 | 20 | 600 | 400 |

Scenario 3:

| $\theta_{2-1}$ | $\theta_{2-2}$ | $\theta_{2-3}$ | $\mu_2$ | $\sigma_2$ | $\mu_3$ | $\sigma_3$ | $\theta_{3-1}$ | $\theta_{4-1}$ | $\alpha_{1-1}$ | $\theta_{3-2}$ | $\theta_{4-2}$ | $\alpha_{1-2}$ | $\theta_{3-3}$ | $\theta_{4-3}$ | $\alpha_{1-3}$ |
|---|---|---|---|---|---|---|---|---|---|---|---|---|---|---|---|
| -10 | -10 | -10 | -80 | 0 | 0 | 0 | 1 | 0.5 | 1 | 1 | 0.1 | 1 | 1 | 0.1 | 1 |
| 10 | 10 | 10 | 80 | 6 | 0 | 0.1 | 200 | 10 | 500 | 400 | 20 | 600 | 400 | 20 | 600 |

Scenario 4:

| $\theta_{2-1}$ | $\theta_{3-1}$ | $\theta_{4-1}$ | $\alpha_{1-1}$ | $\beta_{1-1}$ | $\omega_{1-1}$ | $\theta_{2-2}$ | $\theta_{3-2}$ | $\theta_{4-2}$ | $\alpha_{1-2}$ | $\beta_{1-2}$ | $\omega_{1-2}$ | $\theta_{2-3}$ | $\theta_{3-3}$ | $\theta_{4-3}$ | $\alpha_{1-3}$ | $\beta_{1-3}$ | $\omega_{1-3}$ |
|---|---|---|---|---|---|---|---|---|---|---|---|---|---|---|---|---|---|
| -10 | 1 | 0.1 | 1 | -300 | 0.0001 | -10 | 1 | 0.1 | 1 | -300 | 0 | -10 | 1 | 0.1 | 1 | -300 | 0 |
| 10 | 500 | 10 | 800 | 300 | 0.4 | 10 | 500 | 10 | 800 | 300 | 0.4 | 10 | 500 | 10 | 800 | 300 | 0.4 |

Notes:

$\theta_{2-1}$, $\theta_{2-2}$ and $\theta_{2-3}$ refers to model parameter $\theta_2$ in catchment 225219, 405219 and 405264, respectively; $\theta_{3-1}$, $\theta_{3-2}$ and $\theta_{3-3}$ refer to model parameter $\theta_3$ in catchment 225219, 405219 and 405264, respectively; $\theta_{4-1}$, $\theta_{4-2}$ and $\theta_{4-3}$ refers to model parameter $\theta_4$ in catchment 225219, 405219 and 405264, respectively; $\mu_2$, $\sigma_2$, $\mu_3$ and $\sigma_3$ represent four hyper-parameters; $\alpha_{1-1}$, $\alpha_{1-2}$ and $\alpha_{1-3}$ refer to regression parameter $\alpha$ in catchment 225219, 405219 and

26     405264, respectively; $\beta_{1\text{-}1}$, $\beta_{1\text{-}2}$ and $\beta_{1\text{-}3}$ refer to regression parameter $\beta$ in catchment 225219, 405219 and 405264, respectively; $\omega_{1\text{-}1}$, $\omega_{1\text{-}2}$ and $\omega_{1\text{-}3}$

27     refer to regression parameter $\omega$ in catchment 225219, 405219 and 405264, respectively.

28

29

30

31     $$\varepsilon_c\left[\theta_1,\theta_2,\theta_3,\theta_4\right] = -RMSE\left[\sqrt{Q}\right]\left(1+\left|1+BIAS\right|\right)$$

32

$$\textit{Scenario 1}: \Lambda=\prod_{c=1}^{C}\varepsilon_c\left[\theta_1(t,c),\theta_2(c),\theta_3(c),\theta_4(c)\middle|\alpha(c),\beta,\omega(c)\right]\bullet f_N\left(\beta\middle|\mu_2,\sigma_2\right)$$

$$\textit{Scenario 2}: \Lambda=\prod_{c=1}^{C}\varepsilon_c\left[\theta_1(t,c),\theta_2(c),\theta_3(c),\theta_4(c)\middle|\alpha(c),\beta(c),\omega\right]\bullet f_N\left(\omega\middle|\mu_3,\sigma_3\right)$$

33     $$\textit{Scenario 3}: \Lambda=\prod_{c=1}^{C}\varepsilon_c\left[\theta_1(t,c),\theta_2(c),\theta_3(c),\theta_4(c)\middle|\alpha(c),\beta,\omega\right]\bullet\prod_{n=1}^{2} f_N\left(\beta,\omega\middle|\mu_2,\sigma_2,\mu_3,\sigma_3\right)$$

$$\textit{Scenario 4}: \Lambda=\prod_{c=1}^{C}\varepsilon_c\left[\theta_1(t,c),\theta_2(c),\theta_3(c),\theta_4(c)\right]$$

$$\textit{Scenario 5}: \Lambda=\prod_{c=1}^{C}\varepsilon_c\left[\theta_1(c),\theta_2(c),\theta_3(c),\theta_4(c)\right]$$

---

## Author Response (AR1)

**Responses to all the Referees:**

We sincerely appreciate the comments and advices from the Editor and Referees. Detailed responses to the comments from the Editor and Referees are presented below. In the following Responses, comments by Referee #1 are labeled A, and comments by Referee #2 are labeled B. For example, A1 represents the first comment made by Referee #1 and B1 represents the first comment made by Referee #2. Please also note that the page and line numbers mentioned in reviewers comments refer to the original version, while in the authors' response they refer to the revised version.

**Responses to Editor:**

The paper is potentially valuable, however, as the reviewers pointed out, there are several points of concern. One of the aspects that did not convince me is the combination of heuristic and Bayesian objective functions in Equation 5.

I suggest the author to take into serious consideration the various comments before resubmitting their paper.

**Reply:** We sincerely appreciate the comments from the Editor. Great efforts have been made to address the editor and reviewers' comments. The equation.5 has been modified in the revised manuscript, and more information has been provided in the revised manuscript. Please refer to lines 281-291, pages 14-15.

**Responses to Referee #1:**

This paper analyzes the prediction performance of a lumped hydrological model using different time and spatial dependent parametrizations of one of its parameters. There are several errors in the paper and points that should be explained better and I have a major concern regarding the results.

Comment on the results:

A1: The value of omega looks strange to me. Assuming that the equation 1 you wrote is correct (and therefore it is a frequency and not a phase) and that the order of magnitude of omega is of hundreds (like shown in figures 8 and 9), this mean that your parameter theta1 oscillates hundreds of times per time step. This looks unreal to me since the goal of having time-variant parameters is to represent long term (seasonal) oscillations. Therefore, either there is a problem with the unit of omega or your model is not doing what it was meant for. If omega is a phase (meaning theta1 = alpha + beta*sin(t + omega)) the value of omega makes more sense but theta1 would still complete an oscillations every 6.28 time steps (the time step is days, right?). Don't you also have a frequency that multiplies "t" and have a small value?

**Reply:** We deeply apologize for our mistakes.

(1) $\omega$ represents frequency rather than phase. It has been revised accordingly in the revised manuscript. Please refer to line 222 in page 11 and line 520 in page 26.

(2) We have carefully checked the data for Figures 8 and 9, and found that we misused the results of regression parameter $\alpha$ to plot for $\omega$. We are so sorry for our mistakes. As is stated in the response to comment A18, the Figures 8 and 9

have been modified as violin plots with the right data. Figures are as follows:

[Figure]

Figure 8. Posterior distributions of the regression parameters (β and ω) for the production storage capacity ($\theta_1$) for the four modeling scenarios in all the 3 studied catchments. In this figure, parameters were calibrated in the non-dry period while verified in the dry period. The solid horizontal lines within the violin plots denote the 25[th] and 75[th] percentiles of the posterior distribution, while the dash line denotes median estimates.

[Figure]

Figure 9. Posterior distributions of the regression parameters (β and ω) for the production storage capacity ($\theta_1$) for the four model scenarios in all 3 studied catchments. In this figure, parameters were calibrated in the dry period while verified in the non-dry period. The solid horizontal lines within the violin plots denote the 25[th] and 75[th] percentiles of the posterior distribution, while the dash line denotes median estimates.

As shown in Figure 8, the catchment averages of regression parameter $\omega$ for

different scenarios are 0.24, 0.14, 0.15, and 0.18, while those in Figure 9 are 0.15, 0.26, 0.23, and 0.17 respectively. The phase $T$ of the sine term could be derived based on the estimates of $\omega$ based on equation $T = 2\pi/\omega$. Thus, the mean phases $T$ of model parameter $\theta_1$ for different scenarios are 26.2, 46.3, 41.9 and 35.2 in Figure 8, respectively. Similarly, the mean phases $T$ are 42.9, 24.1, 27.4 and 38.0 in Figure 9, respectively. Please refer to lines 521-527 in pages 26-27.

**Detailed comments:**

A2: line 102-103: There is not a clear definition of pooling, complete pooling and hierarchical Bayesian. I would explain shortly what do they mean and which are the differences since then the paper only writes about hierarchical Bayesian.

**Reply:** Thank you for your comments. The following explanations about the no pooling, complete pooling and hierarchical Bayesian framework have been added in the revised manuscript:

In general, there are three methods to consider the spatial coherence between different catchments in parameter estimation. The first one is no pooling, which means every catchment is modeled independently, and all parameters are catchment-specific. The second one is complete pooling, which means all parameters are considered to be common across all catchments. The third/last one is hierarchical Bayesian (HB) framework, also known as partial pooling, which means some parameters are allowed to vary by catchments and some parameters are assumed to be drawn from a common hyper-distribution across the region that consists of different catchments. Please refer to lines 99-107 in page 5.

A3: line 152-153: It would be beneficial to explain shortly how the method works even if it was already used in other studies.

**Reply:** Thank you for your comment. Definition of the dry period is explained as follows and has been added in the revised manuscript:

Saft et al. (2015) tested several algorithms for dry period delineation, which considered different combinations of dry run length, dry run anomaly and various boundary criteria, and found that the identification results of dry period by one of the algorithms showed marginal dependence on the algorithm and the main results were robust to different algorithms. The detailed processes could be found on Saft et al. (2015) and are also presented as follows.

Firstly, the annual rainfall data were calculated relative to the annual mean, and the anomaly series was divided by the mean annual rainfall and smoothed with a 3 year moving window. Secondly, the first year of the drought remained the start of the first 3 year negative anomaly period. Thirdly, the exact end date of the dry period was determined through analysis of the unsmoothed anomaly data from the last negative 3 year anomaly. The end year was identified as the last year of this 3 year period unless: (i) there was a year with a positive anomaly >15% of the mean, in which case the end year is set to the year prior to that year; or (ii) if the last two years have slightly positive anomalies (but each <15% of the mean), in which case the end year is set to

the first year of positive anomaly; (iii) to ensure that the dry periods are sufficiently long and severe, in the subsequent analysis, the authors use dry periods with the following characteristics: length $\geq 7$ years; mean dry period anomaly<25%.

Please refer to lines 159-176 in pages 8-9 in the revised manuscript.

A4: line 159: Maybe it is more appropriate to use "cross validation" instead. I suggest to avoid making a paragraph with just one sentence and remove paragraphs 2.1.1 and 2.1.2 putting all together in section 2.1.

**Reply:** Thanks. Following the Referee's suggestion, paragraphs 2.1.1 and 2.1.2 have been put together within section 2.1, and the sub-titles of sections 2.1.1 and 2.1.2 have been deleted in the revised manuscript. Please refer to lines 150-183 in pages 8-9 in the revised manuscript.

A5: chapter 2.3: It is not clear to me what do you do with the other parameters of the GR4J model (theta2, theta3, theta4). Do you keep them fixed or do you sample them? What is their effect on the final result?

**Reply:** Thank you for your comments.

(1) All other model parameters ($\theta_2, \theta_3$, and $\theta_4$, except $\theta_1$) are not fixed, but sampled simultaneously with regression parameters $\alpha$, $\beta$ and $\omega$ (if present), and hyper-parameters $\mu_2$, $\sigma_2$, $\mu_3$ and $\sigma_3$ in the SCEM-UA algorithm. In real calculation process, we would set a large variation interval for each unknown quantity first, the estimations of parameters would converge to a smaller interval in MCMC calculation process. Then we checked the model convergence using the Gelman-Rubin convergence value by evolving three parallel chains with 30000 random samples and confirmed that the convergence value was smaller than the threshold 1.2 (Gelman et al., 2013).

Please refer to lines 285-291 and lines 295-298 in page 15 in the revised manuscript.

(2) Previous studies on GR4J model showed that $\theta_2, \theta_3$, and $\theta_4$ are less sensitive than $\theta_1$ under changing climate (Perrin et al., 2003; Renard et al., 2011; Westra et al., 2014). Therefore, we think that it is reasonable to assume that $\theta_1$ is time-varying while other model parameters are temporally invariant. Please refer to lines 206-217, pages 10-11.

A6: line 199: The equation is different from the ones reported in Table 1.

**Reply:** We apologize for our mistakes. The fault equation in Table 1 has been revised as equation 1 in the revised manuscript. Please refer to Table 1 in the revised manuscript.

A7: line 201: You write that omega is the phase while in the equation 1 it is a frequency.

**Reply:** Thank you for pointing out this mistake. The $\omega$ represents the frequency rather than the phase (see response to comment A1). The statement in line 201 in the original manuscript (see line 222, page 11 in the revised manuscript) is wrong and has been modified in the revised manuscript. Please refer to line 222, page 11.

A8: line 202: The combination alpha=beta=omega=0 makes theta 1 to be equal to 0, that indeed it is a constant value but probably it is not what you want.

**Reply:** Thanks. According to the definition of the GR4J model (Perrin et al., 2003), $\theta_1$ represents the primary storage of water in the catchment and must be a positive value. Thus, in order to satisfy this requirement, the combination of $\alpha=\beta=\omega=0$ would be excluded under calculation, and other combinations that made $\theta_1$ equal to zero would be excluded too.

A9: chapter 2.3.2: What happens to alpha? You don't write about it anymore in the rest of the paper. Do you keep it fixed or do you sample also it? What is its effect on the final result?

**Reply:** Thanks.

(1) The regression parameter $\alpha$ represents the constant term in equation 1. Changes in $\alpha$ lead to consistent changes in $\theta_1$ across the whole time series, which doesn't result in temporal variations of model parameter.

(2) Regression parameter $\alpha$ is not fixed in advance but is sampled as same as other unknown quantities. The posterior distribution of $\alpha$ is derived out simultaneously with hyper-parameters $\mu_2$, $\mu_3$, $\sigma_2$ and $\sigma_3$, other regression parameters $\beta$ and $\omega$ (if present), and model parameters $\theta_2$, $\theta_3$ and $\theta_4$ in the SCEM-UA algorithm. Please refer to lines 295-298, page 15.

A10: chapter 2.3.2: It is not clear to me if linking the parameters between catchments means sampling them from the same Gaussian distribution or there is another form of linking.

**Reply:** We are sorry for not making this part clear enough. The link is that regression parameter $\beta$ (or $\omega$) of different catchments is assumed to sample their values in the same Gaussian distribution. This kind of links has been widely used in the field of extreme event analysis, such as Sun et al. (2015, 2016), Lima et al. (2009) and Bracken et al. (2018).

A11: chapter 2.3.2: How do you sample omega and beta when they are not linked?

**Reply:** Thanks. $\omega$ is not linked in scenario 1, while $\beta$ is not linked in scenario 2.

In scenario 4, both $\omega$ and $\beta$ are not linked. Spatially irrelevant parameters of different catchments would be sampled and derived as independent variables. For example, In scenario 1, regression parameter $\beta$ is spatially linked, i.e., $\beta(c) = N(\mu_2, \sigma_2^2)$, which means that the estimates of $\beta$ are shared by all catchments. Meanwhile, independent regression parameters $\omega_{1-1}$, $\omega_{1-2}$, and $\omega_{1-3}$ are used as independent variables to represent the frequency of model parameter $\theta_1$ in different catchments. The name of all unknown quantities and their prior ranges in different scenarios could be found in the supplementary material.
Please refer to lines 254-264, page 13.

A12: line 218: How do you choose the values of mu and sigma, the hyper-parameters of your model?
**Reply:** Thanks. The posterior distributions of all unknown quantities, including model parameters $\theta_2$, $\theta_3$ and $\theta_4$, and regression parameters $\alpha$, $\beta$ and $\omega$, and hyper-parameters $\mu_2$, $\mu_3$, $\sigma_2$ and $\sigma_3$ are sampled and derived simultaneously through the SCEM-UA algorithm. In actual calculation process, we would set a large variation interval for each unknown quantity first, parameters would converge to a small interval in MCMC calculation process. Then we checked the model convergence using the Gelman-Rubin convergence value by evolving three parallel chains with 30000 random samples and confirmed that the convergence value was smaller than the threshold 1.2 (Gelman et al., 2013). Please refer to lines 295-298 in page 16 and lines 306-312 in page 16.

A13: chapter 2.4.1: I wouldn't call "likelihood function" what actually is an objective function.
**Reply:** Thanks. As suggested, the "likelihood function" has been modified as "objective function" in the revised manuscript. Please refer to lines 268, page 14.

A14: line 250: You are mixing an objective function with a prior distribution of the parameters. How do you account for the prior distribution of the parameters when they are not linked?
**Reply:** Thanks. The uniform distribution is used as the prior distribution for all unknown quantities.
(1) The objective function of Eq.1 has been modified as follows:

$$\varepsilon_c\left[\theta_1, \theta_2, \theta_3, \theta_4\right] = -RMSE\left[\sqrt{Q}\right]\left(1 + |1 + BIAS|\right) \tag{1}$$

where

$$RMSE\left[\sqrt{Q}\right] = \sqrt{\frac{1}{T}\sum_{t=1}^{T}\left[Q_{sim}(t) - Q_{obs}(t)\right]^2} \tag{2}$$

and $RMSE\left[\sqrt{Q}\right]$ refers to the root-mean-square error, in which $Q_{sim}$ is derived by the adopted hydrological model.

(2) The objective function of Eq.5 has been modified as follows:

$$\text{Scenario 1: } \Lambda = \prod_{c=1}^{C} \varepsilon_c\left[\theta_1(t,c),\theta_2(c),\theta_3(c),\theta_4(c)\big|\alpha(c),\beta,\omega(c)\right] \bullet f_N\left(\beta\big|\mu_2,\sigma_2\right)$$

$$\text{Scenario 2: } \Lambda = \prod_{c=1}^{C} \varepsilon_c\left[\theta_1(t,c),\theta_2(c),\theta_3(c),\theta_4(c)\big|\alpha(c),\beta(c),\omega\right] \bullet f_N\left(\omega\big|\mu_3,\sigma_3\right)$$

$$\text{Scenario 3: } \Lambda = \prod_{c=1}^{C} \varepsilon_c\left[\theta_1(t,c),\theta_2(c),\theta_3(c),\theta_4(c)\big|\alpha(c),\beta,\omega\right] \bullet \prod_{n=1}^{2} f_N\left(\beta,\omega\big|\mu_2,\sigma_2,\mu_3,\sigma_3\right)$$

$$\text{Scenario 4: } \Lambda = \prod_{c=1}^{C} \varepsilon_c\left[\theta_1(t,c),\theta_2(c),\theta_3(c),\theta_4(c)\right]$$

$$\text{Scenario 5: } \Lambda = \prod_{c=1}^{C} \varepsilon_c\left[\theta_1(c),\theta_2(c),\theta_3(c),\theta_4(c)\right] \tag{5}$$

where the number of catchments in the region is represented by C; $c$ represents the specific catchment; the $t$ is the time step.
Please refer to lines 281-291, pages 14-15.

A15: chapter 2.4.2: You don't say which settings of the sampling method you use (e.g. how many parameters you sample. . .)
**Reply:** Thanks. The sampling method used in this paper is the SCEM-UA algorithm. The detailed descriptions of the settings of SCEM-UA algorithm have been added in the revised manuscript:
(1) Convergence is assessed by evolving three parallel chains with 30000 random samples, the posterior distributions of parameters are evaluated by the Gelman-Rubin convergence value and are confirmed that the convergence value is smaller than the threshold 1.2 (Gelman et al., 2013). Please refer to lines 306-312, page 16.
(2) The number of unknown quantities in different scenarios is as follows: fifteen in scenarios 1 and 2, thirteen in scenario 3 and eighteen in scenario 4. Please refer to lines 261-262, page 13.

A16: chapter 3.2.1: The dataset that you get is unbalanced, since there are more wet years. Is it taken into account? Does it have an effect on the calibration?
**Reply:** Thank you for pointing out this situation. Generally, a longer time series may improve the robustness of hydrological predictions. However, we tested the calibration performance with different lengths of records (> 6 years) in dry (non-dry) period and found that their results are almost the same. Therefore, we used the whole time series of the dry (15 years) and non-dry (10 year) periods into model calibration.

A17: chapter 3.2.3: Figures 7 and 8 are actually 8 and 9.
**Reply:** Thanks. Changes have been made.

A18: Figures 5, 6, 8, 9: Since you want to show a probability distribution I wouldn't use a boxplot but, instead, I suggest to use a violin plot (e.g.https://seaborn.pydata.org/examples/grouped_violinplots.html)
**Reply:** Thank you for your suggestions.
(1) As suggested, Figures 8 and 9 have been modified as violin plot in the revised manuscript, which also could be found in response to comment A1 by Referee #1
(2) Figures 5 and 6 have been revised as violin plot in the revised manuscript.

A19: Figures 8, 9: Why do you change the colors between beta and omega? This makes the plot more difficult to read.
**Reply:** Thanks. Changes have been made as suggested.

**General Comments**

The study of Pan et al. tests a Hierarchical Bayesian framework to incorporate time and spatial variability in model parameters. Specifically, the method was tested for the GR4J-model in three Australian catchments. Four modelling scenarios were tested, and one base scenario was formulated. The study shows that including spatially and temporally variable parameters improves model performance and reduces uncertainty. The article shows interesting work, which could be a nice contribution to the field. Generally, the article needs some more explanations on the method, but there is also some incomplete reasoning. Hence, there are several issues I'd like to address.

**Specific comments**

B1: Key of the article is the hierarchical framework, but the authors may want to work on the explanation of the method. It is especially not clear to me how the hyper-parameters are determined, and how the catchment-specific values follow from that. Are the hyperparameters estimated in SCEM-UA? Or are these pre-defined? The gaussian distributions are defined by the authors as prior distributions, and that makes me assume that the model parameter theta is determined in SCEM-UA starting from this prior distribution, whereas the remaining model parameters are either kept fixed or sampled from a uniform distribution and independently for each catchment. Is that correct? Because if that is the case, the hyper-parameters (and hence the distribution) are determined in advance, so what are these based on? Besides, the choice of a gaussian distribution may seem a logical first guess, but it remains an arbitrary choice. So what is the reasoning behind this choice? In addition, the choice of the prior distribution may lead to some circular reasoning. When spatial coherence is used, the variation in performance goes down, but is this not just an artefact of the pre-defined gaussian distribution? In other words, if the prior distribution is set narrower, the resulting posterior distribution will probably be narrower as well. I believe it is therefore crucial to report also the prior ranges (or fixed values) for especially the (time-invariant) theta-parameter, but also all other model parameters.

**Reply:** Thank you for your comment. Since that several sub-comments have been included in comment B1, for clarification, a point by point response to these sub-comments is made as follows. For example, B1S1 refers to the first sub-comment in B1.

B1S1: Key of the article is the hierarchical framework, but the authors may want to work on the explanation of the method. It is especially not clear to me how the hyper-parameters are determined, and how the catchment-specific values follow from that. Are the hyperparameters estimated in SCEM-UA? Or are these pre-defined?

**Reply:** Thanks for the comments and sorry that we failed to describe it clear enough in the original submission. It is explained here and similar clarification has been made in lines 295-298 in page 15 and lines 306-312 in page 16 in the revised manuscript, and supplementary material.

(1) All the hyper-parameters are not determined in advance. The

hyper-parameters are sampled and determined simultaneously with other unknown quantities in the SCEM-UA algorithm. Actually, all other model parameters ($\theta_2$, $\theta_3$, and $\theta_4$, except $\theta_1$) are sampled simultaneously with regression parameters ($\alpha$, $\beta$ and $\omega$ (if present)) and hyper-parameters ($\mu_2$, $\sigma_2$, $\mu_3$ and $\sigma_3$) in the algorithm. Convergence is assessed by evolving three parallel chains with 30000 random samples, the posterior distributions of parameters are evaluated by the Gelman-Rubin convergence value and are confirmed that the convergence value is smaller than the threshold 1.2 (Gelman et al., 2013).

(2) Regression parameter $\omega$ is not linked in scenario 1, while $\beta$ is not linked in scenario 2. In scenario 4, both $\omega$ and $\beta$ are not linked. It should be noted that the spatially irrelevant parameters would be sampled and derived as independent variables. For example, in scenario 1, regression parameter $\beta(c) = N(\mu_3, \sigma^2)$, which means that $\beta$ is shared by linked catchments, while regression parameters $\omega_{1-1}$, $\omega_{1-2}$, and $\omega_{1-3}$ are used as independent variables to represent the frequency of model parameter $\theta_1$ in different catchments. The names of all unknown quantities in different scenarios have been added in the supplementary material.

B1S2: The gaussian distributions are defined by the authors as prior distributions, and that makes me assume that the model parameter theta is determined in SCEM-UA starting from this prior distribution, whereas the remaining model parameters are either kept fixed or sampled from a uniform distribution and independently for each catchment. Is that correct? Because if that is the case, the hyper-parameters (and hence the distribution) are determined in advance, so what are these based on? Besides, the choice of a gaussian distribution may seem a logical first guess, but it remains an arbitrary choice. So what is the reasoning behind this choice? In addition, the choice of the prior distribution may lead to some circular reasoning. When spatial coherence is used, the variation in performance goes down, but is this not just an artefact of the pre-defined gaussian distribution? In other words, if the prior distribution is set narrower, the resulting posterior distribution will probably be narrower as well. I believe it is therefore crucial to report also the prior ranges (or fixed values) for especially the (time-invariant) theta-parameter, but also all other model parameters.

**Reply:** Thank you. It is explained here and similar clarification has been made in the revised manuscript.

(1) The Gaussian distribution is one of the widely used distributions for describing the prior layer within the HB framework and has been applied in many previous studies, such as Sun et al (2015, 2016) and Chen et al (2014). The choice of the distribution is not our key point, so we just adopt a typical one from historical

(2) Only two things were fixed in advance: the structure of Gaussian distribution and the variation ranges of all unknown quantities. The posterior distributions of all unknown quantities would be derived simultaneously in the SCEM-UA algorithm. In addition, as illustrated in response to comment A14 by Referee #1, convergence is assessed by evolving three parallel chains with 30000 random samples. The posterior distributions of parameters are evaluated by the Gelman-Rubin convergence value and are confirmed that the convergence value is smaller than the threshold 1.2 (Gelman et al., 2013). Please refer to

(3) The prior ranges of all unknown quantities in different scenarios have been added in the supplementary material.

B2: I also wonder how valid it is to assume the catchments are similar. The authors state on p15.L314, that the catchments satisfy the homogeneity assumption. What is this assumption and how do they satisfy this assumption? A clear description of the catchments may be needed to defend that the catchments are the same. Just looking at the DEM and the annual values of rainfall and runoff (Table 2) give me the idea that the Big catchment (405264) behaves fundamentally different compared to the other two. This catchment also reached much higher performances in calibration (Fig.5 and 6) when no spatial coherence is used, and also shows different results in the BIAS comparison (Figure 7).

Sometimes, the conclusions and statements of the authors do not seem to be strongly supported by the data as shown. The boxplots with performances (Figures 5, 6) show relatively similar performances, and, to be honest, a clear pattern is not very obvious. In addition, the authors tend to generalize in some cases findings that mainly apply to just two of the three catchments (see also my minor comments). I believe additional analyses may be needed to support the conclusions more, for example a statistical test to check if the distributions are significantly different. Or the addition of other, multiple performance measures, to assess the performance over multiple aspects (high flows, low flows etc.). Further, all beta-values plot around zero in Figure 8, basically pointing at the absence of a clear trend. Is this indeed true? It would be interesting to show the timeseries of the parameter. The absence of a trend may explain the similar performances for all scenarios, and especially also why the time-varying scenarios do not outperform the others clearly. Besides, when beta is around zero, there is no point of looking at omega, as this does not do much in that case.

Concluding, the authors may need to clarify more what they did and how they arrive at several conclusions. I hope the authors find my comments useful, and I look forward to a revised manuscript.

**Reply:** Thank you for your insightful comments. Since that several sub-comments have been included in comment B2, for clarification, a point by point response to these sub-comments is made as follows.

B2S1: I also wonder how valid it is to assume the catchments are similar. The authors

state on p15.L314, that the catchments satisfy the homogeneity assumption. What is this assumption and how do they satisfy this assumption? A clear description of the catchments may be needed to defend that the catchments are the same. Just looking at the DEM and the annual values of rainfall and runoff (Table 2) give me the idea that the Big catchment (405264) behaves fundamentally different compared to the other two. This catchment also reached much higher performances in calibration (Fig.5 and 6) when no spatial coherence is used, and also shows different results in the BIAS comparison (Figure 7).

**Reply:** We apologize for our mistakes. The homogeneity assumption has be deleted in the revised manuscript, because it is the spatial coherence of adjacent catchments that has been used as effective information to restrict the prediction uncertainties, so the homogeneity assumption of different catchments is not necessary. The studied catchments do have several similar characteristics: i) the average slope is similar, that is, catchment 225219 is 12.8, catchment 405219 is 10.7 and catchment 405264 is 9.7; ii) as shown in Table 2, these catchments have similar climatic conditions including mean annual potential evapotranspiration and rainfall patterns; iii) these catchments have experienced the same prolonged drought, and have similar amplitude of variation of rainfall and runoff between non-dry and dry periods.

B2S2: Sometimes, the conclusions and statements of the authors do not seem to be strongly supported by the data as shown. The boxplots with performances (Figures 5, 6) show relatively similar performances, and, to be honest, a clear pattern is not very obvious.

**Reply:** We agree with the Referee that Figures 5 and 6 showed similar pattern in terms of the ranked orders of NSE amongst four scenarios. In addition, we have made point to point responses to all the ambiguous sentences with objective descriptions in the following technical comments raised by Referee #2.

B2S3: In addition, the authors tend to generalize in some cases findings that mainly apply to just two of the three catchments (see also my minor comments). I believe additional analyses may be needed to support the conclusions more, for example a statistical test to check if the distributions are significantly different. Or the addition of other, multiple performance measures, to assess the performance over multiple aspects (high flows, low flows etc.).

**Reply:** Thank you for your helpful comment. More information about the additional performance measures have been added in the revised manuscript.

(1) Firstly, two performance measures based on high flows (i.e., mean annual maximum flow) and low flows (i.e., mean annual minimum flow) have been used to evaluate the high and low flows. Please refer to lines 340-344, page 18.

(2) Secondly, the results of these measures have been be added as Tables 6 and 7 in the revised manuscript.

(3) Thirdly, discussions of the results of these measures have been added in lines 465-484, pages 24-25 in the revised manuscript.

B2S4: Further, all beta-values plot around zero in Figure 8, basically pointing at the absence of a clear trend. Is this indeed true? It would be interesting to show the timeseries of the parameter. The absence of a trend may explain the similar performances for all scenarios, and especially also why the time-varying scenarios do not outperform the others clearly. Besides, when beta is around zero, there is no point of looking at omega, as this does not do much in that case.

Concluding, the authors may need to clarify more what they did and how they arrive at several conclusions. I hope the authors find my comments useful, and I look forward to a revised manuscript.

**Reply:** We apologize for the mistakes in Figures 8 and 9.

(1) As discussed in response to comment A1 by Referee #1, Figures 8 and 9 have been redrawn.

(2) As discussed in the section 4.2, model parameter $\theta_1$ is time-varying and no spatial coherence is considered ($\beta \neq 0$) in scenario 4, while $\theta_1$ is stationary and of course no spatial coherence is included in scenario 5 ($\beta = 0$). Scenario 4, had a higher median NSE$_{sqrt}$ performance than that of scenario 5 in five of six options (except catchment 405219 in the first DSST scheme), which indicates the validity of the time-varying scheme for improving the model performance. Compared with scenario 5, the introduction of additional regression parameters ($\alpha, \beta$ and $\omega$) in scenario 4 at the same time amplified the model projection uncertainty in two of three catchments (225219 and 405264). However, the appropriate adoption of spatial coherence alleviates this problem. Scenario 3, which considered both spatial coherence of regression parameters $\beta$ and $\omega$ between different catchments, exhibited the optimal median NSE$_{sqrt}$, DIC, and MaxF estimates in most options during the verification period, which illustrated the validity of the inclusion of the spatial coherence of regression parameters $\beta$ and $\omega$. Please refer to lines 389-454, pages 21-23.

(3) The median estimates of $\beta$ in Figures 8 and 9 are not equal to zero. Because the facts that the adopted hydrological model is on a daily scale and the assumed time-varying model parameter $\theta_1$ would change its values in each time step, the regression parameter $\beta$, as the amplitude of the sine term, is supposed to have a small absolute value rather a large one. As shown in Figure 8, the catchment average of the median estimate of $\beta$ is 2.78 in scenario 1, -4.91 in scenario 2, 9.26 in scenario 3, and -39.20 in scenario 4 in the scheme of calibrating in the non-dry period

and verifying in the dry period. Please refer to lines 506-527, pages 26- 27.

(4) After calculation, parameter $\beta$ has a deterministic posterior distribution (as shown in Figures 8 and 9) rather than a time-varying one.

**Technical corrections**

B3: P.7. section 2.1.1. Please elaborate on how the dry periods are defined.
**Reply:** Thank you. Please refer to response of comment A3 about the detailed definition of the dry period, which has been added in the revised manuscript.

B4: P8. Section 2.1.2. Why add this paragraph when you only refer to section 2.5?
**Reply:** This paragraph has been modified. Please refer to lines 180-183, page 9.

B5: P10. L210 Do you mean Eq. 1?
**Reply:** We are sorry for this oversight. The phase "Eq.2" has been revised as "Eq.1" in the revised manuscript.

B6: P10.L210 ...expected to the same. . . ! expected to be the same
**Reply:** Thanks. Change has been made as suggested.

B7: P12. L50. Please define N and n
**Reply:** Thanks. *N* refers to the Gaussian distribution and *n* represents the number of regression parameters that are spatially coherent. The definitions of *N* and *n* have been added in lines 287-288, page 14.

B8: P12.L258. Which parameters are optimized in SCEM-UA?
**Reply:** Thanks. All unknown quantities of different scenarios that needed to be optimized in SCEM-UA have been added in the supplementary material.

B9: P15.L326. Please explain how I can see this from Figure 4, except for the pre-defined red colour. Is this where the black line crosses the axis? Why are the first years not considered?
**Reply:** Thanks. The bars in blue and red colors in Figure 4 represent annual rainfall anomalies during the non-dry and dry periods, respectively. The black line is annual anomaly of rainfall smoothed with the 3-year moving window.

The start of the dry period is defined as the start of first 3-year consecutive negative anomaly period based on Saft et al (2015). According to the definition of dry period, the start of the dry period is not the place where the black line crosses the axis. Because the years near the cross point have positive rainfall anomaly. Similar comment is also raised by the Referee #1 (see comment A3 and our corresponding response). In the revised manuscript, the definition of the dry periods has been added. Please refer to lines 159-176, pages 8-9.

B10: P16.L339-340. Are these references in the right place? You describe your own

results, shouldn't you refer to one of the figures?

**Reply:** Thank you for your comment. These references have been deleted, and the sentence has been modified as follows:

As shown in Figures 5(a), 6(a) and 7, the calibrated model parameters yielded good simulation performance over the calibrated periods for all criteria. Please refer to lines 389-390, page 20.

B11: P17.L355-357. This is, as far as I can see, not true for all catchments. Catchments 225219 and 405264 have a higher median, but the variation is less for 225219.

**Reply:** We apologize for this mistake. In figure 5(b), the variation of NSE$_{sqrt}$ in scenario 4 is less than that in scenario 5. The phrase of the comparison of variation has been deleted in the revised manuscript, because in this sentence we focus on the advantage of scenario 4, i.e., the improvement in median NSE$_{sqrt}$ performance. This sentence has been modified as follows:

Scenario 4 had a higher median NSE$_{sqrt}$ performance than scenario 5 in catchments 225219 and 405264, and was slightly inferior than the latter in catchment 405219, which indicates the validity of the time-varying scheme for improving the model performance. Please refer to lines 406-410, page 21.

B12: P17.L362. As far as I can see, it has only the highest median value for catchment 225219.

**Reply:** We apologize for our mistakes. These sentences have been modified as follows:

In the DSST scheme of calibrating in the dry period and verifying in the non-dry period, scenario 3, which both considered spatial coherence of regression parameters $\beta$ and $\omega$ between different catchments, exhibited the highest median NSE$_{sqrt}$ for all catchments, had the smallest fluctuation range in two catchments (405219 and 405264) and is the second smallest scenario in catchment 22519 during the verification period. In the other DSST scheme, scenario 3 exhibited the smallest fluctuation range of NSE$_{sqrt}$ estimate for all catchments, showed the highest median value in catchment 225219, and was the second best scenario in the other two catchments (405219 and 405264) during the verification period. Please refer to lines 436-441, pages 22-23.

B13: P18.L375. The performances in the verification period seem higher to me? What do you mean calibrated performances were inferior?

**Reply:** We are sorry for the unclearness. In the scheme of calibration in dry period and verification in non-dry period, it is true that the NSE $_{sqrt}$ during the verification period is higher than that in the calibration period. However, the projection performance calibrated using a contrasting climatic condition was inferior to the simulation performance that was directly calibrated from the climatic condition, compared with Figures 5(a) and 6(b), or Figures 6(a) and 5(b). For example, the

NSE$_{sqrt}$ performance in Figure 6(b) is inferior to which in Figure 5(a). In the other words, for the non-dry period: $NSE_{sqrt,Figure\ 5(a)} > NSE_{sqrt,Figure\ 6(b)}$; for the dry period: $NSE_{sqrt,Figure\ 6(a)} > NSE_{sqrt,Figure\ 5(b)}$.

These sentences have been modified as follows in the revised manuscript:

"However, the projection performance calibrated using a contrasting climatic condition was inferior to the simulation performance that was directly calibrated from the climatic condition, compared with Figures 5(a) and 6(b), or Figures 6(a) and 5(b). For example, the NSE$_{sqrt}$ performance in Figure 6(b) is inferior to which in Figure 5(a)." Please refer to lines 427-431, page 22.

B14: P.18L375-377. This is not true for catchment 225219
**Reply:** We apologize for our mistakes. Follow the referee's comment, this sentence has been modified to specify the problem, which is as follows:

By comparing scenarios in the calibration period, it was found that scenarios 4 and 5 exhibited the highest performance in two of three catchments (405219 and 405264), followed successively by scenario 3, scenario 2, and scenario 1.
Please refer to lines 431-434, pages 22.

B15: P18.L379. The ranges seem not very different between scenarios 4 and 5, only slightly.
**Reply:** Thanks. These sentences have been modified as follows:

During the verification period, the median NSE$_{sqrt}$ performance in scenario 4 was 0.80% higher than scenario 5, however, the variation range in scenario 4 was 53% wider than the latter. In the DSST scheme of calibrating in the dry period and verifying in the non-dry period, scenario 3, which considered both spatial coherence of regression parameters $\beta$ and $\omega$ between different catchments, exhibited the highest median NSE$_{sqrt}$ for all catchments, had the smallest fluctuation range in two catchments (405219 and 405264) and is the second smallest scenario in catchment 22519 during the verification period.
Please to lines 434-441, pages 22-23.

B16: P18.L379-380. It's not very obvious that scenario 3 has a higher median performance for catchment 405264.
**Reply:** Thanks. This sentence has been deleted.

B17: P18.L382 This is not very obvious to me.
**Reply:** Thank you. These sentences have been modified. Please refer to response to comment 15.

B18: P18.L394. Compared –> comparing
**Reply:** Thanks. Change has been made as suggested.

B19: P20.L438. Is omega for scenario 4 not the lowest in all cases? Or do you mean the absolute values?

**Reply:** We apologize for our mistakes. This should be regression parameter $\beta$ in this place rather than $\omega$. The catchment averages of the median estimates of $\beta$ in the first three scenarios are 2.78, -4.91, and 9.26 respectively, while that in the fourth scenario is much larger, reached at -39.20. Scenario 3, which considers both spatial coherence of regression parameters $\beta$ and $\omega$, has the narrowest interval of $\beta$ for all catchments, followed successively by scenario 1 (only considered the spatial coherence of the regression parameter $\beta$), scenario 2 (only regression parameter $\omega$ was spatially coherent), and scenario 4 (no regression parameter was spatially coherent). With regards to the regression parameter $\omega$, which denotes the frequency of the sine function (in the lower figures of Figures 8 and 9), its median estimates and variation ranges in both four scenarios differ slightly. The former reached a catchment average of 0.19,0.20,0.19,0.17 for different scenarios.

B20: Figure 2. Please define all symbols and abbreviations in the figure.
**Reply:** Thanks. All symbols and abbreviations in the Figure 2 have been defined in the revised manuscript.

B21: Figure 5,6: I would suggest to plot the boxes for calibration and verification next to each other. It's easier to see whether there is an improvement or not. Please also add the units (also when a unitless number is presented)
**Reply:** Thanks. Changes have been made as suggested.

B22: Figure 7. Please make the labels and text bigger.
**Reply:** Thanks. Changes have been made as suggested.

B23: Figure 8, 9. Maybe use the same colors for the scenarios in both plots. What are the units of beta and omega?
**Reply:** Thanks for thoughtful comment. Similar comment has been raised by Referee #1 (see comment A1). The same color for the scenarios in both plots has been used. Both $\beta$ and $\omega$ are dimensionless. Figures 8 and 9 have been modified as violin plots in the revised manuscript (see response to A1).

---

## Referee Report (RR1)

The authors generally addressed the concerns of the reviewers (I am referring here mostly to the concerns expressed by myself, Anonymous Reviewer 1) and the resulting paper represent an improvement of the previous version.

Nevertheless, there are still some points to discuss:

1. There are still some imprecisions in the naming of the characteristics of the sinusoidal functions. In the equation $y = a \sin(bx + c)$, $a$ represent the amplitude, $b$ the frequency, and $c$ the phase. The period $T$ can be calculated using the equation $T = 2\pi/b$. Please keep it in mind and check carefully the paper before submission since there are still some imprecision (e.g. line 523)

2. Since $\theta_1 = \alpha + \beta \sin(\omega t)$ the combination expressed in line 224 $\alpha = \beta = \omega = 0$ produces $\theta_1 = 0$ for any value of $t$. I am still convinced that, since $\theta_1$ represent a storage in the model, setting it equal to 0 it is not a good choice. I think that the right values to express what you mean are $\beta = 0$, $\alpha = $ const and the value of $\omega$ becomes irrelevant

3. I still have some doubts on the value of the parameter $\omega$: as you write in the paper, your sinusoidal function oscillates with a period around 40 days: I think that the objective of using time varying parameters was to capture seasonality effects (e.g. high storage in winter and low in summer); is it what you want to represent with your model? If yes, I would expect the period to be around 365 days and not 10 times less

4. Line 261: it would be beneficial to list what are the "unknown quantities"

I think that overall the paper has made one step further towards publication. The only mayor concern that I have against it is verifying if the model is doing what it was meant to (see point 3).

---

## Referee Report (RR2)

**Review of *"Improving hydrological projection performance under contrasting climatic conditions using spatial coherence through a hierarchical Bayesian regression framework"* by Pan et al.**

De study of Pan et al. applies a hierarchical bayesian framework in three Australian catchments. The HB-framework involves estimating the spatial and temporal coherence of model parameters by a regression equation. Five scenarios are tested in the study, with different degrees of spatial and temporal coherence. The authors conclude that the time varying setting improved performance but increased uncertainty, spatial coherence reduced uncertainty and that performance decreased when parameters were transfered from dry periods to wet periods.

The article shows quite some improvements compared to the previous version of the manuscript. I am also happy that the authors addressed myprevious comments and made improvements based on that.

Therefore, I appreciate the effort of the authors to clarify their method, but, to be honest, I'm still a bit confused.  It may be just me, and my lack of knowledge here, but I still wonder where the Gaussian distributions come in. The authors state that, in paragraph 2.4.2 and their response, that all parameters, including the regression parameters (L.296-298), are sampled simultaneously and come from a uniform distribution (L.310).  So where are the Gaussian distributions coming in? Are the regression parameters not samples from these Gaussian distributions, which are defined by the hyper-parameters?  So, shouldn't it be 1) sample hyper-parameters and spatially irrelevant parameters from a uniform distribution, and 2) sample the spatially relevant parameters from the Gaussian distributions? I believe this is mainly a textual issue which the authors can easily clarify, because when I look at S1 in the supplement, as an example, beta is not mentioned for scenario 1, which makes me think it is sampled based on the Gaussian defined by the hyper-parameters.  So can you clarify this a bit more?

I am happy with the additional criteria of mean annual maximum flow and mean annual minimum flow. However, as described and presented in the tables now, these are just the numbers obtained by the model. How are these values compared with the observations? Or are these numbers the error between modelled and observed annual maximum  and minimum flow?

With regard to my previous remark on the choice of the Gaussian distribution, and the authors response on that, I fully understand the reason why the authors used a Gaussian distribution. However, in my point of view, it is just really interesting to look a bit further, as there should be a physical reason why storage capacities (and/or their trend) are spatially related by a Gaussian distribution. Maybe the authors can just add some thoughts on the physical reasons for their findings in the discussion, as this is a bit missing in general.

Concluding, I am happy the authors found most of my comments useful and addressed all of them. When the authors also address the minor issues raised above, I would recommend publication of the manuscript.

**Minor comments**

Generally, the terms dry and wet period are a bit confusing, as it makes me think of a wet season and dry season. The authors mean a longer period of dry years and wet years though. Maybe it is better to replace "dry period" and "wet period" throughout the manuscript with "dry years" and "wet years".

L176. Why should the anomaly be less than 25%?

L224. When theta is constant, I think alpha needs to have a value, as written alpha is also zero, and then theta becomes zero too.

L274. Please add what T and t represent for completeness.

L328. Please describe all your variables for completeness.

L333. Please describe all your variables for completeness.

L335. Please describe all your variables for completeness.

L347. Do you mean potential evaporation? Or do you use an estimate of actual evaporation?

L381-387. This paragraph seems a bit odd to me. Why divide your timeseries into a dry and wet period if the change cannot be larger then 11%? In my view, if you want to test the robustness of the model you should actually even have higher differences then 11%. The discussed results of Vaze et al. (2010) only proof that those models were not robust and can not model extreme cases of droughts. Or arguing from the other end, if the change between rainfall in the dry and wet period is just hypothetically 0.0001%, what is the whole point of splitting into dry and wet periods?

L396. What do you mean with the variation?

L411. This is scenario 4 still, correct?

L417. Aren't scenarios 4 and 5 both higher than scenario 3 for 405264? Hard to see in the plot.

L427. Do you mean Figure 6?

L427-431. I agree, but it's quit normal that the period used for calibration outperforms the verification.

L439. This is hard to see in violin plots

L440. Isn't the range for 405219 larger?

L441. Do you mean the second smallest in variation?

L442. Are you comparing here just scenario 4 with 5, or 1-4 with 5? In the last case, this statement is not always true, as far as I can see, and following the discussion above.

L460. When compared→ when comparing

L468. How can I see this? It is just made bold, please add the observed values or present an error measure in the table.

L471. Idem as above.

Figs. 5,6,8,9. I am not sure if the violin plots are much more helpful compared to the boxplots in the previous version of the manuscript. It becomes more complicated to find the median values, which the authors often refer to, especially as some posterior ranges are note nicely equally distributed. I believe the median values are a dashed black line, but this is hard to see. I would suggest different colors or line types for for example the medians.

Fig. 5 Varification –> Verification

Fig.7. I guess the bars are the reference 10-year flow, but please make sure it is clear which x-axis (left-right) belongs to which graph. Add the bars also to the legend.

---

## Author Response (AR2)

Responses to all the Referees:

Dear Editor and Reviewers:

We sincerely appreciate the comments from the Editor and Referees. Detailed responses to the comments raised by the Editor and Referees have been presented below. In the following Responses, there are several simplified labels. For example, A1 represents the comment 1 made by Reviewer #1, B1 represents the comment 1 made by Reviewer #2. Please also note that the page and line numbers mentioned in reviewers comments refer to the original version, while in the authors' response they refer to the revised version.

Sincerely,
Pan Liu, PhD, Professor
School of Water Resources and Hydropower
State Key Laboratory of Water Resources and Hydropower Engineering Science
Wuhan University, Wuhan, Hubei Province, 430072, P. R. China
**E-mail:** liupan@whu.edu.cn

Responses to Editor:
Although the reviewers are generally positive with the paper, there are still some points of concern. Please address the remaining comments thoroughly before resubmission.
**Reply:** We sincerely appreciate the comments from the Editor. Great efforts have been made to address both reviewers' comments.

Responses to Referee #1:
The authors generally addressed the concerns of the reviewers (I am referring here mostly to the concerns expressed by myself, Anonymous Reviewer 1) and the resulting paper represent an improvement of the previous version. Nevertheless, there are still some points to discuss:
A1: There are still some imprecisions in the naming of the characteristics of the sinusoidal functions. In the equation $y = a \sin(bx + c)$, $a$ represent the amplitude, $b$ the frequency, and $c$ the phase. The period $T$ can be calculated using the equation $T = 2\pi/b$. Please keep it in mind and check carefully the paper before submission since there are still some imprecision (e.g. line 523)
**Reply:** Thank you for your comments.
(1) Follow the reviewer's comment, the "regression parameter $\omega$" and "regression parameter $\beta$" have been modified as "frequency $\omega$" and "amplitude $\beta$" in the revised manuscript. Please refer to line 245, 249, 266 on page 12, line 529 on page 25, *etc*. in the revised manuscript.
(2) The "phase $T$" has been modified as "the period $T$". Please refer to line 548, 550, 551 on page 26.

A2. Since $\theta_1 = \alpha + \beta sin(\omega t)$ the combination expressed in line 224 $\alpha = \beta = \omega = 0$ produces $\theta_1 = 0$ for any value of $t$. I am still convinced that, since $\theta_1$ represent a storage in the model, setting it equal to 0 it is not a good choice. I think that the right values to express what you mean are $\beta = 0$, $\alpha$ = const and the value of $\omega$ becomes irrelevant.

**Reply:** We are really sorry for this error. According to the definition of the GR4J model (Perrin et al., 2003), the value of model parameter $\theta_1$ should be larger than zero for any value of $t$. Therefore, this sentence has been modified as "According to the definition of the GR4J model (Perrin et al., 2003), the value of $\theta_1$ must be a positive value. If model parameter $\theta_1$ is constant then $\beta = 0$ and $\alpha > 0$ suffice in Eq.1 Meanwhile, the value of $\omega$ becomes irrelevant, thus the resulting model simplifies to a stationary hydrological model". Please refer to lines 224-228 on page 11 in the revised manuscript.

A3. I still have some doubts on the value of the parameter $\omega$: as you write in the paper, your sinusoidal function oscillates with a period around 40 days: I think that the objective of using time varying parameters was to capture seasonality effects (e.g. high storage in winter and low in summer); is it what you want to represent with your model? If yes, I would expect the period to be around 365 days and not 10 times less.

**Reply:** Thank you.

     (1) The purpose of introducing parameter $\omega$ in Eq.1 was to represent the periodical variation of model parameter $\theta_1$, which might be monthly, seasonal, every half a year, or annual, etc. It should be noted that seasonality is only one of the potential time-varying schemes.

(2) Based on the results from the studied catchments, the mean periods of different scenarios are within 24~47 days, nearly every 0.8~1.5 month. In addition, we used the Hilbert-Huang Transform method (Huang et al., 1998) to identify the potential periods of the series of several climate variables (including the daily rainfall, daily potential evapotranspiration, daily maximum temperature and daily minimum temperature in the studied catchments). It was found that these series have periods of 22.2~49.1 days. Thus, we guess that the potential periods of these climate variables may be the possible reasons for the periods of time-varying parameters. It also should be mentioned that the adopted Hilbert spectrum method is one of the most popular methods for analyzing nonlinear and non-stationary data. Huang et al. (1999) indicated that this method is better than the Fourier transform method and Wavelet Transform method in processing nonlinear and non-stationary data. These analyses also have been added in the revised manuscript. Please refer to lines 552-562 on page 26.

Reference:
Huang, N. E., Shen, Z., Long, S. R., Wu, M. L. C., Shih, H. H., Zheng, Q. N., Yen, N. C., Tung, C. C., and Liu, H. H.: The empirical mode decomposition and the Hilbert

spectrum for nonlinear and non-stationary time series analysis, Proc. R. Soc. A-Math. Phys. Eng. Sci., 454, 903-995, 10.1098/rspa.1998.0193, 1998.

Huang, N. E., Shen, Z., and Long, S. R.: A new view of nonlinear water waves: The Hilbert spectrum, Annu. Rev. Fluid Mech., 31, 417-457, 10.1146/annurev.fluid.31.1.417, 1999.

A4. Line 261: it would be beneficial to list what are the "unknown quantities"

**Reply:** Thank you. Unknown quantities consist of model parameters ($\theta_2, \theta_3$, and $\theta_4$), regression parameters $\alpha$, $\beta$ and $\omega$, and hyper-parameters $\mu_2$, $\sigma_2$, $\mu_3$ and $\sigma_3$. Details about the "unknown quantities" also have been added in the revised manuscript. Please refer to lines 271-277 on page 13.

A5. I think that overall the paper has made one step further towards publication. The only major concern that I have against it is verifying if the model is doing what it was meant to (see point 3).

**Reply:** Thank you for your comments.

(1) As illustrated on lines 211-218 in pages 10, regression parameter $\omega$ is used to account for the periodical variation (rather than seasonality) of model parameter $\theta_1$. Based on the results from the studied catchments, the mean periods of are within 24~47 days, nearly every 0.8~1.5 month. Furthermore, the seasonality effect is not significant in these catchments.

(2) We agree with reviewer #1 that the seasonality effect (e.g. high storage in winter and low in summer) of model parameter $\theta_1$ is a reasonable guess before calculation. However, our results did not prove this point. The reasons behind might be that since a daily hydrological model is adopted, the remarkable day-to-day variation in climate variables (e.g., daily rainfall) masked the seasonality of time-varying parameter.

(3) Explanations also have been made in response to comment A3, please refer to comment A3.

Responses to Referee #2:

De study of Pan et al. applies a hierarchical Bayesian framework in three Australian catchments. The HB-framework involves estimating the spatial and temporal coherence of model parameters by a regression equation. Five scenarios are tested in the study, with different degrees of spatial and temporal coherence. The authors conclude that the time-varying setting improved performance but increased uncertainty, spatial coherence reduced uncertainty and that performance decreased when parameters were transferred from dry periods to wet periods.

The article shows quite some improvements compared to the previous version of the manuscript. I am also happy that the authors addressed my previous comments and made improvements based on that.

B1: Therefore, I appreciate the effort of the authors to clarify their method, but, to be honest, I'm still a bit confused. It may be just me, and my lack of knowledge here, but I still wonder where the Gaussian distributions come in. The authors state that, in paragraph 2.4.2 and their response, that all parameters, including the regression parameters (L.296-298), are sampled simultaneously and come from a uniform distribution (L.310). So where are the Gaussian distributions coming in? Are the regression parameters not samples from these Gaussian distributions, which are defined by the hyper-parameters? So, shouldn't it be 1) sample hyper-parameters and spatially irrelevant parameters from a uniform distribution, and 2) sample the spatially relevant parameters from the Gaussian distributions? I believe this is mainly a textual issue which the authors can easily clarify, because when I look at S1 in the supplement, as an example, beta is not mentioned for scenario 1, which makes me think it is sampled based on the Gaussian defined by the hyper-parameters. So can you clarify this a bit more?

**Reply:** We apologize for unclear descriptions and thank you for your suggestions. Now clarifications have been made in the revised manuscript, please refer to lines 304-308 on pages 14-15. Hyper-parameters and spatially irrelevant parameters are sampled from the uniform distributions, while spatially relevant parameters are sampled from the Gaussian distributions. It should be noted that the prior ranges for unknown quantities are different.

(1) For instance, in scenario 1, regression parameter $\beta$ is not sampled from the uniform distribution but it is the output from the Gaussian distribution $\beta = N(\mu_2, \sigma_2{}^2)$, in which hyper-parameters $\mu_2$ and $\sigma_2$ are sampled from the uniform distributions with different ranges. While, other unknown quantities, i.e., model parameters ($\theta_2$, $\theta_3$, and $\theta_4$), regression parameters $\alpha$ and $\omega$ (no $\beta$), are sampled from the uniform distributions. In conclusion, in scenario 1 (S1), $\beta$ is estimated from the Gaussian distribution and $\omega$ is sampled from the uniform distribution, while hyper-parameter $\mu_3$ and $\sigma_3$ are not considered.

(2) In scenario 2, $\omega$ is not sampled from the uniform distribution but it is the output from the Gaussian distribution $\omega = N(\mu_3, \sigma_3{}^2)$, in which hyper-parameter $\mu_3$ and $\sigma_3$ are sampled from the uniform distributions. In contrast to scenario 1, $\beta$ is sampled from the uniform distribution, while $\mu_2$ and $\sigma_2$ do not exist in scenario 2. More details about the unknown quantities of different scenarios are presented in Supplement.

B2: I am happy with the additional criteria of mean annual maximum flow and mean annual minimum flow. However, as described and presented in the tables now, these are just the numbers obtained by the model. How are these values compared with the observations? Or are these numbers the error between modelled and observed annual maximum and minimum flow?

**Reply:** Thank you for your comments.

(1) These numbers represent the observed values and modeled values. The scenario with the minimum absolute difference between its modeled and observed

values is the best scenario.

(2) The percentage errors between modeled and observed annual maximum and minimum flow have been added to Tables 6 and 7. The scenarios with the minimum absolute errors are recognized as the best scenarios. Please refer to Tables 6 and 7 in the revised manuscript.

B3: With regard to my previous remark on the choice of the Gaussian distribution, and the authors response on that, I fully understand the reason why the authors used a Gaussian distribution. However, in my point of view, it is just really interesting to look a bit further, as there should be a physical reason why storage capacities (and/or their trend) are spatially related by a Gaussian distribution. Maybe the authors can just add some thoughts on the physical reasons for their findings in the discussion, as this is a bit missing in general.

Concluding, I am happy the authors found most of my comments useful and addressed all of them. When the authors also address the minor issues raised above, I would recommend publication of the manuscript.

**Reply:** Thank you.

(1) As a response to reviewer #2 in the previous round, the Gaussian distribution is one of the widely used distributions for describing the prior layer within the HB framework and has been applied in many previous studies, such as Sun et al (2015, 2016) and Chen et al (2014).

(2) The studied catchments are adjacent and have similar climate characteristics, e.g., the similar precipitation pattern and drought period anomaly (see Table 3). It is expected to have similar variation pattern of the catchment storage capacity for these catchments. However, there are still uncountable factors that may have impacts on the spatial coherence between adjacent catchments, which makes the coherence between β and ω tend to converge a central value but with finite variance (central limit theorem). The Gaussian distribution is the most likely distribution to describe the variables that obey the central limit theorem. These discussions also have been added on lines 254-258 on page 12.

Minor comments

B4: Generally, the terms dry and wet period are a bit confusing, as it makes me think of a wet season and dry season. The authors mean a longer period of dry years and wet years though. Maybe it is better to replace "dry period" and "wet period" throughout the manuscript with "dry years" and "wet years".

**Reply:** Thank you. Changes have been made as suggested. "dry period" and "non-dry period" have been modified as "dry years" and "wet years" in the revised manuscript.

B5: L176. Why should the anomaly be less than 25%?

**Reply:** Thank you. Sorry for this typo.

(1) According to Saft et al. (2015), the number here should be "-5%" rather than "25%". The sentence has been modified as "mean dry years anomaly<-5%.". The mean dry years' anomaly should be smaller than -5%, which is to identify dry years with more

than 5% less rainfall than wet years.

(2) It should be noted that "-5%" is an experimental parameter in Saft et al. (2015). Saft et al. (2015) tested several algorithms for dry years delineation, which considered different combinations of dry run length, dry run anomaly and various boundary criteria, and found that one of the algorithms, i.e., the method adopted in our study, showed marginal dependence on the algorithm and the main results were robust to different algorithms. Please refer to lines 166-177 on page 8.

B6: L224. When theta is constant, I think alpha needs to have a value, as written alpha is also zero, and then theta becomes zero too.
**Reply:** Thank you. As a response to comment A2 by reviewer #1, this sentence has been modified as follows.

According to the definition of the GR4J model (Perrin et al., 2003), the value of $\theta_1$ must be a positive value. If model parameter $\theta_1$ is constant, $\beta = 0$ and $\alpha > 0$ suffice in Eq.1. Meanwhile, the value of $\omega$ becomes irrelevant. Thus, the resulting model simplifies to a stationary hydrological model.
Please also refer to lines 224-228 on page 11.

B7: L274. Please add what T and t represent for completeness.
**Reply:** We apologize for this negligence. The meanings of T and $t$ have been added in the revised manuscript. T represents the number of the time series while $t$ represents the time step. Please refer to lines 287-289 on pages 13-14.

B8: L328. Please describe all your variables for completeness.
**Reply:** Thank you. The meanings of all variables in Eq.7 have been added in the revised manuscript. $Q_{sim}(t)$ and $Q_{obs}(t)$ represent the simulated and observed daily streamflow values of the $t^{th}$ day, respectively. $T$ refers to the length of the study period. Please refer to lines 334-336 on page 16 in the revised manuscript.

B9: L333. Please describe all your variables for completeness.
**Reply:** Thank you. The meanings of all variables in Eq.8 have been added in the revised manuscript. $p$ refers to probability, q represents the observations of streamflow and $\xi$ denotes the time series of model input, e.g., rainfall and potential evapotranspiration. Please refer to lines 348-349 on page 17.

B10: L335. Please describe all your variables for completeness.
**Reply:** Thank you. The meanings of all variables in Eq.9 have been added in the revised manuscript. Please refer to lines 348-349 on page 17.

B11: L347. Do you mean potential evaporation? Or do you use an estimate of actual evaporation?
**Reply:** We are sorry for this mistake. It should be "potential evapotranspiration" rather than "evapotranspiration" in this sentence. Change has been made in the revised manuscript. Please refer to line 363 on page 17.

B12: L381-387. This paragraph seems a bit odd to me. Why divide your timeseries into a dry and wet period if the change cannot be larger then 11%? In my view, if you want to test the robustness of the model you should actually even have higher differences then 11%. The discussed results of Vaze et al. (2010) only proof that those models were not robust and can not model extreme cases of droughts. Or arguing from the other end, if the change between rainfall in the dry and wet period is just hypothetically 0.0001%, what is the whole point of splitting into dry and wet periods?

**Reply:** Thank you. The "11%" is the catchment average reduction from wet years to dry years of the three study catchments. It should be noted that it is statistical results rather than evaluation criteria to divide the time series into dry years and wet years. The criteria for segmenting the time series have been presented in section 2.1. Please refer to lines 166-177 on page 8 in the revised manuscript.

Furthermore, this paragraph has been modified as follows:

In terms of changes in rainfall, on average catchments had an 11% reduction from the wet years to the dry years (Table 3). Meanwhile, these catchments experienced a 17.6% decrease in runoff during the dry periods, which is more severe than the reduction in rainfall. The similar findings can be derived out from the comparison of runoff coefficients of different periods, that is, all catchments experienced a decrease in its runoff coefficients during the dry years.

Please refer to lines 397-402 on page 19.

B13: L396. What do you mean with the variation?

**Reply:** Thank you. This sentence has been modified as "Furthermore, the magnitude of performance loss increases along with the variation in rainfall between the calibration and verification periods." Please refer to lines 410-412 on pages 19-20.

B14: L411. This is scenario 4 still, correct?

**Reply:** Yes. To improve the clarity of the manuscript, this sentence has been modified as "However, the introduction of additional regression parameters ($\alpha$, $\beta$ and $\omega$) at the same time amplified the model projection uncertainty in two of three catchments (225219 and 405264) when comparing results from scenarios 4 and 5." Please refer to lines 428-431 on page 20.

B15: L417. Aren't scenarios 4 and 5 both higher than scenario 3 for 405264? Hard to see in the plot.

**Reply:** Thank you. During the verification period, the median estimate of scenario 4 is a little higher than that of scenario 3. Conversely, the median estimate of scenario 5 is inferior to that in scenario 3. In addition, the Figures have been improved, white dots have been added to represent the median estimates of the results in the violin plots.

B16: L427. Do you mean Figure 6?

**Reply:** We are sorry for this typo. It should be Figure 6 rather than Figure 5. Changes

have been made in the revised manuscript accordingly.

B17: L427-431. I agree, but it's quite normal that the period used for calibration outperforms the verification.
Reply: Thank you. We agree with reviewer #2 that it's quite normal that the period used for calibration outperforms the verification. The purpose of this sentence is to verify the rationality of the results, which is the basis for further analysis.

B18: L439. This is hard to see in violin plots
Reply: Thank you. The violin plots have been modified to improve clarity. The white dots have been added to represent the median estimates of the results in the violin plots.

B19: L440. Isn't the range for 405219 larger?
Reply: We are sorry for this typo. It should be catchment 225219 rather than 405219 here. The sentence has been modified in the revised manuscript. Please refer to lines 463-474 on page 22.

B20: L441. Do you mean the second smallest in variation?
Reply: Yes. The phrase "in variation" now has been added in this sentence. Please refer to the response to comment B19.

B21: L442. Are you comparing here just scenario 4 with 5, or 1-4 with 5? In the last case, this statement is not always true, as far as I can see, and following the discussion above.
Reply: Thank you. In this sentence, we compared the results from scenarios 4 and 5 rather than compared scenarios 1-4 with 5, because the only difference (Principle of a single variable) between these two scenarios is that scenario 4 adopted the time-varying parameters while scenario 5 adopted the temporal invariant parameters. We did not compare scenarios 1-3 with 5 here either, because the former not only adopted the time-varying parameters but also used the spatial coherence, and it would be not easy to distinguish the impacts by each individual.

    This sentence has been modified as follows:

    These results demonstrate that the time-varying scheme (scenario 4) for model parameters improved the median $NSE_{sqrt}$ performance but also amplified the projection uncertainty compared with the results from the stationary scheme (scenario 5) for model parameters.

    Please refer to lines 460-463 on page 22.

B22: L460. When compared→ when comparing

Reply: Thank you. Change has been made as suggested. Please refer to line 486 on page 23.

B23: L468. How can I see this? It is just made bold, please add the observed values or

present an error measure in the table.

**Reply:** Thank you. The observed MinF and MaxF were presented at the first row of Tables 6 and 7. As a response to comment B2 by reviewer #2, Tables 6 and 7 have been modified, the % variation between the modeled value and the observed value have been presented in the revised manuscript. The scenarios with the least absolute variation between the modeled values and the observed values are recognized as the best scenarios.

Further explanations have been made in the revised manuscript, please refer to lines 358-360 on page 17.

B24: L471. Idem as above.

**Reply:** Thank you. Changes have been made to address the reviewer's comment. Please refer to the response to comment B23 by Reviewer #2.

B25: Figs. 5,6,8,9. I am not sure if the violin plots are much more helpful compared to the boxplots in the previous version of the manuscript. It becomes more complicated to find the median values, which the authors often refer to, especially as some posterior ranges are noted nicely equally distributed. I believe the median values are a dashed black line, but this is hard to see. I would suggest different colors or line types for example the medians.

**Reply:** Thank you. Follow the reviewer's comment. whites dots that represent the median estimates of the results have been added in the revised figures. We hope that it can help to relieve the reviewer's doubt. Please refer to Figure 5,6,8 and 9 in the revised manuscript.

B26: Fig. 5 Varification –> Verification.

**Reply:** Thank you. Changes have been made as suggested.

B27: Fig.7. I guess the bars are the reference 10-year flow, but please make sure it is clear which x-axis (left-right) belongs to which graph. Add the bars also to the legend.

**Reply:** Thank you. The main scales of the figures have been added, which are helpful to find out which x-axis belongs to which graph easily. In addition, the bars have been added in the revised Figure 7. Please refer to Page 44.